# Understanding Private Learning From Feature Perspective

Meng Ding[1]  Mingxi Lei[1]  Shaopeng Fu[2]  Shaowei Wang[3]  Di Wang[2]  Jinhui Xu[4]

## Abstract

Differentially private Stochastic Gradient Descent (DP-SGD) has become integral to privacy-preserving machine learning, ensuring robust privacy guarantees in sensitive domains. Despite notable empirical advances leveraging features from non-private, pre-trained models to enhance DP-SGD training, a theoretical understanding of feature dynamics in private learning remains underexplored. This paper presents the first theoretical framework to analyze private training through a feature learning perspective. Building on the multi-patch data structure from prior work, our analysis distinguishes between label-dependent feature signals and label-independent noise—a critical aspect overlooked by existing analyses in the DP community. Employing a two-layer CNN with polynomial ReLU activation, we theoretically characterize both feature signal learning and data noise memorization in private training via noisy gradient descent. Our findings reveal that (1) Effective private signal learning requires a higher signal-to-noise ratio (SNR) compared to non-private training, and (2) When data noise memorization occurs in non-private learning, it will also occur in private learning, leading to poor generalization despite small training loss. Our findings highlight the challenges of private learning and prove the benefit of feature enhancement to improve SNR. Experiments on synthetic and real-world datasets also validate our theoretical findings.

[1]Department of CSE, University at Buffalo, Buffalo, NY, USA [2]Division of CEMSE, KAUST, Thuwal, Saudi Arabia [3]Institute of Artificial Intelligence and Blockchain, Guangzhou University, Guangzhou, China [4]School of Information Science and Technology, University of Science and Technology of China, Hefei, China. Correspondence to: Di Wang <di.wang@kaust.edu.sa>, Jinhui Xu <jhxu@ustc.edu.cn>.

*Proceedings of the 43rd International Conference on Machine Learning*, Seoul, South Korea. PMLR 306, 2026. Copyright 2026 by the author(s).

## 1. Introduction

Differentially private (DP) learning has emerged as a cornerstone of privacy-preserving machine learning, addressing growing concerns about data privacy in sensitive domains such as healthcare (Lundervold & Lundervold, 2019; Chlap et al., 2021; Shamshad et al., 2023), finance (Ozbayoglu et al., 2020; Bi & Lian, 2024), and user-centric applications (Oroojlooy & Hajinezhad, 2023). Differential Privacy, introduced by (Dwork et al., 2006), provides robust privacy guarantees by limiting the impact of any individual data points on the model's output. Among DP learning methods, differentially private stochastic gradient descent (DP-SGD) (Abadi et al., 2016) has emerged as a canonical algorithm for training private machine learning models.

However, DP-SGD often comes with a significant cost in model accuracy (Shokri & Shmatikov, 2015; Abadi et al., 2016; Bagdasaryan et al., 2019). To improve the performance, recent work (Tramer & Boneh, 2020) shows that DP-SGD training benefits from handcrafted features and can achieve better performance by leveraging features learned from public data in a similar domain. Similarly, Tang et al. (2024b) highlights the advantages of transferring features learned from synthetic data to private training, while Sun et al. (2023) and Bao et al. (2023) illustrate the importance of feature preprocessing in private learning. These findings suggest that improving feature quality is essential for effective private learning. Benefiting from this principle and the advent of large-scale foundation models, DP-SGD has demonstrated significant performance boosts by learning features from non-private models pre-trained on large public datasets (Tramer & Boneh, 2020; Li et al., 2021; De et al., 2022; Arora & Ré, 2022; Kurakin et al., 2022; Mehta et al., 2023; Nasr et al., 2023; Tang et al., 2024a; Bu et al., 2024b).

Despite the empirical success of DP-SGD from enhanced features, the theoretical understanding of these phenomena remains in its infancy. Previous work on DP learning has primarily focused on analyzing the utility bounds of private models, such as DP-SGD and its variants, with a particular emphasis on both convex (Bassily et al., 2014; Wang et al., 2017; Bassily et al., 2019; Feldman et al., 2020; Song et al., 2020; Su & Wang, 2021; Asi et al., 2021; Bassily et al., 2021b; Kulkarni et al., 2021; Tao et al., 2022; Su et al., 2023; 2024) and non-convex models (Zhang et al., 2017;

Wang et al., 2017; Wang & Xu, 2019; Zhang et al., 2021; Bassily et al., 2021a; Wang et al., 2023; Ding et al., 2024), leaving the role and explanation from the feature perspective largely unexplored.

Only two recent works have studied the theoretical aspects of features in private learning, both with several limitations. Sun et al. (2023) confines its analysis to simple tasks using linear classification models without addressing applicability to neural networks. Wang et al. (2024) investigates feature shifts during private fine-tuning of the last layer under the framework of neural collapse (Papyan et al., 2020), simplifying the private model with the assumption of the equiangular tight frame (ETF). Furthermore, both works focus exclusively on utility, providing limited explanations of the learning dynamics of features in private learning. We will provide more discussions later.

In this paper, we develop a novel theoretical framework that studies the private learning dynamics from a feature learning perspective in noisy gradient descent (NoisyGD), a simple version of DP-SGD. Inspired by the structure of image data, we consider a data distribution modeled as a multiple-patch structure, $\mathbf{x} = [y \cdot \mathbf{v}, \boldsymbol{\xi}] \in (\mathbb{R}^d)^2$, where $y \in \{+1, -1\}$ is the label, $\mathbf{v}$ represents the useful label-dependent feature signals, $\boldsymbol{\xi}$ refers to label-independent data noise randomly sampled from a Gaussian distribution with standard deviation $\sigma_\xi$ and $d$ is the dimension. For example, as illustrated in Figure 1, the wheel serves as a feature for the class 'car,' while the background acts as label-independent data noise.

Beyond the linear classification model, we utilize a two-layer convolutional neural network (CNN) with a polynomial ReLU activation function: $\sigma(z) = \max\{0, z\}^q$, where $q > 2$ is a hyperparameter. Given a training dataset of $n$ samples, we quantify noisy gradient descent in terms of feature signal learning and data noise memorization, measured through the private model $\mathbf{w}$ with signal and data noise. Specifically, we present the following (informal) results:

**Theorem 1.1** (Informal). *Let* $\mathrm{SNR} := \|\mathbf{v}\|_2/\|\boldsymbol{\xi}\|_2$ [1] *be the signal-to-noise ratio and* $\varepsilon$ *be the privacy budget. Under appropriate conditions, it holds that*

- *When* $\min\{\mathrm{SNR} \cdot n\varepsilon, \mathrm{SNR}^q \cdot n\} \geq \widetilde{\Omega}(1)$*, the private CNN model can capture the feature signal.*

- *When* $\min\{\mathrm{SNR}^{-1} \cdot \varepsilon, \mathrm{SNR}^{-q} \cdot n^{-1}\} \geq \widetilde{\Omega}(1)$*, the private CNN model can capture the data noise.*

---

[1] Our data model is analogous to the patch model in (Cao et al., 2022): the signal vector $\mathbf{v}$ is fixed, while the noise $\boldsymbol{\xi}$ is drawn from an isotropic Gaussian. When the dimension $d$ is large, $\|\boldsymbol{\xi}\|_2 \approx (\mathbb{E}\|\boldsymbol{\xi}\|_2^2)^{1/2}$ by standard concentration, so we treat $\|\mathbf{v}\|_2/\|\boldsymbol{\xi}\|_2$ (and hence $\|\mathbf{v}\|_2/(\sigma_\xi\sqrt{d})$) as the signal-to-noise ratio for ease of discussion similar to (Cao et al., 2022).

Theorem 1.1 demonstrates the two interesting results during private training: **1**) When $\min\{\mathrm{SNR} \cdot n\varepsilon, \mathrm{SNR}^q \cdot n\} \geq \widetilde{\Omega}(1)$ and $n^{-1/2} \leq \varepsilon \leq \mathrm{SNR}^{q-1}$, a lower privacy budget requires a higher SNR compared to standard non-private training to effectively capture the signal, emphasizing the need for feature enhancement to improve SNR. **2**) When data noise memorization occurs in standard non-private learning, it will also occur in private learning as long as $\varepsilon \geq \mathrm{SNR}^{1-q} n^{-1}$.

Moreover, under additional data assumptions, we have the following results:

**Corollary 1.2** (Informal). *Let* $\mathrm{SNR} := \|\mathbf{v}\|_2/\|\boldsymbol{\xi}\|_2$ *be the signal-to-noise ratio and* $\varepsilon$ *be the privacy budget. Under appropriate conditions and assumptions, for any* $\kappa > 0$*, it holds that*

- *When* $\min\{\mathrm{SNR} \cdot n\varepsilon, \mathrm{SNR}^q \cdot n\} \geq \widetilde{\Omega}(1)$*, the training loss can converge to* $\kappa$*, and the trained CNN achieves a test loss of* $6\kappa + \exp((n\varepsilon)^{-1-1/q})$*.*

- *When* $\min\{\mathrm{SNR}^{-1} \cdot \varepsilon, \mathrm{SNR}^{-q} \cdot n^{-1}\} \geq \widetilde{\Omega}(1)$*, the training loss can converge to* $\kappa$*, but the trained CNN incurs a constant-order test loss regardless of the sample size* $n$ *and privacy budget* $\varepsilon$*.*

Corollary 1.2 demonstrate two conclusions. First, in an ideal scenario, private learning can achieve an arbitrarily small training loss, and its test performance is influenced by both the sample size $n$ and privacy budgets $\varepsilon$. Second, even if private learning achieves a small training loss, it may still fail to deliver good test performance, regardless of how the sample size and privacy budget are chosen. This limitation arises because the private model primarily learns label-independent data noise rather than label-dependent feature signals. We summarize our contributions below:

- We present the first theoretical framework of the NoisyGD, a simple version of DP-SGD, dynamic through a feature learning perspective. Building on the multi-patch data structure introduced in Allen-Zhu & Li (2020); Cao et al. (2022), our framework explicitly distinguishes between label-dependent features and label-independent noise—a critical aspect overlooked by existing analyses of DP-SGD. This distinction enables us to introduce the signal-to-noise ratio (SNR) as an essential factor influencing the private training process.

- We provide a detailed theoretical analysis of feature signal learning and data noise memorization in the private setting with a two-layer convolutional neural network model. Specifically, based on the signal-to-noise ratio, we show that **1**) Effective private signal learning requires a higher signal-to-noise ratio compared to

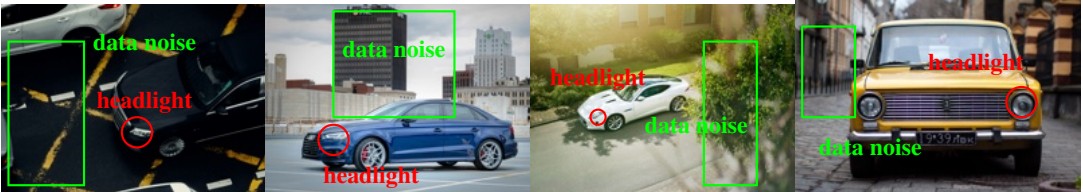

*Figure 1.* Illustration of images with feature signal and data noise.

non-private training. **2**) When data noise memorization occurs in standard non-private learning, it will also occur in private learning as long as $\varepsilon \geq \text{SNR}^{1-q} n^{-1}$. Consequently, the private model fails to generalize well, even when achieving a small training loss, regardless of sample size and privacy budget.

- Our findings underscore the importance of feature enhancement techniques in improving SNR for effective private learning, aligning with the principles established in previous empirical work (Tramer & Boneh, 2020; Sun et al., 2023; Bao et al., 2023; Tang et al., 2024b). We conduct simulation experiments on CNNs and validate our theoretical analysis across various privacy budgets and signal-to-noise ratios.

Due to space constraints, we defer additional related work, extended experimental results, and the conclusion to the Appendix.

**Conflict of Interest Disclosure.** The authors declare no financial conflicts of interest related to this work.

## 2. Related Work

**Theory on Differentially Private Learning** A growing body of work has focused on the differential private optimization problems, including standard results for private empirical risk minimization (Chaudhuri et al., 2011; Bassily et al., 2014; Wang et al., 2017; Wang & Xu, 2019) and private stochastic convex optimization (Bassily et al., 2019; Feldman et al., 2020; Bassily et al., 2021a). These studies have also been extended under various assumptions, such as heavy-tailed data (Wang et al., 2020; Hu et al., 2022; Kamath et al., 2022) and non-Euclidean spaces (Bassily et al., 2021a; Asi et al., 2021; Su et al., 2023). Despite extensive research on DP optimization theory, the theoretical understanding of private deep learning remains largely unexplored, particularly from the feature perspective. Only two recent studies have explored the theoretical aspects of features in private learning. Specifically, Sun et al. (2023) focuses on a linear classification model, whereas we analyze a more challenging two-layer neural network model with a polynomial ReLU activation function. Wang et al. (2024) considers a last-layer model converging to the columns of an equiangular tight frame (ETF), which simplifies the learned features to normal vectors via a rotation map and the model is still in a linear form. In contrast, our approach goes beyond this simplification. Moreover, while Wang et al. (2024) emphasizes feature shift behavior, our work primarily focuses on training dynamics and the importance of feature enhancement.

**Feature Learning Theory** Feature learning has emerged as a powerful framework for the theoretical understanding of deep learning in recent years. It was first introduced by (Allen-Zhu & Li, 2022) to explain the advantages of adversarial training in achieving robustness and was then extended by (Allen-Zhu & Li, 2020) to demonstrate the role of ensemble methods in improving generalization. Then, the framework has been applied to a broad range of model architectures, including graph neural networks (Huang et al., 2023), convolutional neural networks (Cao et al., 2022; Kou et al., 2023), vision transformers (Jelassi et al., 2022; Li et al., 2023), and diffusion models (Han et al., 2024). In addition to architectural studies, the feature learning perspective has also been employed to analyze the behavior of various optimization algorithms and training strategies, such as Adam (Zou et al., 2023), momentum-based methods (Jelassi & Li, 2022), and data augmentation techniques like Mixup (Zou et al., 2023). To the best of our knowledge, this work is the first to examine private learning through the lens of feature learning. Unlike standard learning, private learning introduces additional challenges—most notably, the injection of private noise can destabilize training, necessitating a careful analysis of the learning dynamics.

## 3. Preliminaries

In this section, we introduce the necessary definitions and formally describe private learning under the multi-patch data distribution and the convolutional neural network (CNN). Our analysis focuses on binary classification, and the data distribution is defined as follows.

**Definition 3.1** (Data Distribution). Let $\mathbf{v} \in \mathbb{R}^d$ be a fixed vector representing the feature signal contained in each data point. Each data point $(\mathbf{x}, y)$ with input $\mathbf{x} = [\mathbf{x}_1, \mathbf{x}_2] \in (\mathbb{R}^d)^2$ and label $y \in \{+1, -1\}$ is generated from the following distribution:

(1) The label $y$ is generated as a Rademacher random variable;

(2) The input $\mathbf{x}$ is generated as a vector of 2 patches, i.e., $\mathbf{x} = [\mathbf{x}_1, \mathbf{x}_2] \in (\mathbb{R}^d)^2$. The first patch is given by $\mathbf{x}_1 = y \cdot \mathbf{v}$ and the second patch is given by $\mathbf{x}_2 = \boldsymbol{\xi}$, where $\boldsymbol{\xi} \sim \mathcal{N}(0, \sigma_\xi^2 \cdot \mathbf{H})$ and is independent of the label $y$, where $\mathbf{H} = (\mathbf{I} - \mathbf{v}\mathbf{v}^\top \cdot \|\mathbf{v}\|_2^{-2})$.

Note that here $\mathbf{H}$ is designed to ensure that $\mathbf{v}$ is orthogonal to $\boldsymbol{\xi}$, i.e., the data noise is unrelated to the feature. Our data generation model is inspired by the structure of image data, which has been widely utilized in the feature learning area (Allen-Zhu & Li, 2020; Cao et al., 2022; Jelassi & Li, 2022; Kou et al., 2023; Zou et al., 2023; Ding et al., 2026; Zhang et al., 2026a;b). Notably, we introduce a term, data noise $\boldsymbol{\xi}$, into the data distribution, which is often overlooked in analyses within the differential privacy community. However, this seemingly 'negligible' component significantly influences the model's generalization ability, as underscored by the signal-to-noise ratio.

**Learner Model.** We consider a two-layer convolutional neural network (CNN) that processes input data by applying convolutional filters to two patches, $\mathbf{x}_1$ and $\mathbf{x}_2$, separately. The second-layer parameters of this network are fixed as $+1/m$ and $-1/m$, respectively, leading to the following network representation:

$$f(\mathbf{W}, \mathbf{x}) = F_{+1}(\mathbf{W}_{+1}, \mathbf{x}) - F_{-1}(\mathbf{W}_{-1}, \mathbf{x}),$$

where $F_{+1}(\mathbf{W}_{+1}, \mathbf{x})$ and $F_{-1}(\mathbf{W}_{-1}, \mathbf{x})$ are defined as:

$$F_j(\mathbf{W}_j, \mathbf{x}) = \frac{1}{m} \sum_{r=1}^m [\sigma(\langle \mathbf{w}_{j,r}, \mathbf{x}_1 \rangle) + \sigma(\langle \mathbf{w}_{j,r}, \mathbf{x}_2 \rangle)], \quad (1)$$

for $j \in \{\pm 1\}$, where $m$ denotes the number of convolutional filters in each of $F_{+1}$ and $F_{-1}$. Here, $\sigma(z) = (\max\{0, z\})^q$ represents the polynomial ReLU activation function with $q > 2$, $\mathbf{W}_j$ denotes the set of model weights associated with $F_j$, corresponding to the positive or negative filters. Each weight vector $\mathbf{w}_{j,r} \in \mathbb{R}^d$ is the parameters of the $r$-th neuron/filter in $\mathbf{W}_j$. We use $\mathbf{W}$ to represent the complete set of model weights across all filters.

**Differential Private Learning.** Given a training dataset $D = \{(\mathbf{x}_i, y_i)\}_{i=1}^n$, sampled from a joint distribution $\mathcal{D}$ over $\mathbf{x} \times y$, the goal is to train the learner model by minimizing the following empirical risk, measured by logistic loss, while simultaneously preserving privacy:

$$L_D(\mathbf{W}) = \frac{1}{n} \sum_{i=1}^n \ell[y_i \cdot f(\mathbf{W}, \mathbf{x}_i)], \quad (2)$$

where $\ell(z) = \log(1 + \exp(-z))$. More formally, the trained private model $\mathbf{W}$ should satisfy the mathematical definition of differential privacy as follows:

**Definition 3.2** ((Dwork et al., 2006)). Two datasets $D, D'$ are neighbors if they differ by only one element, which is denoted as $D \sim D'$. A randomized algorithm $\mathcal{A}$ is $(\varepsilon, \delta)$-differentially private (DP) if for all adjacent datasets $D, D'$ and for all events $S$ in the output space of $\mathcal{A}$, we have $\mathbb{P}(\mathcal{A}(D) \in S) \leq e^\varepsilon \cdot \mathbb{P}(\mathcal{A}(D') \in S) + \delta$.

**Definition 3.3** (Gaussian Mechanism (Dwork et al., 2010)). For a function $f : \mathcal{X}^n \mapsto \mathbb{R}^d$ with $L_2$-sensitivity $\Delta_2(f) = \max_{D,D'} \|f(D) - f(D')\|_2$, where $D$ and $D'$ are neighboring datasets, the Gaussian Mechanism outputs $f(D) + \mathbf{z}$. Here, $\mathbf{z} \sim \mathcal{N}(0, \sigma_z^2 \mathbb{I}_d)$ is Gaussian noise with scale $\sigma_z \geq \frac{\Delta_2(f)\sqrt{2\ln(1.25/\delta)}}{\varepsilon}$. This mechanism satisfies $(\varepsilon, \delta)$-differential privacy.

**Noisy Gradient Descent.** Noisy Gradient Descent (NoisyGD) and its stochastic counterpart, Noisy Stochastic Gradient Descent (Song et al., 2013; Abadi et al., 2016), are fundamental algorithms in differentially private deep learning. In this paper, we apply the NoisyGD algorithm to optimize Equation (2) and to update the filters in the CNN with the Gaussian mechanism.[2] Specifically,

$$
\begin{aligned}
\mathbf{w}_{j,r}^{(t+1)} &= \mathbf{w}_{j,r}^{(t)} - \eta \cdot (\nabla_{\mathbf{w}_{j,r}} L_D(\mathbf{W}^{(t)}) + \mathbf{z}_t) \\
&= \mathbf{w}_{j,r}^{(t)} - \frac{\eta}{nm} \sum_{i=1}^n \ell_i'^{(t)} \cdot \sigma'(\langle \mathbf{w}_{j,r}^{(t)}, \boldsymbol{\xi}_i \rangle) \cdot j y_i \boldsymbol{\xi}_i \\
&\quad - \frac{\eta}{nm} \sum_{i=1}^n \ell_i'^{(t)} \cdot \sigma'(\langle \mathbf{w}_{j,r}^{(t)}, y_i \mathbf{v} \rangle) \cdot j \mathbf{v} - \eta \mathbf{z}_t. \quad (3)
\end{aligned}
$$

where $\mathbf{z}_t$ is the private noise sampled from $\mathcal{N}(0, \sigma_z^2 \mathbb{I}_d)$, $\ell_i'^{(t)}$ is a shorthand notation of $\ell'[y_i \cdot f(\mathbf{W}^{(t)}, \mathbf{x}_i)]$. We assume that the noisy gradient descent algorithm starts from a Gaussian initialization, where each element of $\mathbf{W}_{+1}$ and $\mathbf{W}_{-1}$ is drawn from a Gaussian distribution $N(0, \sigma_0^2)$ with $\sigma_0^2$ representing the variance.

# 4. Main Results

In this section, we present our main theoretical results, demonstrating how the signal-noise-decomposition (Cao et al., 2022; Jelassi & Li, 2022; Kou et al., 2023; Zou et al., 2023) behaves during private learning using noisy gradient descent. It is clear that Equation (3) can be represented as a linear combination of random initialization, the signal feature, the data noise, and the accumulation of private noise, which can be formulated as the following definition.

**Definition 4.1.** Let $\mathbf{w}_{j,r}^{(t)}$ for $j \in \{\pm 1\}, r \in [m]$ be the convolution filters of the CNN at the $t$-th iteration of noisy gradient descent. Then there exist unique coefficients $\Gamma_{j,r}^{(t)} \geq$

---

[2]Note that, similar to previous studies on the theory of DP-SGD, we assume there is no clipping on gradients.

0 and $\Phi_{j,r,i}^{(t)}$ such that

$$\mathbf{w}_{j,r}^{(t)} = \mathbf{w}_{j,r}^{(0)} + j \cdot \Gamma_{j,r}^{(t)} \cdot \frac{\mathbf{v}}{\|\mathbf{v}\|_2^2} + \sum_{i=1}^{n} \Phi_{j,r,i}^{(t)} \cdot \frac{\boldsymbol{\xi}_i}{\|\boldsymbol{\xi}_i\|_2^2} - \eta \sum_{s=1}^{t} \mathbf{z}_s.$$

We further denote $\bar{\Phi}_{j,r,i}^{(t)} := \Phi_{j,r,i}^{(t)} \mathbb{1}(\Phi_{j,r,i}^{(t)} \geq 0), \underline{\Phi}_{j,r,i}^{(t)} := \Phi_{j,r,i}^{(t)} \mathbb{1}(\underline{\Phi}_{j,r,i}^{(t)} \leq 0)$. Then, we have that

$$\mathbf{w}_{j,r}^{(t)} = \mathbf{w}_{j,r}^{(0)} + j \cdot \Gamma_{j,r}^{(t)} \cdot \|\mathbf{v}\|_2^{-2} \cdot \mathbf{v} + \sum_{i=1}^{n} \bar{\Phi}_{j,r,i}^{(t)} \cdot \|\boldsymbol{\xi}_i\|_2^{-2} \cdot \boldsymbol{\xi}_i$$
$$+ \sum_{i=1}^{n} \underline{\Phi}_{j,r,i}^{(t)} \cdot \|\boldsymbol{\xi}_i\|_2^{-2} \cdot \boldsymbol{\xi}_i - \eta \sum_{s=1}^{t} \mathbf{z}_s. \quad (4)$$

In the decomposition of Equation (4), $\Gamma_{j,r}^{(t)}$ represents the extent to which the model learns the feature signal from data, whereas $\bar{\Phi}_{j,r,i}^{(t)}$ quantifies the degree of data noise memorization by the model. Both components are influenced by the interplay of private noise, as shown in the following lemma.

**Lemma 4.2.** *The coefficients* $\Gamma_{j,r}^{(t)}, \bar{\Phi}_{j,r,i}^{(t)}, \underline{\Phi}_{j,r,i}^{(t)}$ *in Definition 4.1 satisfy the following equations:*

$$\Gamma_{j,r}^{(0)}, \bar{\Phi}_{j,r,i}^{(0)}, \underline{\Phi}_{j,r,i}^{(0)} = 0$$

$$\Gamma_{j,r}^{(t+1)} = \Gamma_{j,r}^{(t)} - \frac{\eta}{nm} \cdot \sum_{i=1}^{n} \ell_i'^{(t)} \cdot \sigma'(\langle \mathbf{w}_{j,r}^{(t)}, y_i \cdot \mathbf{v} \rangle) \cdot \|\mathbf{v}\|_2^2$$

$$\bar{\Phi}_{j,r,i}^{(t+1)} = \bar{\Phi}_{j,r,i}^{(t)} - \frac{\eta}{nm} \cdot \ell_i'^{(t)} \cdot \sigma'(\langle \mathbf{w}_{j,r}^{(t)}, \boldsymbol{\xi}_i \rangle) \cdot \|\boldsymbol{\xi}_i\|_2^2 \cdot \mathbb{1}(y_i = j),$$

$$\underline{\Phi}_{j,r,i}^{(t+1)} = \underline{\Phi}_{j,r,i}^{(t)} + \frac{\eta}{nm} \cdot \ell_i'^{(t)} \cdot \sigma'(\langle \mathbf{w}_{j,r}^{(t)}, \boldsymbol{\xi}_i \rangle) \cdot \|\boldsymbol{\xi}_i\|_2^2 \cdot \mathbb{1}(y_i = -j)$$

Lemma 4.2 reveals that the process of private learning can be explored by the iterative dynamics of $\Gamma_{j,r}^{(t)}$, $\bar{\Phi}_{j,r,i}^{(t)}$ and $\underline{\Phi}_{j,r,i}^{(t)}$. It is noticed that private noise only influences the interior of $\sigma'(\cdot)$. Since $\ell_i'^{(t)} < 0$, Lemma 4.2 provides favorable properties for the dynamics of $\Gamma_{j,r}^{(t)}$, $\bar{\Phi}_{j,r,i}^{(t)}$: $\Gamma_{j,r}^{(t)}$ and $\bar{\Phi}_{j,r,i}^{(t)}$ increase monotonically, while $\underline{\Phi}j,r,i^{(t)}$ decreases monotonically. Then, we could demonstrate that these coefficients remain bounded throughout the private training process.

**Proposition 4.3.** *Under certain conditions, let* $T^* = \eta^{-1} \mathrm{poly}(\kappa^{-1}, \|\mathbf{v}\|_2^{-1}, d^{-1}\sigma_\xi^{-2}, \sigma_0^{-1}, n, m, d)$ *and* $T_p^* = \min\{T^*, \eta^{-1}Cmn\varepsilon\sigma_0\mu^{-1}(\|\mathbf{v}\|_2 + \|\boldsymbol{\xi}\|_2)^{-1}\}$*, where* $C = 4\log(T^*) = \widetilde{O}(1)$ *and* $\mu = \max\{1, \|\mathbf{v}\|_2, \|\boldsymbol{\xi}\|_2\}$*. Then, with at least probability* $1 - 1/d$*, it holds that, for* $t \leq T_p^*$ *:*

- $0 \leq \Gamma_{j,r}^{(t)}, \bar{\Phi}_{j,r,i}^{(t)} \leq 4\log(T_p^*)$ *for all* $j \in \{\pm 1\}, r \in [m]$ *and* $i \in [n]$.

- $0 \geq \underline{\Phi}_{j,r,i}^{(t)} \geq -2\max_{i,j,r}\{|\langle \mathbf{w}_{j,r}^{(0)}, \mathbf{v} \rangle|, |\langle \mathbf{w}_{j,r}^{(0)}, \boldsymbol{\xi}_i \rangle|\} - 16n\sqrt{\frac{\log(4n^2/\delta)}{d}} - 0.2 \geq -4\log(T_p^*)$ *for all* $j \in \{\pm 1\}$, $r \in [m]$ *and* $i \in [n]$.

Compared to standard training (Cao et al., 2022; Kou et al., 2023), private learning also ensures bounded coefficients but further restricts the number of training iterations due to the cumulative effect of private noise.

**Lemma 4.4.** *For any iteration* $t$*, with at least probability* $1 - 1/d$*, it holds that*

$$|\eta \sum_{s=1}^{t} \langle \mathbf{z}_s, \mathbf{v} \rangle| \leq \frac{\eta C\sqrt{tT}\|\mathbf{v}\|_2^2 \log(d)}{mn\varepsilon}(1 + \frac{1}{\mathrm{SNR}}), \quad (5)$$

$$|\eta \sum_{s=1}^{t} \langle \mathbf{z}_s, \boldsymbol{\xi}_i \rangle| \leq \frac{\eta C\sqrt{tT}\|\boldsymbol{\xi}\|_2^2 \log(d)}{mn\varepsilon}(1 + \mathrm{SNR}). \quad (6)$$

Lemma 4.4 characterizes the influence of private noise $(\mathbf{z}_t)$ on the training process by decomposing its interaction with the feature signal $(\mathbf{v})$ and data noise $(\boldsymbol{\xi})$. According to Equation (3), let $E$ denote the *good event* given by the intersection of the events in Lemma 4.4 and Proposition F.3 (Appendix), on which all internal coefficients remain bounded. On this event $E$, the per-step gradient has $\ell_2$-sensitivity at most $\frac{C(\|\mathbf{v}\|_2 + \|\boldsymbol{\xi}\|_2)}{nm}$, where $C = \widetilde{O}(1)$ follows from the existing work (Cao et al., 2022). Under Definition 3.1, we show that $\Pr_{\text{data} \sim \text{Definition 3.1}}(E) \geq 1 - \beta$ for a small $\beta$ (e.g., polynomially small in $1/d$ ). Conditioned on $E$, the Gaussian noise added by NoisyGD is calibrated exactly as in the Gaussian mechanism, so the algorithm is $(\varepsilon, \delta)$-DP according to our DP definition. Thus, under the generative model of Definition 3.1, we obtain the following *distributional* DP guarantee: $\Pr_{\text{data} \sim \text{Definition 3.1}}[\text{NoisyGD is } (\varepsilon, \delta) - \mathrm{DP}] \geq 1 - \beta$.

Moreover, $T$ denotes the total training iterations, which differs from $T_p^*$, the maximum permissible iterations. Moreover, recall the definition of $\mathrm{SNR} := \|\mathbf{v}\|_2/\|\boldsymbol{\xi}\|_2$, then the Equation (6) can be further represented as $|\eta \sum_{s=1}^{t} \langle \mathbf{z}_s, \boldsymbol{\xi}_i \rangle| \leq \frac{\eta C\sqrt{tT}\|\mathbf{v}\|_2^2 \log(1/\delta)}{mn\varepsilon}(\frac{1}{\mathrm{SNR}^2} + \frac{1}{\mathrm{SNR}})$. It indicates that the cumulative influences of private noise on both the feature signal and data noise are affected by the SNR.

Our analysis considers an over-parameterized model, which guarantees sufficient capacity to capture meaningful feature signals. However, this expressive power also raises a critical question: given the over-parameterization, the model may also learn substantial data noise. Thus, under what conditions does the model prioritize learning feature signals over data noise? To answer this, we examine two scenarios characterized by the signal-to-noise ratio, highlighting how it governs the trade-off between effective signal learning and data noise memorization.

### 4.1. Feature Signal Learning

Next, we first introduce conditions that guarantee the model's ability to learn feature signals.

**Condition 4.5** (Conditions of Signal Learning). *Suppose that:*

- *Dimension $d$ is sufficiently large, specifically $d = \widetilde{\Omega}(m^{2 \vee [4/(q-2)]} n^{4 \vee [(2q-2)/(q-2)]})$.*

- *Training sample size $n$ and neural network width $m$ satisfy $n, m = \Omega(\text{poly} \log(d))$.*

- *The learning rate $\eta \leq \widetilde{O}(\min\{\|\mathbf{v}\|_2^{-2}, \|\boldsymbol{\xi}\|_2^{-2}\})$.*

- *The standard deviation of Gaussian initialization $\sigma_0$ is appropriately chosen such that $\widetilde{O}((n\varepsilon)^{-\frac{1}{q}} \|\mathbf{v}\|_2^{-1}) \leq \sigma_0 \leq \widetilde{O}(\min\{\varepsilon^{-\frac{1}{q}} \|\boldsymbol{\xi}\|_2^{-1}, (n)^{-\frac{1}{2q}} \|\mathbf{v}\|_2^{-1}, (n\varepsilon)^{\frac{-q-1}{q}} \|\boldsymbol{\xi}\|_2^{-1})\})$.*

The conditions on $d, m, n$ are set to ensure that the learning problem is in a sufficiently over-parameterized setting, similar to the assumptions adopted in (Cao et al., 2022; Frei et al., 2022; Chatterji & Long, 2023; Kou et al., 2023). Additionally, the conditions on initialization $\sigma_0$ and step size $\eta$ are to guarantee that gradient descent can effectively minimize the training loss. In private deep learning, the privacy budget is typically moderately larger compared to private optimization theory (Abadi et al., 2016; Tramer & Boneh, 2020; De et al., 2022; Sun et al., 2023; Bao et al., 2023; Tang et al., 2024b;a). Therefore, we assume that the privacy budget remains larger than $1/\sqrt{n}$ here.

**Theorem 4.6.** *Under the same conditions as signal learning, if $\min\{\text{SNR} \cdot n\varepsilon, \text{SNR}^q \cdot n\} \geq \widetilde{\Omega}(1)$, with at least probability $1 - 1/d$, there exists $T_1 = O(\frac{\log(1/\sigma_0 \|\mathbf{v}\|_2) 4^{q-1} m}{\eta q \sigma_0^{q-2} \|\mathbf{v}\|_2^q})$ such that*

- *$\max_r \Gamma_{j,r}^{(T_1)} \geq 2$ for $j \in \{\pm 1\}$.*

- *$|\Phi_{j,r,i}^{(t)}| = O(\sigma_0 \sigma_\xi \sqrt{d})$ for all $j \in \{\pm 1\}, r \in [m], i \in [n]$ and $0 \leq t \leq T_1$.*

Here, we present one of our formal results. Based on Prop. 4.3 and Theorem 4.6, at the end of the training stage $T_1$, when $\min\{\text{SNR} \cdot n\varepsilon, \text{SNR}^q \cdot n\} \geq \widetilde{\Omega}(1)$, the maximum signal learning, $\max_r \Gamma_{j,r}^{(T_1)}$, achieves $\widetilde{\Theta}(1)$. Additionally, as the initialization scale $\sigma_0$ satisfies $\sigma_0 \leq \widetilde{O}((n\varepsilon)^{-1-1/q} \|\boldsymbol{\xi}\|_2^{-1})$ in Condition 4.5, this indicates the memorization of data noise, $|\Phi_{j,r,i}^{(t)}|$ remains bounded by $\widetilde{O}((n\varepsilon)^{-1-1/q})$ and it is smaller than the feature signal when $\varepsilon \geq 1/\sqrt{n}$. Moreover, compared to non-private learning, the condition $\text{SNR} \cdot n\varepsilon \geq \widetilde{\Omega}(1)$ introduces more challenges. Even when the feature learning conditions ($\text{SNR}^q \cdot n \geq \widetilde{\Omega}(1)$) for standard non-private learning are satisfied, private learning may still fail to capture the feature signal if $\text{SNR} \cdot n\varepsilon \leq \widetilde{\Omega}(1) \leq \text{SNR}^q \cdot n$. It demonstrates

stronger feature signals are required in private learning compared to non-private learning, which aligns the empirical principles in previous work (Tramer & Boneh, 2020; Sun et al., 2023; Bao et al., 2023; Tang et al., 2024b).

Moreover, we can further show that if some stronger assumptions are satisfied, the following corollary ensures that private learning can achieve training loss comparable to those of non-private learning.

**Corollary 4.7.** *Let $T, T_1$ be defined as above. Then, under the same conditions as signal learning, for any $t \in [T_1, T]$, it holds that $|\Phi_{j,r,i}^{(t)}| \leq \sigma_0 \sigma_\xi \sqrt{d}$ for all $j \in \{\pm 1\}, r \in [m]$ and $i \in [n]$ if $(n\varepsilon)^{q+1} \geq m$. Moreover, let $\mathbf{W}^*$ denote the collection of CNN parameters with convolution filters defined as $\mathbf{w}_{j,r}^* = \mathbf{w}_{j,r}^{(0)} + 2qm \log(2q/\kappa) \cdot j \cdot \|\mathbf{v}\|_2^{-2} \cdot \mathbf{v}$ for a constant $\kappa > 0$. Then, with at least probability $1 - 1/d$, the following bound holds*

$$\sum_{s=T_1}^{t} L_D(\mathbf{W}^{(s)}) \leq \underbrace{\frac{\|\mathbf{W}^{(T_1)} - \mathbf{W}^*\|_F^2}{(2q-1)\eta} + \frac{(t-T_1+1)\kappa}{(2q-1)}}_{\textit{Non-private terms}} \quad (7)$$

$$+ \underbrace{(t-T_1+1) \cdot \frac{\eta d \sigma_z^2 + \widetilde{O}(\sigma_z m^{3/2} \|\mathbf{v}\|_2^{-1})}{(2q-1)}}_{\textit{Private terms}}$$

*for some $t \in [T_1, T]$, where we denote $\|\mathbf{W}\|_F = \sqrt{\|\mathbf{W}_{+1}\|_F^2 + \|\mathbf{W}_{-1}\|_F^2}$.*

Corollary 4.7 characterizes the empirical risk of private learning under signal learning conditions, which can be decomposed into two terms. For standard non-private learning, by setting $T = T_1 + \lfloor \frac{\|\mathbf{W}^{(T_1)} - \mathbf{W}^*\|_F^2}{2\eta\kappa} \rfloor$ and dividing $t - T_1 + 1$ into both sides, the non-private term in the empirical loss can be bounded by $\frac{3\kappa}{2q-1}$, allowing the empirical loss to converge to $\kappa$. However, private learning introduces two key differences: 1) There are stricter limitations on the total training time, which may prevent $T$ from being as large as necessary for stable training. 2) In addition to the non-private term, the empirical risk involves a private term that appears unbounded, posing additional challenges to achieving convergence. Nonetheless, if we can assume the data satisfies: $T = T_p^* = O(\frac{mn\varepsilon\sigma_0}{\eta\mu(\|\mathbf{v}\|_2 + \|\boldsymbol{\xi}\|_2)}) = T^* \geq \kappa^{-1}$, where we denote $\mu = \max\{1, \|\mathbf{v}\|_2, \|\boldsymbol{\xi}\|_2\}$. Recalling the scale of private noise, we can obtain that $\sigma_z = \sigma_0/\eta\mu\sqrt{T}$. Then, it can be verified that the empirical risk in Equation (7) will be upper bounded by $O(\kappa)$, provided there exists a step size $\eta$ satisfying: $\eta \geq \max\left\{\frac{2d\sigma_0^2}{\mu^2 T\kappa}, \frac{2m^{2/3} \|\mathbf{v}\|_2^{-1} \sigma_0}{\mu\sqrt{T}\kappa}\right\}$.

By combining the above results, we derive the following corollary, which states that the private CNN can achieve a test loss related to privacy budget $\varepsilon$ under Condition 4.5.

**Corollary 4.8.** *Under the same conditions as above, suppose $\text{SNR} \cdot n\varepsilon \geq \widetilde{\Omega}(1)$ and $(n\varepsilon)^{1/q+1} \geq \widetilde{\Omega}(1)$. Then, with at least probability $1 - 1/d$ and with $L_D(\mathbf{W}^{(t)}) \leq O(\kappa)$*

*for some $t \leq T$, the test error satisfies $L_{\mathcal{D}}(\mathbf{W}^{(t)}) \leq O(\kappa + \exp((n\varepsilon)^{-1-1/q}))$.*

Here, the test error is defined as $L_{\mathcal{D}}(\mathbf{W}) := \mathbb{P}_{(\mathbf{x},y)\sim\mathcal{D}}[y \cdot (f(\mathbf{W}, \mathbf{x})) < 0]$.

### 4.2. Data Noise Memorization

Next, we explore the scenario where the model primarily learns label-independent noise rather than the feature signal.

**Condition 4.9** (Conditions of Data Noise Memorization). *Suppose that the initial three conditions of data noise memorization are the same as Condition 4.5. Additionally, we have*

- *The standard deviation of $\sigma_0$ is appropriately chosen such that $\widetilde{O}(\max\{\varepsilon^{-1/q}\|\boldsymbol{\xi}\|_2^{-1}, (n/\sqrt{d})\|\mathbf{v}\|_2^{-1}\}) \leq \sigma_0 \leq \widetilde{O}(\min\{(n\varepsilon)^{-1/q}\|\mathbf{v}\|_2^{-1}, \|\boldsymbol{\xi}\|_2^{-1}\})$.*

Similar to feature signal learning, we also establish conditions for data noise memorization, but with a difference in the initialization setup and requirement on $\varepsilon \geq 1$.

**Theorem 4.10.** *Under the same conditions as data noise memorization, if $\min\{\mathrm{SNR}^{-1}\cdot\varepsilon, \mathrm{SNR}^{-q}\cdot n^{-1}\} \geq \widetilde{\Omega}(1)$, with at least probability $1 - 1/d$, there exists $T_1 = O(\frac{\log(1/(\sigma_0\sigma_\xi\sqrt{d}))mn}{0.15^{q-2}\eta q\sigma_0^{q-2}(\sigma_\xi^2\sqrt{d})^q})$ such that*

- $\max_{j,r} \bar{\Phi}_{j,r,i}^{(T_1)} \geq 2$ *for all $i \in [n]$.*

- $\max_{j,r} \Gamma_{j,r}^{(t)} = \widetilde{O}(\sigma_0\|\mathbf{v}\|_2)$ *for all $0 \leq t \leq T_1$.*

- $\max_{j,r,i} |\underline{\Phi}_{j,r,i}^{(t)}| = \widetilde{O}(\sigma_0\sigma_\xi\sqrt{d})$ *for all $0 \leq t \leq T_1$.*

Theorem 4.10 shows that the data noise memorization, $\max_{j,r} \bar{\Phi}_{j,r,i}^{(T_1)}$, exceeds the feature signal learning at the end of the first training stage $T_1$, provided that $\min\{\mathrm{SNR}^{-1}\cdot\varepsilon, \mathrm{SNR}^{-q}\cdot n^{-1}\} \geq \widetilde{\Omega}(1)$. In contrast to Theorem 4.6, here demonstrates that when $\mathrm{SNR}^{-1}\cdot\varepsilon \leq \widetilde{\Omega}(1) \leq \mathrm{SNR}^{-q}\cdot n^{-1}$, the memorization of data noise does not occur in private learning, even though it may happen in standard non-private learning. However, it is important to note that this scenario only arises under the highly restrictive condition $\varepsilon \leq n^{-q}$, which is an overly stringent condition and unlikely to be met in practical private deep learning scenarios. In other words, when data noise memorization occurs in standard non-private learning, it will also occur in private learning as long as $\varepsilon \geq \mathrm{SNR}^{1-q}\,n^{-1}$. Moreover, it is noticed that $\max_{j,r,i} |\underline{\Phi}_{j,r,i}^{(t)}|$ is bounded by $\widetilde{O}(\sigma_0\sigma_\xi\sqrt{d})$, while under stricter privacy budget (smaller $\varepsilon$), this term is much larger than the non-private learning, indicating that private learning may amplifying the data noise memorization of the other filter.

Since we assume the model is over-parameterized, even if it fails to learn a good feature signal, it can still fit the data noise well enough for the training loss to converge to a small value, similar to the case of feature signal learning.

**Corollary 4.11.** *Let $T, T_1$ be defined as above. Then under the same conditions as data noise memorization, for any $t \in [T_1, T]$, it holds that $|\Gamma_{j,r,i}^{(t)}| \leq \sigma_0\|\mathbf{v}\|_2$ for all $j \in \{\pm 1\}$ and $r \in [m]$ if $n^{1/q}\varepsilon \geq m$. Moreover, let $\mathbf{W}^*$ be the collection of CNN parameters with convolution filters $\mathbf{w}_{j,r}^* = \mathbf{w}_{j,r}^{(0)} + 2qm\log(2q/\kappa))[\sum_{i=1}^n \mathbb{1}(j = y_i) \cdot \frac{\boldsymbol{\xi}_i}{\|\boldsymbol{\xi}_i\|_2}]$. Then, with at least probability $1 - 1/d$, the following bound holds*

$$\sum_{s=T_1}^t L_D(\mathbf{W}^{(s)}) \leq \underbrace{\frac{\|\mathbf{W}^{(T_1)} - \mathbf{W}^*\|_F^2}{(2q-1)\eta} + \frac{(t - T_1 + 1)\kappa}{(2q-1)}}_{\text{Non-private terms}} \quad (8)$$
$$+ \underbrace{(t - T_1 + 1) \cdot \frac{\eta d\sigma_z^2 + \widetilde{O}(\sigma_z m^2 n^{1/2}\|\boldsymbol{\xi}\|_2^{-1})}{(2q-1)}}_{\text{Private terms}}$$

*for some $t \in [T_1, T]$, where we denote $\|\mathbf{W}\|_F = \sqrt{\|\mathbf{W}_{+1}\|_F^2 + \|\mathbf{W}_{-1}\|_F^2}$.*

Corollary 4.11 shares the same empirical loss structure as Corollary 4.7, with the only difference being the bound on $\|\mathbf{W}^{(T_1)} - \mathbf{W}^*\|_F$. This results in the term $\widetilde{O}(\sigma_z m^2 n^{1/2}\|\boldsymbol{\xi}\|_2^{-1})$ appearing here. Therefore, under the same data assumptions as Corollary 4.7, the empirical risk in Equation (8) can also be upper bounded by $O(\kappa)$, provided there exists a step size $\eta$ satisfying: $\eta \geq \max\left\{\frac{2d\sigma_0^2}{\mu^2 T\kappa}, \frac{2m^2 n^{1/2}\|\boldsymbol{\xi}\|_2^{-1}\sigma_0}{\mu\sqrt{T}\kappa}\right\}$. However, even if the private CNN model achieves a sufficiently small training loss under the data noise memorization scenario, it still fails to exhibit good generalization ability, as it primarily learns the label-independent data noise rather than the label-dependent feature signals.

**Corollary 4.12.** *Under the same conditions as data noise memorization, within $T$ iterations, regardless of how the sample size $n$ and privacy budget $\varepsilon$ chosen, with at least probability $1 - 1/d$, we can find $\mathbf{W}^{(\widetilde{T})}$ such that $L_D(\mathbf{W}^{(\widetilde{T})}) \leq O(\kappa)$. Additionally, for some $0 \leq t \leq \widetilde{T}$ we have that $L_{\mathcal{D}}(\mathbf{W}^{(t)}) \geq 0.1$.*

## 5. Experiment

Due to space limitations, we present only a synthetic data experiment here. Experimental setup and additional results on the CIFAR-10 dataset (Krizhevsky, 2009), including an exploration of SNR, are provided in the Appendix. In the synthetic experiment, we compare the evolution of feature signal and data noise between NoisyGD and standard (non-private) training.

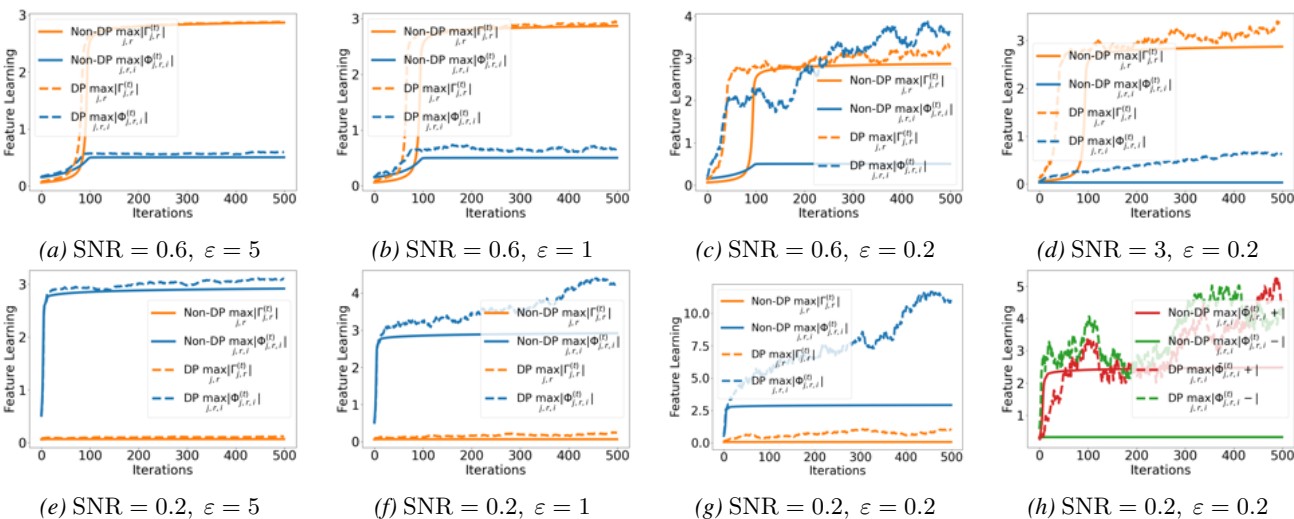

*Figure 2.* Comparison of Feature Signal and Data Noise in (Non) Private Learning. The figure compares the dynamics of feature learning $\max_{j,r}|\Gamma_{j,r}^{(t)}|$ and data noise $\max_{j,r,i}|\Phi_{j,r,i}^{(t)}|$ across varying privacy budgets $\varepsilon$ and SNR. Subfigures 2a–2c correspond to SNR = 0.6 with $\varepsilon \in \{5, 1, 0.2\}$, subfigure 2d corresponds to SNR = 3 with $\varepsilon$ = 0.2, and subfigures 2e–2g correspond to SNR = 0.2 with $\varepsilon \in \{5, 1, 0.2\}$. Subfigure 2h shows a decomposition of $\max_{j,r,i}|\Phi_{j,r,i}^{(t)}|$ at SNR = 0.2, $\varepsilon$ = 0.2.

**Experimental Setup.** We conducted experiments by generating synthetic data defined in Definition 3.1 with a controlled signal-to-noise ratio (SNR = 0.2, 0.6, 3). The dataset was constructed using a fixed signal vector $\mathbf{v}$ and a data noise component $\boldsymbol{\xi}$. We implemented a two-layer CNN with an input dimension of $d = 1000, m = 10$ filters, and a polynomial ReLU activation function with a power parameter $q = 3$. The model weights were initialized randomly from a normal distribution with a small variance ($\sigma_0 = 0.001$), and training was performed using gradient descent with a learning rate of $\eta = 0.01$ over 500 epochs. For private learning, we consider the noisy gradient descent during weight updates, with a privacy budget of $\epsilon = 0.2, 1, 5$ and $\delta = 10^{-5}$. Throughout the training process, we monitored the maximum inner products between the learned weights and the signal and data noise components, denoted as $\max_{j,r}|\Gamma_{j,r}^{(t)}|$ and $\max_{j,r,i}|\Phi_{j,r,i}^{(t)}|$, respectively.

**Private learning requires stronger feature signal.** In Figure 2, we compare the dynamics of feature signal learning and data noise memorization during the non-private and private training process under the higher SNR. In subfigures 2a and 2b, it can be observed that when the SNR = 0.6 is sufficient for non-private training and the privacy budgets $\varepsilon = 1, 5$ are moderately larger, the feature learning trajectories in private training closely align with those of non-private training. However, in subfigures 2c, when the SNR = 0.6 is sufficient for non-private training and $\varepsilon = 0.2$ is relatively smaller, the data noise memorization (represented by the blue dashed line) no longer remains below the feature signal like the non-private training. This indicates the SNR = 0.6 is insufficient for private training under stronger privacy

constraints. When we increase SNR to 3 in Figure 2d, we observe that private training regains its ability to perform feature learning, exhibiting trends similar to non-private training. This demonstrates that as the privacy budget $\varepsilon$ decreases (i.e., stronger privacy guarantees), the requirement for a higher SNR becomes more pronounced.

**Private learning may amplify data noise memorization.** In Figure 2e-Figure 2g, we observe that if non-private training exhibits data noise memorization (the blue line is higher than the orange line), private training is also prone to this behavior (corresponding to the dashed line). Moreover, as illustrated in Figure 2h, the green DP line (representing noise memorization under private learning) shows a noticeable increase, while the corresponding non-DP green line remains unchanged. This indicates that private training, particularly with smaller privacy budgets, not only fails to suppress data noise but also amplifies its memorization for other filters, aligning with our theoretical analysis in Theorem 4.10.

## 6. Conclusion

In this paper, we introduced the first theoretical framework to analyze the dynamics of feature learning in differentially private learning, focusing on the trade-offs between feature signals and data noise through a decomposition of these components. Using a two-layer CNN, we demonstrated that private learning necessitates a higher signal-to-noise ratio (SNR) compared to non-private training to effectively capture features, particularly under stringent privacy budgets. Additionally, we showed that data noise memorization, if present in non-private learning, persists in private learning,

resulting in poor generalization even when training losses are minimized. Our findings highlight the critical role of feature enhancement in private learning, aligning with prior empirical studies and providing valuable insights for designing effective privacy-preserving learning systems.

## Impact Statement

This paper presents work whose goal is to advance the field of Machine Learning. There are many potential societal consequences of our work, none which we feel must be specifically highlighted here.

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

## A. The Use of Large Language Models

We used LLMs as assistive tools for writing. For writing, LLMs were used to polish language (grammar, wording, and flow) and suggest alternative phrasings; the research problem setup, preliminaries, methods, analyses, and conclusions were conceived and written by the authors.

## B. Notations Table

*Table 1.* Notation Summary

| Symbol | Description |
|--------|-------------|
| $\mathbf{x}$ | Input data point with multi-patch structure $\mathbf{x} = [y\mathbf{v}, \boldsymbol{\xi}] \in (\mathbb{R}^d)^2$ |
| $y$ | Binary label ($\pm 1$) |
| $y \cdot \mathbf{v}$ | Label-dependent feature vector (signal component) |
| $\boldsymbol{\xi}$ | Label-independent Gaussian noise $\sim \mathcal{N}(0, \sigma_\xi^2 \mathbf{H})$ |
| $\mathbf{z}_t$ | Gaussian privacy noise added at iteration $t$ |
| SNR | Signal-to-noise ratio $\|\mathbf{v}\|_2 / \|\boldsymbol{\xi}\|_2$ |
| $T$ | Total number of training iterations |
| $T_p^*$ | Maximum number of private training iterations |
| $m$ | Number of convolutional filters per class |
| $d$ | Dimension of feature/noise vectors |
| $n$ | Number of training samples |
| $\eta$ | Learning rate in noisy gradient descent |
| $\sigma_0$ | Standard deviation of Gaussian weight initialization |
| $\sigma_\xi$ | Standard deviation of Gaussian data noise |
| $\sigma_z$ | Standard deviation of Gaussian private noise |
| $\varepsilon, \delta$ | $(\varepsilon, \delta)$-differential privacy parameters |
| $\Gamma_{j,r}^{(t)}$ | Signal learning coefficient for filter $r$ in class $j$ at iteration $t$ |
| $\Phi_{j,r,i}^{(t)}$ | Noise memorization coefficient for sample $i$ and filter $r$ in class $j$ |
| $\sigma(z)$ | Polynomial ReLU activation: $\max\{0, z\}^q$ with $q > 2$ |
| $L_D(\mathbf{W})$ | Empirical risk with logistic loss over dataset $D$ |
| $L_{\mathcal{D}}(\mathbf{W})$ | Population risk with logistic loss over data distribution $\mathcal{D}$ |
| $q$ | Polynomial degree in activation function ($q > 2$) |
| $\mathbf{H}$ | Orthogonal projection matrix $\mathbf{I} - \mathbf{v}\mathbf{v}^\top / \|\mathbf{v}\|_2^2$ |
| $\kappa$ | Convergence threshold for training loss |

## C. Additional Related Work

**Differentially Private Learning** The most widely used technique for differentially private training in deep learning is differentially private stochastic gradient descent (DP-SGD). However, the accuracy of private deep learning still significantly lags behind that of standard non-private learning across several benchmarks (McMahan et al., 2017; Papernot et al., 2021; Tramer & Boneh, 2020; De et al., 2022). To bridge this gap, various techniques have been proposed to enhance DP learning, including adaptive gradient clipping methods that dynamically adjust clipping thresholds (Andrew et al., 2021; Bu et al., 2024a), feature extraction or pre-processing before applying DP-SGD (Abadi et al., 2016; Tramer & Boneh, 2020; De et al., 2022; Sun et al., 2023; Bao et al., 2023; Tang et al., 2024b), parameter-efficient training strategies via adapters, low-rank weights, or quantization (Yu et al., 2021; Luo et al., 2021), and private noise reduction techniques using tree aggregation mechanisms or filters (Kairouz et al., 2021; Zhang et al., 2024).

## D. Additional Experimental Details

**Experimental Setup for Synthetic dataset.** We conducted experiments by generating synthetic data defined in Definition 3.1 with a controlled signal-to-noise ratio (SNR $= 0.2, 0.6, 3$). The dataset was constructed using a fixed signal vector $\mathbf{v}$ and a data noise component $\boldsymbol{\xi}$. We implemented a two-layer CNN with an input dimension of $d = 1000, m = 10$ filters, and a

polynomial ReLU activation function with a power parameter $q = 3$. The model weights were initialized randomly from a normal distribution with a small variance ($\sigma_0 = 0.001$), and training was performed using gradient descent with a learning rate of $\eta = 0.01$ over 500 epochs. For private learning, we consider the noisy gradient descent during weight updates, with a privacy budget of $\epsilon = 0.2, 1, 5$ and $\delta = 10^{-5}$. Throughout the training process, we monitored the maximum inner products between the learned weights and the signal and data noise components, denoted as $\max_{j,r} |\Gamma_{j,r}^{(t)}|$ and $\max_{j,r,i} |\Phi_{j,r,i}^{(t)}|$, respectively.

### D.1. Real-world Data Experiment

In this experiment, we explore the impact of SNR in the private learning. Due to the space limitation, we provide experimental setup and more results in Appendix D.1.

**Higher SNR Improves Accuracy Across Various Privacy Budgets.**

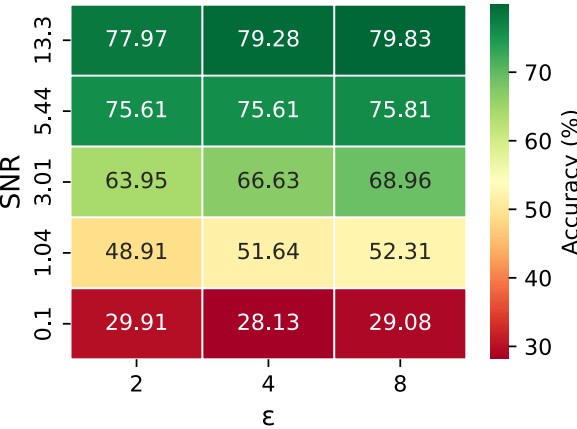

*Figure 3.* Impact of SNR and $\varepsilon$ on CIFAR-10 Accuracy.

The results, illustrated in Figure 3, reveal that higher SNR values consistently lead to improved model accuracy under various privacy budgets, as the cleaner signal allows the model to better learn useful feature signals. Moreover, as the privacy budget $\varepsilon$ decreases (indicating stronger privacy guarantees), the model's accuracy degrades, particularly under low SNR conditions. This degradation is attributed to the combined effects of data noise and the additional noise introduced by private learning.

**Experimental Setup for Appendix D.1.** We conducted experiments on the CIFAR-10 dataset (Krizhevsky, 2009), training on a version of the dataset corrupted with Gaussian noise applied at varying scales to control the signal-to-noise ratio (Hendrycks & Dietterich, 2019). The training data was corrupted with noise levels corresponding to different SNR values, while the test set remained clean. A ResNet-20 architecture (He et al., 2016) was employed as the baseline model, designed for CIFAR-10 with an input image size of $32 \times 32$ pixels. Training was performed using noisy gradient descent with an initial learning rate of $0.1$. The model was trained for 100 epochs with a batch size of 1000.

Figure 4, Figure 5 and Figure 6 depict the Class Activation Maps (CAMs) using GradCAM (Selvaraju et al., 2020) for different classes in the CIFAR-10 dataset under varying SNR. In CAMs, the colors represent the intensity of activation in specific regions of the input image. These activations indicate how strongly the model associates different areas of the image with a specific class. CAMs highlight the most important regions contributing to the model's prediction.

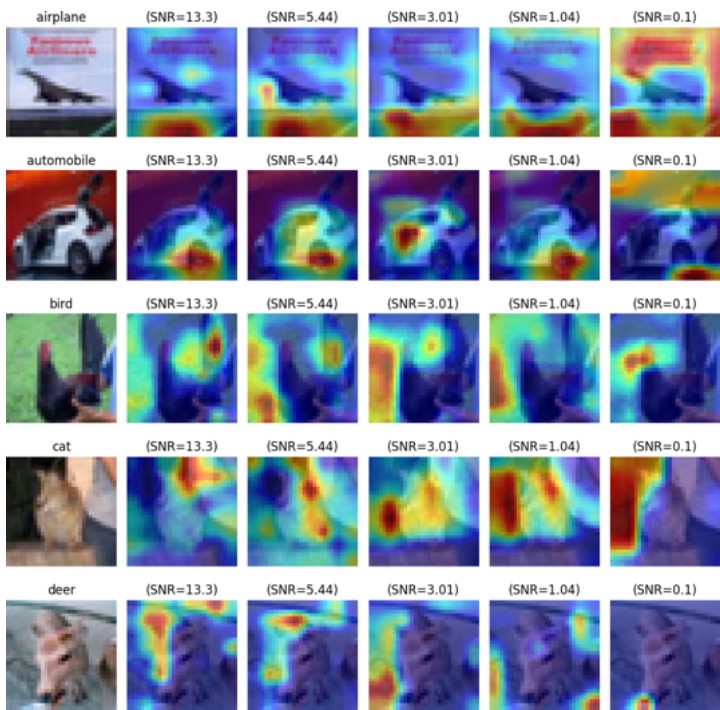

*Figure 4.* Class Activation Mappings for CIFAR-10 Across Different SNRs ($\epsilon = 8$).

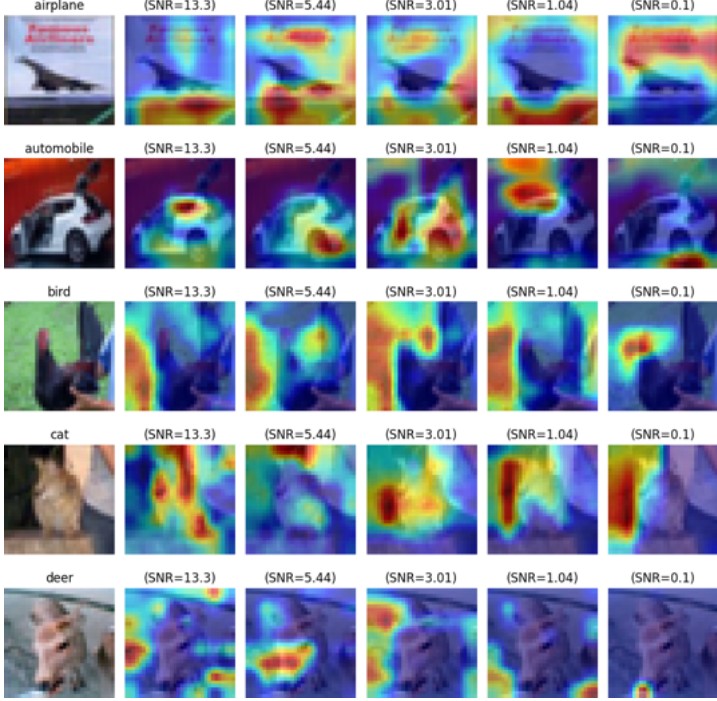

*Figure 5.* Class Activation Mappings for CIFAR-10 Across Different SNRs ($\epsilon = 4$).

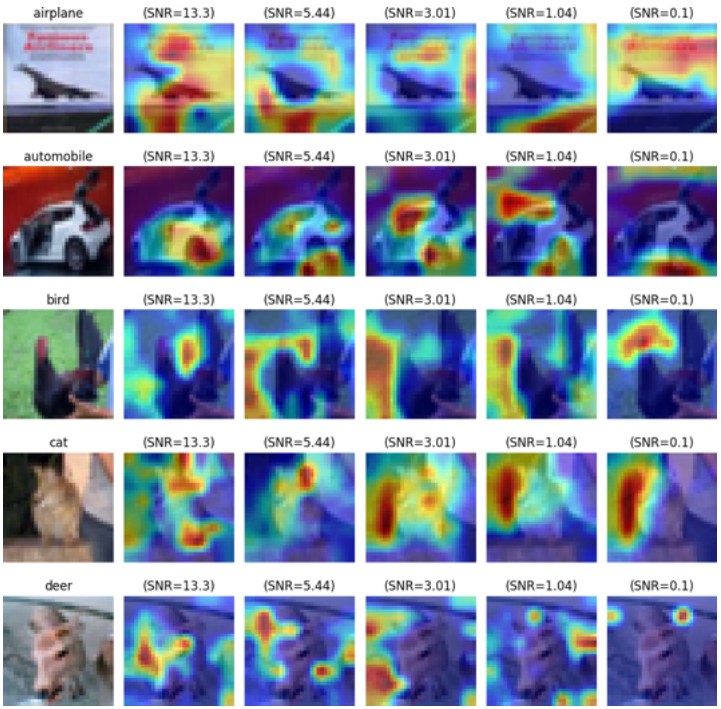

*Figure 6.* Class Activation Mappings for CIFAR-10 Across Different SNRs ($\epsilon = 2$).

## E. Support Lemmas

**Lemma E.1.** *Suppose that $\delta_x > 0$ and $n \geq 8\log(4/\delta_x)$. Then with probability at least $1 - \delta_x$,*

$$|\{i \in [n] : y_i = 1\}|, |\{i \in [n] : y_i = -1\}| \geq n/4.$$

**Proof of Lemma E.1.** We first establish the bound for $|\{i \in [n] : y_i = 1\}|$ and the bound for $|\{i \in [n] : y_i = -1\}|$ follows identically. Using Hoeffding's inequality, we know that with probability at least $1 - \delta/2$, the following holds:

$$\left|\frac{1}{n}\sum_{i=1}^{n} \mathbb{1}\{y_i = 1\} - \frac{1}{2}\right| \leq \sqrt{\frac{\log(4/\delta)}{2n}}.$$

Thus, for $n \geq 8\log(4/\delta)$, the size of the subset where $y_i = 1$ satisfies:

$$|\{i \in [n] : y_i = 1\}| = \sum_{i=1}^{n} \mathbb{1}\{y_i = 1\} \geq \frac{n}{2} - n \cdot \sqrt{\frac{\log(4/\delta)}{2n}} \geq \frac{n}{4}.$$

$\square$

**Lemma E.2.** *Suppose that $\delta_\xi > 0$ and $d = \Omega(\log(4n/\delta_\xi))$. Then, for all $i, i' \in [n]$, with probability at least $1 - \delta_\xi$,*

$$\sigma_\xi^2 d/2 \leq \|\boldsymbol{\xi}_i\|_2^2 \leq 3\sigma_\xi^2 d/2$$

$$|\langle \boldsymbol{\xi}_i, \boldsymbol{\xi}_{i'} \rangle| \leq 2\sigma_\xi^2 \cdot \sqrt{d\log(4n^2/\delta_\xi)}.$$

*Meanwhile, there is $\delta_z$ such that $\delta_z > 0$ and $d = \Omega(\log(4n/\delta_z))$. It holds, with probability at least $1 - \delta_z$,*

$$G^2\sigma_z^2 d/2 \leq \|\mathbf{z}_i\|_2^2 \leq 3G^2\sigma_z^2 d/2$$

$$|\langle \mathbf{z}_i, \mathbf{z}_{i'} \rangle| \leq 2G^2\sigma_z^2 \cdot \sqrt{d\log(4n^2/\delta_z)}.$$

*Proof of Lemma E.2.* Both $\boldsymbol{\xi}$ and $\mathbf{z}$ follow Gaussian distributions; therefore, it suffices to provide the proof for one case. Using Bernstein's inequality, we find that with probability at least $1 - \delta/(2n)$, the following holds:

$$\left|\|\boldsymbol{\xi}_i\|_2^2 - \sigma_\xi^2 d\right| = O(\sigma_\xi^2 \cdot \sqrt{d\log(4n/\delta)}).$$

Thus, when $d = \Omega(\log(4n/\delta))$, we have: $\frac{\sigma_\xi^2 d}{2} \leq \|\boldsymbol{\xi}_i\|_2^2 \leq \frac{3\sigma_\xi^2 d}{2}$. Next, note that $\langle \boldsymbol{\xi}_i, \boldsymbol{\xi}_{i'} \rangle$ has a mean of zero for any $i \neq i'$. Again, by Bernstein's inequality, with probability at least $1 - \delta/(2n^2)$, the following bound holds: $|\langle \boldsymbol{\xi}_i, \boldsymbol{\xi}_{i'} \rangle| \leq 2\sigma_\xi^2 \cdot \sqrt{d\log(4n^2/\delta)}$. Finally, applying a union bound over all $i$ and $i'$ completes the proof. $\qquad\square$

**Lemma E.3.** *Suppose that $d \geq \Omega(\log(mn/\delta)), m = \Omega(\log(1/\delta))$. Then with probability at least $1 - \delta$,*

$$|\langle \mathbf{w}_{j,r}^{(0)}, \mathbf{v} \rangle| \leq \sqrt{2\log(8m/\delta)} \cdot \sigma_0 \|\mathbf{v}\|_2$$
$$|\langle \mathbf{w}_{j,r}^{(0)}, \boldsymbol{\xi}_i \rangle| \leq 2\sqrt{\log(8mn/\delta)} \cdot \sigma_0 \sigma_\xi \sqrt{d}$$

*for all $r \in [m], j \in \{\pm 1\}$ and $i \in [n]$. Moreover,*

$$\sigma_0 \|\mathbf{v}\|_2/2 \leq \max_{r \in [m]} j \cdot \langle \mathbf{w}_{j,r}^{(0)}, \mathbf{v} \rangle \leq \sqrt{2\log(8m/\delta)} \cdot \sigma_0 \|\mathbf{v}\|_2,$$
$$\sigma_0 \sigma_\xi \sqrt{d}/4 \leq \max_{r \in [m]} j \cdot \langle \mathbf{w}_{j,r}^{(0)}, \boldsymbol{\xi}_i \rangle \leq 2\sqrt{\log(8mn/\delta)} \cdot \sigma_0 \sigma_\xi \sqrt{d}$$

*for all $j \in \{\pm 1\}$ and $i \in [n]$.*

**Proof of Lemma E.3.** For each $r \in [m]$, the term $j \cdot \langle \mathbf{w}_{j,r}^{(0)}, \mathbf{v} \rangle$ is a Gaussian random variable with mean zero and variance $\sigma_0^2 \|\mathbf{v}\|_2^2$. Applying the Gaussian tail bound and the union bound, we conclude that with probability at least $1 - \delta/4$,

$$j \cdot \langle \mathbf{w}_{j,r}^{(0)}, \mathbf{v} \rangle \leq |\langle \mathbf{w}_{j,r}^{(0)}, \mathbf{v} \rangle| \leq \sqrt{2\log(8m/\delta)} \cdot \sigma_0 \|\mathbf{v}\|_2.$$

Furthermore, the probability $\mathbb{P}(\sigma_0\|\mathbf{v}\|_2/2 > j \cdot \langle \mathbf{w}_{j,r}^{(0)}, \mathbf{v} \rangle)$ is an absolute constant. Thus, under the given condition on $m$, we have

$$\mathbb{P}(\sigma_0\|\mathbf{v}\|_2/2 \leq \max_{r \in [m]} j \cdot \langle \mathbf{w}_{j,r}^{(0)}, \mathbf{v} \rangle) = 1 - \mathbb{P}(\sigma_0\|\mathbf{v}\|_2/2 > \max_{r \in [m]} j \cdot \langle \mathbf{w}_{j,r}^{(0)}, \mathbf{v} \rangle)$$
$$= 1 - \mathbb{P}(\sigma_0\|\mathbf{v}\|_2/2 > j \cdot \langle \mathbf{w}_{j,r}^{(0)}, \mathbf{v} \rangle)^{2m}$$
$$\geq 1 - \delta/4.$$

By Lemma E.2, with probability at least $1 - \delta/4$, the inequality $\sigma_\xi \sqrt{d}/\sqrt{2} \leq \|\boldsymbol{\xi}_i\|_2 \leq \sqrt{3/2} \cdot \sigma_\xi \sqrt{d}$ holds for all $i \in [n]$. Consequently, the result for $\langle \mathbf{w}_{j,r}^{(0)}, \boldsymbol{\xi}_i \rangle$ can be derived using the same argument as for $j \cdot \langle \mathbf{w}_{j,r}^{(0)}, \mathbf{v} \rangle$. $\qquad\square$

## F. Decomposition

**Definition F.1** (Restatement of Definition 4.1)**.** Let $\mathbf{w}_{j,r}^{(t)}$ for $j \in \{\pm 1\}, r \in [m]$ be the convolution filters of the CNN at the $t$-th iteration of noisy gradient descent. Then there exist unique coefficients $\Gamma_{j,r}^{(t)} \geq 0$ and $\Phi_{j,r,i}^{(t)}$ such that

$$\mathbf{w}_{j,r}^{(t)} = \mathbf{w}_{j,r}^{(0)} + j \cdot \Gamma_{j,r}^{(t)} \cdot \|\mathbf{v}\|_2^{-2} \cdot \mathbf{v} + \sum_{i=1}^n \Phi_{j,r,i}^{(t)} \cdot \|\boldsymbol{\xi}_i\|_2^{-2} \cdot \boldsymbol{\xi}_i - \eta \sum_{s=1}^t \mathbf{z}_s \qquad (9)$$

We further denote $\bar{\Phi}_{j,r,i}^{(t)} := \Phi_{j,r,i}^{(t)} \mathbb{1}(\Phi_{j,r,i}^{(t)} \geq 0), \underline{\Phi}_{j,r,i}^{(t)} := \Phi_{j,r,i}^{(t)} \mathbb{1}(\Phi_{j,r,i}^{(t)} \leq 0)$. Then we have that

$$\mathbf{w}_{j,r}^{(t)} = \mathbf{w}_{j,r}^{(0)} + j \cdot \Gamma_{j,r}^{(t)} \cdot \|\mathbf{v}\|_2^{-2} \cdot \mathbf{v} + \sum_{i=1}^n \bar{\Phi}_{j,r,i}^{(t)} \cdot \|\boldsymbol{\xi}_i\|_2^{-2} \cdot \boldsymbol{\xi}_i + \sum_{i=1}^n \underline{\Phi}_{j,r,i}^{(t)} \cdot \|\boldsymbol{\xi}_i\|_2^{-2} \cdot \boldsymbol{\xi}_i - \eta \sum_{s=1}^t \mathbf{z}_s.$$

**Lemma F.2** (Restatement of Lemma 4.2)**.** *The coefficients* $\Gamma_{j,r}^{(t)}, \bar{\Phi}_{j,r,i}^{(t)}, \underline{\Phi}_{j,r,i}^{(t)}$ *in Definition 4.1 satisfy the following equations:*

$$\Gamma_{j,r}^{(0)}, \bar{\Phi}_{j,r,i}^{(0)}, \underline{\Phi}_{j,r,i}^{(0)} = 0$$

$$\Gamma_{j,r}^{(t+1)} = \Gamma_{j,r}^{(t)} - \frac{\eta}{nm} \cdot \sum_{i=1}^{n} \ell_i'^{(t)} \cdot \sigma'(\langle \mathbf{w}_{j,r}^{(t)}, y_i \cdot \mathbf{v} \rangle) \cdot \|\mathbf{v}\|_2^2$$

$$\bar{\Phi}_{j,r,i}^{(t+1)} = \bar{\Phi}_{j,r,i}^{(t)} - \frac{\eta}{nm} \cdot \ell_i'^{(t)} \cdot \sigma'(\langle \mathbf{w}_{j,r}^{(t)}, \boldsymbol{\xi}_i \rangle) \cdot \|\boldsymbol{\xi}_i\|_2^2 \cdot \mathbb{1}(y_i = j),$$

$$\underline{\Phi}_{j,r,i}^{(t+1)} = \underline{\Phi}_{j,r,i}^{(t)} + \frac{\eta}{nm} \cdot \ell_i'^{(t)} \cdot \sigma'(\langle \mathbf{w}_{j,r}^{(t)}, \boldsymbol{\xi}_i \rangle) \cdot \|\boldsymbol{\xi}_i\|_2^2 \cdot \mathbb{1}(y_i = -j)$$

**Proof of Lemma F.2.** We prove the statement by induction. For the base case $t = 0$, it holds that $\Gamma_{j,r}^{(0)} = 0$ and $\Phi_{j,r,i}^{(0)} = 0$. Now, assume the statement holds for $t = k$. We proceed to the inductive step, considering $t = k + 1$:

$$\mathbf{w}_{j,r}^{(k+1)} = \mathbf{w}_{j,r}^{(k)} - \frac{1}{nm} \sum_{i=1}^{n} \ell_i'^{(k)} \cdot \sigma'(\langle \mathbf{w}_{j,r}^{(k)}, \boldsymbol{\xi}_i \rangle) \cdot j y_i \boldsymbol{\xi}_i + \frac{\eta}{nm} \sum_{i=1}^{n} \ell_i'^{(k)} \cdot \sigma'(\langle \mathbf{w}_{j,r}^{(k)}, y_i \mathbf{v} \rangle) \cdot j\mathbf{v} - \eta \mathbf{z}_{k+1}$$

$$= \mathbf{w}_{j,r}^{(0)} + j \cdot \Gamma_{j,r}^{(k)} \cdot \|\mathbf{v}\|_2^{-2} \cdot \mathbf{v} + \sum_{i=1}^{n} \Phi_{j,r,i}^{(k)} \cdot \|\boldsymbol{\xi}_i\|_2^{-2} \cdot \boldsymbol{\xi}_i - \eta \sum_{s=1}^{k} \mathbf{z}_s$$

$$- \frac{1}{nm} \sum_{i=1}^{n} \ell_i'^{(k)} \cdot \sigma'(\langle \mathbf{w}_{j,r}^{(k)}, \boldsymbol{\xi}_i \rangle) \cdot j y_i \boldsymbol{\xi}_i - \frac{\eta}{nm} \sum_{i=1}^{n} \ell_i'^{(k)} \cdot \sigma'(\langle \mathbf{w}_{j,r}^{(k)}, y_i \mathbf{v} \rangle) \cdot j\mathbf{v} - \eta \mathbf{z}_{k+1}$$

$$= \mathbf{w}_{j,r}^{(0)} + j \cdot \|\mathbf{v}\|_2^{-2} \cdot \mathbf{v}(\Gamma_{j,r}^{(k)} - \frac{\eta}{nm} \cdot \sum_{i=1}^{n} \ell_i'^{(k)} \cdot \sigma'(\langle \mathbf{w}_{j,r}^{(k)}, y_i \cdot \mathbf{v} \rangle) \cdot \|\mathbf{v}\|_2^2)$$

$$+ \sum_{i=1}^{n} \|\boldsymbol{\xi}_i\|_2^{-2} \cdot \boldsymbol{\xi}_i \cdot (\Phi_{j,r,i}^{(k)} - \frac{\eta}{nm} \cdot \ell_i'^{(k)} \cdot \sigma'(\langle \mathbf{w}_{j,r}^{(k)}, \boldsymbol{\xi}_i \rangle) \cdot \|\boldsymbol{\xi}_i\|_2^2) - \sum_{s=1}^{k+1} \mathbf{z}_s$$

$$= \mathbf{w}_{j,r}^{(0)} + j \cdot \Gamma_{j,r}^{(k+1)} \cdot \|\mathbf{v}\|_2^{-2} \cdot \mathbf{v} + \sum_{i=1}^{n} \Phi_{j,r,i}^{(k+1)} \cdot \|\boldsymbol{\xi}_i\|_2^{-2} \cdot \boldsymbol{\xi}_i - \eta \sum_{s=1}^{k+1} \mathbf{z}_s.$$

The last equality follows directly from the data distribution and it is clear that the vectors involved are linearly independent. Thus, the decomposition is unique. Then, we have:

$$\Phi_{j,r,i}^{(t)} = -\sum_{s=0}^{t-1} \frac{\eta}{nm} \cdot \ell_i'^{(s)} \cdot \sigma'(\langle \mathbf{w}_{j,r}^{(s)}, \boldsymbol{\xi}_i \rangle) \cdot \|\boldsymbol{\xi}_i\|_2^2 \cdot y_i \cdot j.$$

Moreover, note that $\ell_i'^{(t)} < 0$ due to the definition of the cross-entropy loss. Consequently,

$$\bar{\Phi}_{j,r,i}^{(t)} = -\sum_{s=0}^{t-1} \frac{\eta}{nm} \cdot \ell_i'^{(s)} \cdot \sigma'(\langle \mathbf{w}_{j,r}^{(s)}, \boldsymbol{\xi}_i \rangle) \cdot \|\boldsymbol{\xi}_i\|_2^2 \cdot \mathbb{1}(y_i = j),$$

$$\underline{\Phi}_{j,r,i}^{(t)} = -\sum_{s=0}^{t-1} \frac{\eta}{nm} \cdot \ell_i'^{(s)} \cdot \sigma'(\langle \mathbf{w}_{j,r}^{(s)}, \boldsymbol{\xi}_i \rangle) \cdot \|\boldsymbol{\xi}_i\|_2^2 \cdot \mathbb{1}(y_i = -j),$$

which completes the proof.

$\square$

**Parameters.** Let In the following analysis, we demonstrate that for effective private learning, the learning of feature signals and noise will remain controlled throughout the training process. Let

$$T_p^* = \eta^{-1} \min\{(\text{poly}(\Gamma^{-1}, \|\mathbf{v}\|_2^{-1}, d^{-1}\sigma_n^{-2}, \sigma_0^{-1}, n, m, d)), \frac{mn\varepsilon \log(d)}{5C\mu(\|\mathbf{v}\|_2 + \|\boldsymbol{\xi}\|_2)},$$

represent the maximum allowable number of iterations. Denote $\alpha = \log(T_p^*)$.

$$\eta = O(\min\{nm/(q\sigma_\xi^2 d), nm/(q2^{q+2}\alpha^{q-2}\sigma_\xi^2 d), mn/(q2^{q+2}\alpha^{q-2}\|\mathbf{v}\|_2^2)\}) \tag{10}$$

$$d \geq 1024 \log(4n^2/\delta)\alpha^2 n^2 \tag{11}$$

Let $\beta = 2\max_{i,j,r}\{|\langle \mathbf{w}_{j,r}^{(0)}, \mathbf{v}\rangle|, |\langle \mathbf{w}_{j,r}^{(0)}, \boldsymbol{\xi}_i\rangle|\}$. By Lemma E.3, with probability at least $1 - \delta$, we can bound $\beta$ as follows: $\beta \leq 4\sqrt{\log(\frac{8mn}{\delta})} \cdot \sigma_0 \cdot \max\{\|\mathbf{v}\|_2, \sigma_\xi\sqrt{d}\}$. Using $\sigma_0$ (we can add additional $\log$ constrain on the $\sigma_0$ in the main), and Equation (11), it can be obtained that

$$4\max\left\{\beta, 8n\sqrt{\frac{\log(\frac{4n^2}{\delta})}{d}}\alpha, \frac{\eta C T_p^* \mu \log(1/\delta)((\|\mathbf{v}\|_2 + \|\boldsymbol{\xi}\|_2))}{mn\varepsilon},\right\} \leq 1. \tag{12}$$

Given that the above conditions hold, we claim that the following property is satisfied for $0 \leq t \leq T_p^*$.

**Proposition F.3.** *Under Condition 4.2, for $0 \leq t \leq T_p^*$, we have that*

$$0 \leq \Gamma_{j,r}^{(t)}, \bar{\Phi}_{j,r,i}^{(t)} \leq \alpha, \tag{13}$$

$$0 \geq \underline{\Phi}_{j,r,i}^{(t)} \geq -\beta - 16n\sqrt{\frac{\log(4n^2/\delta)}{d}}\alpha - 0.2 \geq -\alpha. \tag{14}$$

*for all $r \in [m], j \in \{\pm 1\}$ and $i \in [n]$.*

**Bounds of coefficients.** In the following, we first prove the bounds of coefficients.

**Lemma F.4.** *For any $t \geq 0$, it holds that $\langle \mathbf{w}_{j,r}^{(t)} - \mathbf{w}_{j,r}^{(0)}, \mathbf{v}\rangle = j \cdot \Gamma_{j,r}^{(t)} - \eta \sum_{s=1}^{t}\langle \mathbf{z}_s, \mathbf{v}\rangle$ for all $r \in [m], j \in \{\pm 1\}$.*

**Proof of Lemma F.4.** For any time $t \geq 0$, according to the decomposition Equation (9), it holds that:

$$\langle \mathbf{w}_{j,r}^{(t)} - \mathbf{w}_{j,r}^{(0)}, \mathbf{v}\rangle = j \cdot \Gamma_{j,r}^{(t)} + \sum_{i'=1}^{n}\bar{\Phi}_{j,r,i'}^{(t)}\|\boldsymbol{\xi}_{i'}\|_2^{-2} \cdot \langle \boldsymbol{\xi}_{i'}, \mathbf{v}\rangle + \sum_{i'=1}^{n}\underline{\Phi}_{j,r,i'}^{(t)}\|\boldsymbol{\xi}_{i'}\|_2^{-2} \cdot \langle \boldsymbol{\xi}_{i'}, \mathbf{v}\rangle - \eta\sum_{s=1}^{t}\langle \mathbf{z}_s, \mathbf{v}\rangle$$

$$= j \cdot \Gamma_{j,r}^{(t)} - \eta\sum_{s=1}^{t}\langle \mathbf{z}_s, \mathbf{v}\rangle.$$

$\square$

**Lemma F.5.** *Under Parameter Choices, suppose Proposition F.3 holds at iteration t. Then, it holds that*

$$When\ y_i \neq j: \quad \langle \mathbf{w}_{j,r}^{(t)} - \mathbf{w}_{j,r}^{(0)}, \boldsymbol{\xi}_i\rangle \{ \begin{array}{l} \leq \underline{\Phi}_{j,r,i}^{(t)} + 8n\sqrt{\frac{\log(4n^2/\delta)}{d}}\alpha - \eta\sum_{s=1}^{t}\langle \mathbf{z}_s, \boldsymbol{\xi}_i\rangle, \\ \geq \underline{\Phi}_{j,r,i}^{(t)} - 8n\sqrt{\frac{\log(4n^2/\delta)}{d}}\alpha - \eta\sum_{s=1}^{t}\langle \mathbf{z}_s, \boldsymbol{\xi}_i\rangle \end{array}.$$

$$When\ y_i = j: \quad \langle \mathbf{w}_{j,r}^{(t)} - \mathbf{w}_{j,r}^{(0)}, \boldsymbol{\xi}_i\rangle \{ \begin{array}{l} \leq \bar{\Phi}_{j,r,i}^{(t)} + 8n\sqrt{\frac{\log(4n^2/\delta)}{d}}\alpha - \eta\sum_{s=1}^{t}\langle \mathbf{z}_s, \boldsymbol{\xi}_i\rangle, \\ \geq \bar{\Phi}_{j,r,i}^{(t)} - 8n\sqrt{\frac{\log(4n^2/\delta)}{d}}\alpha - \eta\sum_{s=1}^{t}\langle \mathbf{z}_s, \boldsymbol{\xi}_i\rangle \end{array}.$$

*for all $r \in [m], j \in \{\pm 1\}$ and $i \in [n]$.*

**Proof of Lemma F.5.** For $j \neq y_i$, it holds that $\bar{\Phi}_{j,r,i}^{(t)} = 0$ and

$$\langle \mathbf{w}_{j,r}^{(t)} - \mathbf{w}_{j,r}^{(0)}, \boldsymbol{\xi}_i\rangle = \sum_{i'=1}^{n}\bar{\Phi}_{j,r,i'}^{(t)}\|\boldsymbol{\xi}_{i'}\|_2^{-2} \cdot \langle \boldsymbol{\xi}_{i'}, \boldsymbol{\xi}_i\rangle + \sum_{i'=1}^{n}\underline{\Phi}_{j,r,i'}^{(t)}\|\boldsymbol{\xi}_{i'}\|_2^{-2} \cdot \langle \boldsymbol{\xi}_{i'}, \boldsymbol{\xi}_i\rangle - \eta\sum_{s=1}^{t}\langle \mathbf{z}_s, \boldsymbol{\xi}_i\rangle$$

$$\overset{(i)}{\leq} 4\sqrt{\frac{\log(4n^2/\delta)}{d}}\sum_{i'\neq i}|\bar{\Phi}_{j,r,i'}^{(t)}| + 4\sqrt{\frac{\log(4n^2/\delta)}{d}}\sum_{i'\neq i}|\underline{\Phi}_{j,r,i'}^{(t)}| + \underline{\Phi}_{j,r,i}^{(t)} - \eta\sum_{s=1}^{t}\langle \mathbf{z}_s, \boldsymbol{\xi}_i\rangle$$

$$\overset{(ii)}{\leq} \underline{\Phi}_{j,r,i}^{(t)} + 8n\sqrt{\frac{\log(4n^2/\delta)}{d}}\alpha - \eta\sum_{s=1}^{t}\langle \mathbf{z}_s, \boldsymbol{\xi}_i\rangle.$$

The second inequality (i) is derived by Lemma E.2 and the last inequality (ii) holds due to Proposition F.3. Similarly, for $y_i = j$, it holds that $\underline{\Phi}_{j,r,i}^{(t)} = 0$ and

$$
\begin{aligned}
\langle \mathbf{w}_{j,r}^{(t)} - \mathbf{w}_{j,r}^{(0)}, \boldsymbol{\xi}_i \rangle &= \sum_{i'=1}^{n} \bar{\Phi}_{j,r,i'}^{(t)} \|\boldsymbol{\xi}_{i'}\|_2^{-2} \cdot \langle \boldsymbol{\xi}_{i'}, \boldsymbol{\xi}_i \rangle + \sum_{i'=1}^{n} \Phi_{j,r,i'}^{(t)} \|\boldsymbol{\xi}_{i'}\|_2^{-2} \cdot \langle \boldsymbol{\xi}_{i'}, \boldsymbol{\xi}_i \rangle - \eta \sum_{s=1}^{t} \langle \mathbf{z}_s, \boldsymbol{\xi}_i \rangle \\
&\overset{\text{(iii)}}{\le} 4\sqrt{\frac{\log(4n^2/\delta)}{d}} \sum_{i' \ne i} |\bar{\Phi}_{j,r,i'}^{(t)}| + 4\sqrt{\frac{\log(4n^2/\delta)}{d}} \sum_{i' \ne i} |\underline{\Phi}_{j,r,i'}^{(t)}| + \bar{\Phi}_{j,r,i}^{(t)} - \eta \sum_{s=1}^{t} \langle \mathbf{z}_s, \boldsymbol{\xi}_i \rangle \\
&\overset{\text{(iv)}}{\le} \bar{\Phi}_{j,r,i}^{(t)} + 8n\sqrt{\frac{\log(4n^2/\delta)}{d}} \alpha - \eta \sum_{s=1}^{t} \langle \mathbf{z}_s, \boldsymbol{\xi}_i \rangle.
\end{aligned}
$$

Moreover, it is clear that, according to Lemma E.2, inequalities (i) can also be bounded by:

$$
\begin{aligned}
\text{When } y_i \ne j: \overset{\text{(i)}'}{\ge} &\underline{\Phi}_{j,r,i}^{(t)} - 4\sqrt{\frac{\log(4n^2/\delta)}{d}} \sum_{i' \ne i} |\bar{\Phi}_{j,r,i'}^{(t)}| - 4\sqrt{\frac{\log(4n^2/\delta)}{d}} \sum_{i' \ne i} |\underline{\Phi}_{j,r,i'}^{(t)}| - \eta \sum_{s=1}^{t} \langle \mathbf{z}_s, \boldsymbol{\xi}_i \rangle \\
&\overset{\text{(ii)}'}{\ge} \underline{\Phi}_{j,r,i}^{(t)} - 8n\sqrt{\frac{\log(4n^2/\delta)}{d}} \alpha - \eta \sum_{s=1}^{t} \langle \mathbf{z}_s, \boldsymbol{\xi}_i \rangle,
\end{aligned}
$$

Similarly, the following also holds for the inequalities (iii):

$$
\begin{aligned}
\text{When } y_i = j: \overset{\text{(iii)}'}{\ge} &\underline{\Phi}_{j,r,i}^{(t)} - 4\sqrt{\frac{\log(4n^2/\delta)}{d}} \sum_{i' \ne i} |\bar{\Phi}_{j,r,i'}^{(t)}| - 4\sqrt{\frac{\log(4n^2/\delta)}{d}} \sum_{i' \ne i} |\underline{\Phi}_{j,r,i'}^{(t)}| - \eta \sum_{s=1}^{t} \langle \mathbf{z}_s, \boldsymbol{\xi}_i \rangle \\
&\overset{\text{(iv)}'}{\ge} \bar{\Phi}_{j,r,i}^{(t)} - 8n\sqrt{\frac{\log(4n^2/\delta)}{d}} \alpha - \eta \sum_{s=1}^{t} \langle \mathbf{z}_s, \boldsymbol{\xi}_i \rangle,
\end{aligned}
$$

which completed the proof.

$\square$

**Lemma F.6.** *Under Parameter Choices, suppose Proposition F.3 holds at iteration t. Then, it holds that*

$$
\langle \mathbf{w}_{j,r}^{(t)}, y_i \mathbf{v} \rangle \le \langle \mathbf{w}_{j,r}^{(t)}, y_i \mathbf{v} \rangle - \eta \sum_{s=1}^{t} \langle \mathbf{z}_s, y_i \mathbf{v} \rangle,
$$

$$
\langle \mathbf{w}_{j,r}^{(t)}, \boldsymbol{\xi}_i \rangle \le \langle \mathbf{w}_{j,r}^{(t)}, \boldsymbol{\xi}_i \rangle - \eta \sum_{s=1}^{t} \langle \mathbf{z}_s, \boldsymbol{\xi}_i \rangle + 8n \frac{\log(4n^2/\delta)}{d} \alpha,
$$

$$
F_j(\mathbf{W}_j^{(t)}, \mathbf{x}_i) \le 1
$$

*for all $r \in [m]$ and $j \ne y_i$.*

**Proof of Lemma F.6.** According to Lemma F.2, it is clear that $\Gamma_{j,r}^{(t)}$ is increasing and $\Gamma_{j,r}^{(t)} \ge 0$ holds. For $y_i \ne j$, it holds that

$$
\langle \mathbf{w}_{j,r}^{(t)}, y_i \mathbf{v} \rangle = \langle \mathbf{w}_{j,r}^{(0)}, y_i \mathbf{v} \rangle + y_i \cdot j \cdot \Gamma_{j,r}^{(t)} - \eta \sum_{s=1}^{t} \langle \mathbf{z}_s, y_i \mathbf{v} \rangle \le \langle \mathbf{w}_{j,r}^{(0)}, y_i \mathbf{v} \rangle - \eta \sum_{s=1}^{t} \langle \mathbf{z}_s, y_i \mathbf{v} \rangle.
$$

Moreover, according to Lemma F.5, we have

$$
\langle \mathbf{w}_{j,r}^{(t)}, \boldsymbol{\xi}_i \rangle \le \langle \mathbf{w}_{j,r}^{(0)}, \boldsymbol{\xi}_i \rangle + \bar{\Phi}_{j,r,i}^{(t)} + 8n\sqrt{\frac{\log(4n^2/\delta)}{d}} \alpha - \eta \sum_{s=1}^{t} \langle \mathbf{z}_s, \boldsymbol{\xi}_i \rangle \le \langle \mathbf{w}_{j,r}^{(0)}, \boldsymbol{\xi}_i \rangle + 8n\sqrt{\frac{\log(4n^2/\delta)}{d}} \alpha - \eta \sum_{s=1}^{t} \langle \mathbf{z}_s, \boldsymbol{\xi}_i \rangle,
$$

where the last inequality is due to $\underline{\Phi}_{j,r,i}^{(t)} \le 0$. Therefore, we can further obtain

$$
\begin{aligned}
F_j(\mathbf{W}_j^{(t)}, \mathbf{x}_i) &= \frac{1}{m} \sum_{r=1}^{m} [\sigma(\langle \mathbf{w}_{j,r}^{(t)}, -j \cdot \mathbf{v} \rangle) + \sigma(\langle \mathbf{w}_{j,r}^{(t)}, \boldsymbol{\xi}_i \rangle)] \\
&\le 2^{q+1} \max_{j,r,i} \{ |\langle \mathbf{w}_{j,r}^{(0)}, \mathbf{v} \rangle|, |\langle \mathbf{w}_{j,r}^{(0)}, \boldsymbol{\xi}_i \rangle|, 8n\sqrt{\frac{\log(4n^2/\delta)}{d}}\alpha, \frac{\eta C T_p^* \mu \log(1/\delta)(\|\mathbf{v}\|_2 + \|\boldsymbol{\xi}\|_2)}{mn\varepsilon} \}^q \\
&\le 1.
\end{aligned}
$$

The first inequality derives from the previous two conclusions and the second inequality is by Equation (12). $\qquad \square$

**Lemma F.7.** *Under Parameter Choices, suppose Proposition F.3 holds at iteration t. Then, it holds that*

$$
\langle \mathbf{w}_{j,r}^{(t)}, y_i \mathbf{v} \rangle = \langle \mathbf{w}_{j,r}^{(0)}, y_i \mathbf{v} \rangle + \Gamma_{j,r}^{(t)} - \eta \sum_{s=1}^{t} \langle \mathbf{z}_s, y_i \mathbf{v} \rangle,
$$

$$
\langle \mathbf{w}_{j,r}^{(t)}, \boldsymbol{\xi}_i \rangle \le \langle \mathbf{w}_{j,r}^{(0)}, \boldsymbol{\xi}_i \rangle + \bar{\Phi}_{j,r,i}^{(t)} + 8n\sqrt{\frac{\log(4n^2/\delta)}{d}}\alpha - \eta \sum_{s=1}^{t} \langle \mathbf{z}_s, \boldsymbol{\xi}_i \rangle,
$$

*for all $r \in [m]$ and $j \in \{\pm 1\}$ and $i \in [n]$. If $\max\{\bar{\Phi}_{j,r,i}^{(t)}, \Gamma_{j,r}^{(t)}\} = O(1)$, we further have $F_j(\mathbf{W}_j^{(t)}, \mathbf{x}_i) = O(1)$.*

**Proof of Lemma F.7.** According to Lemma F.2, it is clear that, for $y_i = j$, we have

$$
\langle \mathbf{w}_{j,r}^{(t)}, y_i \mathbf{v} \rangle = \langle \mathbf{w}_{j,r}^{(0)}, y_i \mathbf{v} \rangle + \Gamma_{j,r}^{(t)} - \eta \sum_{s=1}^{t} \langle \mathbf{z}_s, y_i \mathbf{v} \rangle.
$$

Moreover, according to Lemma F.5, we have

$$
\langle \mathbf{w}_{j,r}^{(t)}, \boldsymbol{\xi}_i \rangle \le \langle \mathbf{w}_{j,r}^{(0)}, \boldsymbol{\xi}_i \rangle + \bar{\Phi}_{j,r,i}^{(t)} + 8n\sqrt{\frac{\log(4n^2/\delta)}{d}}\alpha - \eta \sum_{s=1}^{t} \langle \mathbf{z}_s, \boldsymbol{\xi}_i \rangle.
$$

If $\max\{\bar{\Phi}_{j,r,i}^{(t)}, \Gamma_{j,r}^{(t)}\} = O(1)$, we have

$$
\begin{aligned}
F_j(\mathbf{W}_j^{(t)}, \mathbf{x}_i) &= \frac{1}{m} \sum_{r=1}^{m} [\sigma(\langle \mathbf{w}_{j,r}^{(t)}, -j \cdot \mathbf{v} \rangle) + \sigma(\langle \mathbf{w}_{j,r}^{(t)}, \boldsymbol{\xi}_i \rangle)] \\
&\le 2 * 5^q \max_{j,r,i} \{ |\langle \mathbf{w}_{j,r}^{(0)}, \mathbf{v} \rangle|, |\langle \mathbf{w}_{j,r}^{(0)}, \boldsymbol{\xi}_i \rangle|, 8n\sqrt{\frac{\log(4n^2/\delta)}{d}}\alpha, \frac{\eta C T_p^* \mu \log(1/\delta)(\|\mathbf{v}\|_2 + \|\boldsymbol{\xi}\|_2)}{mn\varepsilon} \}^q \\
&= O(1).
\end{aligned}
$$

The first inequality derives from the previous two conclusions and the second inequality is by Equation (12). $\qquad \square$

**Lemma F.8.** *For any iteration t, with probability $1 - \delta$, it holds that*

$$
|\eta \sum_{s=1}^{t} \langle \mathbf{z}_s, \mathbf{v} \rangle| \le \frac{\eta C \sqrt{tT} \|\mathbf{v}\|_2^2 \log(1/\delta)}{mn\varepsilon}(1 + \frac{1}{\text{SNR}}), \quad |\eta \sum_{s=1}^{t} \langle \mathbf{z}_s, \boldsymbol{\xi}_i \rangle| \le \frac{\eta C \sqrt{tT} \|\boldsymbol{\xi}\|_2^2 \log(1/\delta)}{mn\varepsilon}(1 + \text{SNR}). \tag{15}
$$

**Proof of Lemma F.8.** According to Proposition F.3 and the update of $\mathbf{w}_{j,r}^{(t)}$ in Equation (4), it is clear that the sensitivity can be bounded by $\frac{C}{mn}(\|\mathbf{v}\|_2 + \|\boldsymbol{\xi}\|_2)$ with $C = \widetilde{O}(1)$. Therefore, with probability $1 - \delta$, we have:

$$
|\eta \sum_{s=1}^{t} \langle \mathbf{z}_s, \mathbf{v} \rangle| \le \frac{\eta C \sqrt{tT} \|\mathbf{v}\|_2^2 \log(1/\delta)}{mn\varepsilon}(1 + \frac{1}{\text{SNR}}), \quad |\eta \sum_{s=1}^{t} \langle \mathbf{z}_s, \boldsymbol{\xi}_i \rangle| \le \frac{\eta C \sqrt{tT} \|\boldsymbol{\xi}\|_2^2 \log(1/\delta)}{mn\varepsilon}(1 + \text{SNR}).
$$

where we use Hoeffding's inequality and the definition of $\text{SNR} = \|\mathbf{v}\|_2 / \|\boldsymbol{\xi}\|_2$.

$\qquad \square$

Now we are ready to provide the proof of Proposition F.3.

**Proof of Proposition F.3.** We prove Proposition F.3 using an induction process. It is clear that the claim holds for $t = 0$ since the coefficients are zero in this case. Suppose there exists $s \leq T_p^*$ such that Proposition F.3 holds for all $t \leq s - 1$. Our goal is to prove that the claim also holds for $t = s$.

We first consider when $\underline{\Phi}_{j,r,i}^{(t)} \leq -0.5\beta - 8n\sqrt{\frac{\log(4n^2/\delta)}{d}}\alpha - 0.1$. Notice that for any $j = y_i$, we have $\underline{\Phi}_{j,r,i}^{(t)} = 0$. Thus, we only need to focus on $j \neq y_i$. Additionally, according to Lemma F.5 and Lemma F.8, it holds that

$$\langle \mathbf{w}_{j,r}^{(t)}, \boldsymbol{\xi}_i \rangle \leq \underline{\Phi}_{j,r,i}^{(t)} + \langle \mathbf{w}_{j,r}^{(0)}, \boldsymbol{\xi}_i \rangle + 8n\sqrt{\frac{\log(4n^2/\delta)}{d}}\alpha - \eta\sum_{s=1}^{t}\langle \mathbf{z}_s, \boldsymbol{\xi}_i \rangle \leq 0.$$

Therefore, for $t + 1$, we have

$$\begin{aligned}
\underline{\Phi}_{j,r,i}^{(t+1)} &= \underline{\Phi}_{j,r,i}^{(t)} + \frac{\eta}{nm} \cdot \ell_i'^{(t)} \cdot \sigma'(\langle \mathbf{w}_{j,r}^{(t)}, \boldsymbol{\xi}_i \rangle) \cdot \mathbb{1}(y_i = -j)\|\boldsymbol{\xi}_i\|_2^2 \\
&= \underline{\Phi}_{j,r,i}^{(t)} \\
&\geq -\beta - 16n\sqrt{\frac{\log(4n^2/\delta)}{d}}\alpha - 0.2.
\end{aligned}$$

When considering $\underline{\Phi}_{j,r,i}^{(t)} \geq -0.5\beta - 8n\sqrt{\frac{\log(4n^2/\delta)}{d}}\alpha - 0.1$, it holds that

$$\begin{aligned}
\underline{\Phi}_{j,r,i}^{(t+1)} &= \underline{\Phi}_{j,r,i}^{(t)} + \frac{\eta}{nm} \cdot \ell_i'^{(t)} \cdot \sigma'(\langle \mathbf{w}_{j,r}^{(T-1)}, \boldsymbol{\xi}_i \rangle) \cdot \mathbb{1}(y_i = -j)\|\boldsymbol{\xi}_i\|_2^2 \\
&\overset{(i)}{\geq} -0.5\beta - 8n\sqrt{\frac{\log(4n^2/\delta)}{d}}\alpha - 0.1 - O(\frac{\eta\sigma_\xi^2 d}{nm})\sigma'(0.5\beta + 8n\sqrt{\frac{\log(4n^2/\delta)}{d}}\alpha + 0.1) \\
&\overset{(ii)}{\geq} -0.5\beta - 8n\sqrt{\frac{\log(4n^2/\delta)}{d}}\alpha - 0.1 - O(\frac{\eta q\sigma_\xi^2 d}{nm})(0.5\beta + 8n\sqrt{\frac{\log(4n^2/\delta)}{d}}\alpha + 0.1) \\
&\overset{(iii)}{\geq} -\beta - 16n\sqrt{\frac{\log(4n^2/\delta)}{d}}\alpha - 0.2,
\end{aligned}$$

where $(i)$ holds due to $\ell_i'^{(t)} \leq 1$ and $\|\boldsymbol{\xi}_i\|_2 = O(\sigma_\xi^2 d)$; $(ii)$ holds because of $0.5\beta + 8n\sqrt{\frac{\log(4n^2/\delta)}{d}}\alpha + 0.1 \leq 1$; $(iii)$ derives from $\eta = O(nm/(q\sigma_\xi^2 d))$.

Now, we proceed to prove Equation (13). Recall the update rule for $\Phi_{j,r,i}^{(t+1)}$ and denote $t_{j,r,i}$ as the first time such that $\Phi_{j,r,i}^{(t)} \geq 0.5\alpha$, then we can decompose the following

$$\begin{aligned}
\bar{\Phi}_{j,r,i}^{(t+1)} &= \bar{\Phi}_{j,r,i}^{(t)} - \frac{\eta}{nm} \cdot \ell_i'^{(t)} \cdot \sigma'(\langle \mathbf{w}_{j,r}^{(t)}, \boldsymbol{\xi}_i \rangle) \cdot \mathbb{1}(y_i = j)\|\boldsymbol{\xi}_i\|_2^2 \\
&= \bar{\Phi}_{j,r,i}^{(t_{j,r,i})} \underbrace{- \frac{\eta}{nm} \cdot \ell_i'^{(t_{j,r,i})} \cdot \sigma'(\langle \mathbf{w}_{j,r}^{(t)}, \boldsymbol{\xi}_i \rangle) \cdot \mathbb{1}(y_i = j)\|\boldsymbol{\xi}_i\|_2^2}_{\text{Term 1}} \\
&\quad \underbrace{- \sum_{p=t_{j,r,i}+1}^{t} \frac{\eta}{nm}\ell_i'^{(p)} \cdot \sigma'(\langle \mathbf{w}_{j,r}^{(p)}, \boldsymbol{\xi}_i \rangle) \cdot \mathbb{1}(y_i = j)\|\boldsymbol{\xi}_i\|_2^2}_{\text{Term 2}}.
\end{aligned} \tag{16}$$

Then, we need to bound Term 1 and Term 2, respectively. For Term 1, we have

$$\begin{aligned}
|\text{Term 1}| &= \frac{\eta}{nm} \cdot \ell_i'^{(t_{j,r,i})} \cdot \sigma'(\langle \mathbf{w}_{j,r}^{(t)}, \boldsymbol{\xi}_i \rangle) \cdot \mathbb{1}(y_i = j)\|\boldsymbol{\xi}_i\|_2^2 \\
&\overset{(i)}{\leq} 2qn^{-1}m^{-1}\eta(y_i\Gamma_{j,r}^{(t_{j,r})} + \langle \mathbf{w}_{j,r}^{(0)}, \boldsymbol{\xi}_i \rangle + 0.2)^{q-1}\|\boldsymbol{\xi}_i\|_2^2 \\
&\overset{(ii)}{\leq} 2^q n^{-1}m^{-1}\eta\alpha^{q-1}\|\boldsymbol{\xi}_i\|_2^2 \\
&\overset{(iii)}{\leq} 0.25\alpha.
\end{aligned}$$

Here, $(i)$ holds due to Lemma F.4, Lemma E.3, Lemma F.8 and the parameter choices of $T_p^*$; $(ii)$ derives from the parameter choices of $\sigma_0$ and induction hypothesis; $(iii)$ holds by $\eta \leq O(nm/(q2^{q+2}\alpha^{q-2}\|\boldsymbol{\xi}\|_2^2))$.

For Term 2 and $y_i = j$, we have

$$
\begin{aligned}
\langle \mathbf{w}_{j,r}^{(t+1)}, \boldsymbol{\xi}_i \rangle =& \langle \mathbf{w}_{j,r}^{(0)}, \boldsymbol{\xi}_i \rangle + \bar{\Phi}_{j,r,i}^{(t)} - 8n\sqrt{\frac{\log(4n^2/\delta)}{d}}\alpha - \eta \sum_{s=1}^{t} \langle \mathbf{z}_s, \cdot \boldsymbol{\xi}_i \rangle \\
& \overset{(i)}{\geq} -0.5\beta + 0.5\alpha - 0.2 \geq 0.25\alpha.
\end{aligned}
$$

Here, $(i)$ holds due to the definition of $t_{j,r,i}$ and Lemma E.3. Similarly, we can also upper bound $\langle \mathbf{w}_{j,r}^{(t+1)}, \boldsymbol{\xi}_i \rangle$ as follows:

$$
\begin{aligned}
\langle \mathbf{w}_{j,r}^{(t+1)}, \boldsymbol{\xi}_i \rangle =& \langle \mathbf{w}_{j,r}^{(0)}, \boldsymbol{\xi}_i \rangle + \bar{\Phi}_{j,r,i}^{(t)} + 8n\sqrt{\frac{\log(4n^2/\delta)}{d}}\alpha - \eta \cdot \sum_{s=1}^{t} \langle \mathbf{z}_s, \boldsymbol{\xi}_i \rangle \\
& \leq 0.5\beta + \alpha + 0.2 \leq 2\alpha.
\end{aligned}
$$

Combining the above upper and lower bounds of $\langle \mathbf{w}_{j,r}^{(t+1)}, \boldsymbol{\xi}_i \rangle$ into Term 2, it holds that

$$
\begin{aligned}
|\text{Term } 2| =& \Big| \sum_{p=t_{j,r,i}+1}^{t} \frac{\eta}{nm} \cdot \ell_i'^{(p)} \cdot \sigma'(\langle \mathbf{w}_{j,r}^{(p)}, \boldsymbol{\xi}_i \rangle) \cdot \mathbb{1}(y_i = j)\|\boldsymbol{\xi}_i\|_2^2 \Big| \\
& \overset{(i)}{\leq} \sum_{p=t_{j,r}+1}^{t} \frac{\eta}{nm} \cdot \exp(-\sigma(\langle \mathbf{w}_{j,r}^{(t)}, \boldsymbol{\xi}_i \rangle) + 1) \cdot \sigma'(\langle \mathbf{w}_{j,r}^{(t)}, \boldsymbol{\xi}_i \rangle) \cdot \|\boldsymbol{\xi}_i\|_2^2 \\
& \overset{(ii)}{\leq} eq2^q \eta T_p^* \exp(-\alpha^q/4^q)\alpha^{q-1}\|\boldsymbol{\xi}_i\|_2^2 \\
& \overset{(i)}{\leq} 0.25 T_p^* \exp(-\alpha^q/4^q)\alpha \\
& \overset{(iii)}{\leq} 0.25 T_p^* \exp(-\log(T_p^*)^q)\alpha \\
& \overset{(iv)}{\leq} 0.25\alpha.
\end{aligned}
$$

Here, $(ii)$ follows from Lemma E.2; $(ii)$ is derived from the parameter choice of $\eta$; $(iii)$ and $(iv)$ holds due to the choice $\alpha = 4\log(T_p^*)$ and the fact $\log^q(T_p^*) \geq \log(T_p^*)$. Additionally, $(i)$ is established as follows:

$$
\begin{aligned}
|\ell_i'^{(t)}| =& \frac{1}{1 + \exp\{y_i \cdot [F_{+1}(\mathbf{W}_{+1}^{(t)}, \mathbf{x}_i) - F_{-1}(\mathbf{W}_{-1}^{(t)}, \mathbf{x}_i)]\}} \\
& \leq \exp\{-y_i \cdot [F_{+1}(\mathbf{W}_{+1}^{(t)}, \mathbf{x}_i) - F_{-1}(\mathbf{W}_{-1}^{(t)}, \mathbf{x}_i)]\} \\
& \leq \exp\{-F_{y_i}(\mathbf{W}_{y_i}^{(t)}, \mathbf{x}_i) + 1\},
\end{aligned}
$$

where the last inequality follows from Lemma F.6. Combining the bounds for Term 1 and Term 2 into Equation (16), we complete the proof of Equation (13). The same procedure applies to prove $\Gamma \leq \alpha$, with parameter choice $\eta = O(nm/(q2^{q+2}\alpha^{q-2}\|\mathbf{v}\|_2^2))$. $\qquad\square$

**Lemma F.9.** *Under parameter choices, for $0 \leq t \leq T_p^*$, with probability $1 - \delta$, the following result holds.*

$$
\|\nabla L_D(\mathbf{W}^{(t)})\|_F^2 \leq O(\max\{\|\mathbf{v}\|_2^2, \sigma_\xi^2 d\}) L_D(\mathbf{W}^{(t)}) + O(\sigma_z^2 d \log(1/\delta)).
$$

**Proof of Lemma F.9.** According to the triangle inequality and the definition of noisy gradient, we have

$$
\|\nabla L_S(\mathbf{W}^{(t)})\|_F^2 \leq 2\Big[\frac{1}{n}\sum_{i=1}^{n} \ell'(y_i f(\mathbf{W}^{(t)}, \mathbf{x}_i))\|\nabla f(\mathbf{W}^{(t)}, \mathbf{x}_i)\|_F\Big]^2 + 2\|\mathbf{z}_t\|_F^2. \tag{17}
$$

The first term is bounded using Lemma C.7 in (Cao et al., 2022), as it shares the same properties, while the second term is bounded by Lemma E.2. This concludes the proof. $\qquad\square$

# G. Signal Learning

## G.1. First Stage

**Lemma G.1** (Restatement of Theorem 4.6). *Under the same conditions as signal learning, in particular, if we choose*

$$\text{SNR} \cdot n\varepsilon \geq \frac{4^q \log(16/e^{1/2}\sigma_0\|\mathbf{v}\|_2)}{C_1 q}, n \cdot \text{SNR}^q \geq C_1 \log(6/\sigma_0\|\mathbf{v}\|_2)2^{2q+6}[4\log(8mn/\delta)]^{(q-1)/2} \tag{18}$$

*where $C_1 = O(1)$ is a positive constant, there exists $T_1 = \frac{\log(16/\sigma_0\|\mathbf{v}\|_2)4^{q-1}m}{C_1\eta q\sigma_0^{q-2}\|\mathbf{v}\|_2^q}$ such that*

- $\max_r \Gamma_{j,r}^{(T_1)} = \Omega(1)$ *for $j \in \{\pm 1\}$.*

- $|\Phi_{j,r,i}^{(t)}| = O(\sigma_0\sigma_\xi\sqrt{d})$ *for all $j \in \{\pm 1\}, r \in [m], i \in [n]$ and $0 \leq t \leq T_1$.*

**Proof of Lemma G.1.** First, when $t \leq T_1^+ = \min\{\frac{nm\eta^{-1}\sigma_0^{2-q}\sigma_\xi^{-q}d^{-q/2}}{2^{q+4}q[4\log(8mn/\delta)]^{(q-1)/2}}, \frac{\sigma_0 mn\varepsilon}{\eta\|\boldsymbol{\xi}\|_2(\|\mathbf{v}\|_2+\|\boldsymbol{\xi}\|_2)}, \frac{\sigma_0 mn\varepsilon}{\eta(\|\mathbf{v}\|_2+\|\boldsymbol{\xi}\|_2)}\}$, it can be noticed that the noise remains well controlled. To proceed, define $\Psi^{(t)} = \max_{j,r,i}|\Phi_{j,r,i}^{(t)}| = \max_{j,r,i}\{\bar{\Phi}_{j,r,i}^{(t)}, -\underline{\Phi}_{j,r,i}^{(t)}\}$. and assume $\Psi^{(t)} \leq \frac{\sigma_0\sigma_\xi\sqrt{d}}{2}$ for all $0 \leq t \leq T_1^+$. Then, we aim to prove that the same holds for $t+1$ using an induction process. Recall the update of $\bar{\Phi}$ and $\underline{\Phi}$ as follows:

$$\bar{\Phi}_{j,r,i}^{(t)} = -\sum_{s=0}^{t-1}\frac{\eta}{nm}\cdot\ell_i'^{(s)}\cdot\sigma'(\langle\mathbf{w}_{j,r}^{(s)}, \boldsymbol{\xi}_i\rangle)\cdot\|\boldsymbol{\xi}_i\|_2^2\cdot\mathbb{1}(y_i = j),$$

$$\underline{\Phi}_{j,r,i}^{(t)} = -\sum_{s=0}^{t-1}\frac{\eta}{nm}\cdot\ell_i'^{(s)}\cdot\sigma'(\langle\mathbf{w}_{j,r}^{(s)}, \boldsymbol{\xi}_i\rangle)\cdot\|\boldsymbol{\xi}_i\|_2^2\cdot\mathbb{1}(y_i = -j).$$

Moreover, according to Definition 4.1, we have

$$\mathbf{w}_{j,r}^{(t)} = \mathbf{w}_{j,r}^{(0)} + j\cdot\Gamma_{j,r}^{(t)}\cdot\frac{\mathbf{v}}{\|\mathbf{v}\|_2^2} + \sum_{i=1}^n\Phi_{j,r,i}^{(t)}\cdot\frac{\boldsymbol{\xi}_i}{\|\boldsymbol{\xi}_i\|_2^2} - \eta\sum_{s=1}^t\mathbf{z}_s.$$

Substituting this expression into the updates for $\bar{\Phi}$ and $\Phi$, it follows that:

$$\bar{\Phi}_{j,r,i}^{(t+1)} = \bar{\Phi}_{j,r,i}^{(t)} - \frac{\eta}{nm}\cdot\ell_i'^{(t)}\sigma'(\langle\mathbf{w}_{j,r}^{(0)}, \boldsymbol{\xi}_i\rangle + \sum_{i'=1}^n\bar{\Phi}_{j,r,i'}^{(t)}\frac{\langle\boldsymbol{\xi}_{i'}, \boldsymbol{\xi}_i\rangle}{\|\boldsymbol{\xi}_{i'}\|_2^2} + \sum_{i'=1}^n\underline{\Phi}_{j,r,i'}^{(t)}\frac{\langle\boldsymbol{\xi}_{i'}, \boldsymbol{\xi}_i\rangle}{\|\boldsymbol{\xi}_{i'}\|_2^2} - \eta\sum_{s=1}^t\langle\mathbf{z}_s, \boldsymbol{\xi}_i\rangle)\cdot\|\boldsymbol{\xi}_i\|_2^2,$$

$$\underline{\Phi}_{j,r,i}^{(t+1)} = \underline{\Phi}_{j,r,i}^{(t)} + \frac{\eta}{nm}\cdot\ell_i'^{(t)}\sigma'(\langle\mathbf{w}_{j,r}^{(0)}, \boldsymbol{\xi}_i\rangle + \sum_{i'=1}^n\bar{\Phi}_{j,r,i'}^{(t)}\frac{\langle\boldsymbol{\xi}_{i'}, \boldsymbol{\xi}_i\rangle}{\|\boldsymbol{\xi}_{i'}\|_2^2} + \sum_{i'=1}^n\underline{\Phi}_{j,r,i'}^{(t)}\frac{\langle\boldsymbol{\xi}_{i'}, \boldsymbol{\xi}_i\rangle}{\|\boldsymbol{\xi}_{i'}\|_2^2} - \eta\sum_{s=1}^t\langle\mathbf{z}_s, \boldsymbol{\xi}_i\rangle)\cdot\|\boldsymbol{\xi}_i\|_2^2.$$

Therefore, it holds that

$$\Psi^{(t+1)} \leq \Psi^{(t)} + \max_{j,r,i}\{\frac{\eta}{nm} \cdot |\ell_i'^{(t)}| \cdot \sigma'(\langle \mathbf{w}_{j,r}^{(0)}, \boldsymbol{\xi}_i\rangle + \sum_{i'=1}^n \Psi^{(t)} \cdot \frac{|\langle \boldsymbol{\xi}_{i'}, \boldsymbol{\xi}_i\rangle|}{\|\boldsymbol{\xi}_{i'}\|_2^2}$$

$$+ \sum_{i'=1}^n \Psi^{(t)} \cdot \frac{|\langle \boldsymbol{\xi}_{i'}, \boldsymbol{\xi}_i\rangle|}{\|\boldsymbol{\xi}_{i'}\|_2^2} - \eta \sum_{s=1}^t \langle \mathbf{z}_s, \boldsymbol{\xi}_i\rangle) \cdot \|\boldsymbol{\xi}_i\|_2^2\}$$

$$\overset{(i)}{\leq} \Psi^{(t)} + \max_{j,r,i}\{\frac{\eta}{nm} \cdot \sigma'(\langle \mathbf{w}_{j,r}^{(0)}, \boldsymbol{\xi}_i\rangle + 2 \cdot \sum_{i'=1}^n \Psi^{(t)} \cdot \frac{|\langle \boldsymbol{\xi}_{i'}, \boldsymbol{\xi}_i\rangle|}{\|\boldsymbol{\xi}_{i'}\|_2^2} - \eta \sum_{s=1}^t \langle \mathbf{z}_s, \boldsymbol{\xi}_i\rangle) \cdot \|\boldsymbol{\xi}_i\|_2^2\}$$

$$\overset{(ii)}{=} \Psi^{(t)} + \max_{j,r,i}\{\frac{\eta}{nm} \cdot \sigma'(\langle \mathbf{w}_{j,r}^{(0)}, \boldsymbol{\xi}_i\rangle + 2\Psi^{(t)} + 2 \cdot \sum_{i' \neq i}^n \Psi^{(t)} \cdot \frac{|\langle \boldsymbol{\xi}_{i'}, \boldsymbol{\xi}_i\rangle|}{\|\boldsymbol{\xi}_{i'}\|_2^2} - \eta \sum_{s=1}^t \langle \mathbf{z}_s, \boldsymbol{\xi}_i\rangle) \cdot \|\boldsymbol{\xi}_i\|_2^2\}$$

$$\overset{(iii)}{\leq} \Psi^{(t)} + \frac{\eta q}{nm} \cdot [2 \cdot \sqrt{\log(8mn/\delta)} \cdot \sigma_0 \sigma_\xi \sqrt{d}$$

$$+ (2 + \frac{4n\sigma_\xi^2 \cdot \sqrt{d\log(4n^2/\delta)}}{\sigma_\xi^2 d/2}) \cdot \Psi^{(t)} - \eta \sum_{s=1}^t \langle \mathbf{z}_s, \boldsymbol{\xi}_i\rangle]^{q-1} \cdot 2\sigma_\xi^2 d$$

$$\overset{(iv)}{\leq} \Psi^{(t)} + \frac{\eta q}{nm} \cdot (2 \cdot \sqrt{\log(8mn/\delta)} \cdot \sigma_0 \sigma_\xi \sqrt{d} + 4\Psi^{(t)} - \eta \sum_{s=1}^t \langle \mathbf{z}_s, \boldsymbol{\xi}_i\rangle)^{q-1} \cdot 2\sigma_\xi^2 d$$

$$\overset{(v)}{\leq} \Psi^{(t)} + \frac{\eta q}{nm} \cdot (4 \cdot \sqrt{\log(8mn/\delta)} \cdot \sigma_0 \sigma_\xi \sqrt{d})^{q-1} \cdot 2\sigma_\xi^2 d.$$

Here, the inequality $(i)$ holds due to the fact $|\ell_i'^{(t)}| \leq 1$; the equality $(ii)$ is the decomposition of index $i$; the inequality $(iii)$ comes from Lemma E.2; $(iv)$ is derived from the condition of $d \geq 16n^2 \log(4n^2/\delta)$; $(v)$ follows from the induction hypothesis, Lemma F.8 and $T_1^+$. By applying a telescoping sum over $t$, we obtain the following result:

$$\Psi^{(t+1)} \leq (t+1) \cdot \frac{\eta q}{nm} \cdot (4 \cdot \sqrt{\log(8mn/\delta)} \cdot \sigma_0 \sigma_\xi \sqrt{d})^{q-1} \cdot 2\sigma_\xi^2 d$$

$$\leq T_1^+ \cdot \frac{\eta q}{nm} \cdot (4 \cdot \sqrt{\log(8mn/\delta)} \cdot \sigma_0 \sigma_\xi \sqrt{d})^{q-1} \cdot 2\sigma_\xi^2 d$$

$$\leq \frac{\sigma_0 \sigma_\xi \sqrt{d}}{2},$$

where the second inequality follows from the induction hypothesis, while the last inequality is due to the range of $T_1^+$.

Now, we move forward to the proof of $\Gamma$. Without loss of generality, we first consider $j = 1$. Let $T_{1,1}$ be the first time such that $\max_r \Gamma_{1,r}^{(t)} \leq 2$ in the period of $[0, T_1^+]$, then it also holds that $\max_{j,r,i}\{|\Phi_{j,r,i}^{(t)}|\} = O(\sigma_0 \sigma_\xi \sqrt{d}) = O(1)$.

According to Lemma F.4 and Lemma F.5, it holds that $F_{-1}(\mathbf{W}_{-1}^{(t)}, \mathbf{x}_i), F_{+1}(\mathbf{W}_{+1}^{(t)}, \mathbf{x}_i) = O(1)$ for all $i$ with $y_i = 1$ with the conditions that $\max_r \Gamma_{1,r}^{(t)}, \max_{j,i,r} |\Phi_{j,r,i}^{(t)}|$ are bounded. Moreover, we know that $|\ell_i'^{(t)}| = \frac{1}{1+\exp\{y_i \cdot [F_{+1}(\mathbf{W}_{+1}^{(t)}, \mathbf{x}_i) - F_{-1}(\mathbf{W}_{-1}^{(t)}, \mathbf{x}_i)]\}}$, which implies the existence of a positive constant $C_1$ such that $-\ell_i'^{(t)} \geq C_1$ for all $i$ with $y_i = 1$.

Thus, with the update of $\Gamma$ in Definition F.1, we have

$$\Gamma_{1,r}^{(t+1)} = \Gamma_{1,r}^{(t)} - \frac{\eta}{nm} \cdot \sum_{i=1}^n \ell_i'^{(t)} \cdot \sigma'(y_i \cdot \langle \mathbf{w}_{1,r}^{(0)}, \mathbf{v}\rangle + y_i \cdot \Gamma_{1,r}^{(t)} - \eta \sum_{s=1}^t \langle \mathbf{z}_s, y_i \mathbf{v}\rangle) \cdot \|\mathbf{v}\|_2^2$$

$$\geq \Gamma_{1,r}^{(t)} + \frac{C_1 \eta}{nm} \cdot \sum_{w=1} \sigma'(\langle \mathbf{w}_{1,r}^{(0)}, \mathbf{v}\rangle + \Gamma_{1,r}^{(t)} - \eta \sum_{s=1}^t \langle \mathbf{z}_s, \mathbf{v}\rangle) \cdot \|\mathbf{v}\|_2^2.$$

It is noticed that $T_1^+ \leq \frac{\sigma_0 mn\varepsilon}{4\eta(\|\mathbf{v}\|_2 + \|\boldsymbol{\xi}\|_2)}$ and $\max_r \langle \mathbf{w}_{1,r}^{(0)}, \mathbf{v}\rangle \geq \sigma_0 \|\mathbf{v}\|_2/2$ due to Lemma E.3, then it holds that $\max_r \langle \mathbf{w}_{1,r}^{(0)}, \mathbf{v}\rangle - \eta \sum_{s=1}^t \langle \mathbf{z}_s, \mathbf{v}\rangle \geq \sigma_0 \|\mathbf{v}\|_2/4$. Denote $\widehat{\Gamma}_{1,r}^{(t)} = \Gamma_{1,r}^{(t)} + \langle \mathbf{w}_{1,r}^{(0)}, \mathbf{v}\rangle - \eta \sum_{s=1}^t \langle \mathbf{z}_s, \mathbf{v}\rangle$ and let $A^{(t)} = \max \widehat{\Gamma}_{1,r}^{(t)}$,

then it follows that

$$A^{(t+1)} \geq A^{(t)} + \frac{C_1\eta}{nm} \cdot \sum_{y_i=1} \sigma'(A^{(t)}) \cdot \|\mathbf{v}\|_2^2 - \eta\langle\mathbf{z}_{t+1},\mathbf{v}\rangle$$

$$\overset{(i)}{\geq} A^{(t)} + \frac{C_1\eta q\|\mathbf{v}\|_2^2}{4m}[A^{(t)}]^{q-1} - \eta\langle\mathbf{z}_{t+1},\mathbf{v}\rangle$$

$$\overset{(ii)}{\geq} (1 + \frac{C_1\eta q\|\mathbf{v}\|_2^2}{4m}[A^{(0)}]^{q-2})A^{(t)} - \eta\langle\mathbf{z}_{t+1},\mathbf{v}\rangle$$

$$\overset{(iii)}{\geq} (1 + \frac{C_1\eta q\sigma_0^{q-2}\|\mathbf{v}\|_2^q}{4^{q-1}m})A^{(t)} - \eta\langle\mathbf{z}_{t+1},\mathbf{v}\rangle$$

Let $\quad q_\Gamma = (1 + \frac{C_1\eta q\sigma_0^{q-2}\|\mathbf{v}\|_2^q}{4^{q-1}m})$, then $\quad \overset{(iv)}{=} q_\Gamma^t A^{(0)} - (\frac{q_\Gamma^t - 1}{q_\Gamma - 1})\eta\langle\mathbf{z}_{t+1},\mathbf{v}\rangle.$

Here, $(i)$ holds due to the lower bound on the number of positive data in Lemma E.1; $(ii)$ and $(iii)$ follow from that the facts $A^{(t)}$ is increasing and $\max_r\langle\mathbf{w}_{1,r}^{(0)},\mathbf{v}\rangle - \eta\sum_{s=1}^t\langle\mathbf{z}_s,\mathbf{v}\rangle \geq \sigma_0\|\mathbf{v}\|_2/4$; $(iv)$ is the summation of geometric series. In addition, we know that $1 + z \geq \exp(z/2)$ for $z \leq 2$ and $1 + z \leq \exp(z)$ for $z \geq 0$, then the following inequality holds

$$A^{(t)} \geq (1 + \frac{C_1\eta q\sigma_0^{q-2}\|\mathbf{v}\|_2^q}{4^{q-1}m})^t A^{(0)} - \frac{4^{q-1}m}{C_1 q\sigma_0^{q-2}\|\mathbf{v}\|_2^q}(q_\Gamma^t - 1) \cdot \frac{\|\mathbf{v}\|_2(\|\mathbf{v}\|_2 + \|\boldsymbol{\xi}\|_2)}{mn\varepsilon}$$

$$\geq \exp(\frac{C_1\eta q\sigma_0^{q-2}\|\mathbf{v}\|_2^q}{4^{q-1}*2m}t)\frac{\sigma_0\|\mathbf{v}\|_2}{2} - \frac{4^{q-1}m}{C_1 q\sigma_0^{q-2}\|\mathbf{v}\|_2^q} \cdot \exp(\frac{C_1\eta q\sigma_0^{q-2}\|\mathbf{v}\|_2^q}{4^{q-1}m}t) \cdot \frac{\|\mathbf{v}\|_2(\|\mathbf{v}\|_2 + \|\boldsymbol{\xi}\|_2)}{mn\varepsilon}$$

$$= (\frac{e^{1/2}\sigma_0\|\mathbf{v}\|_2}{2} - \frac{4^{q-1}(\|\mathbf{v}\|_2 + \|\boldsymbol{\xi}\|_2)}{C_1 q\sigma_0^{q-2}\|\mathbf{v}\|_2^{q-1}n\varepsilon}) \cdot \exp(\frac{C_1\eta q\sigma_0^{q-2}\|\mathbf{v}\|_2^q}{4^{q-1}m}t)$$

$$\geq \frac{e^{1/2}\sigma_0\|\mathbf{v}\|_2}{4} \cdot \exp(\frac{C_1\eta q\sigma_0^{q-2}\|\mathbf{v}\|_2^q}{4^{q-1}m}t),$$

where the last inequality holds due to the choice of $\sigma_0 \geq O(\|\mathbf{v}\|_2^{-1}(n\varepsilon)^{-1/q-1})$. Therefore, it is clear that $A^{(t)}$ will reach 4 within $T_1 = \frac{\log(16/\sigma_0\|\mathbf{v}\|_2)4^{q-1}m}{C_1\eta q\sigma_0^{q-2}\|\mathbf{v}\|_2^q}$ iterations, which indicates that $\max_r\Gamma_{1,r}^{(t)}$ will reach 2 within $T_1$ iterations. Moreover, we can verify that

$$T_1 = \frac{\log(16/\sigma_0\|\mathbf{v}\|_2)4^{q-1}m}{C_1\eta q\sigma_0^{q-2}\|\mathbf{v}\|_2^q} \leq \eta^{-1}\sigma_0 mn\varepsilon(\|\mathbf{v}\|_2 + \|\boldsymbol{\xi}\|_2)^{-1} \leq T_1^+,$$

The inequality follows from the SNR condition in Equation (18) and the choice of $\sigma_0$. Moreover, since we also have $\sigma_0 \leq O(\|\boldsymbol{\xi}\|_2^{-1}\varepsilon^{-1/q})$, it holds that

$$T_1 = \frac{\log(16/\sigma_0\|\mathbf{v}\|_2)4^{q-1}m}{C_1\eta q\sigma_0^{q-2}\|\mathbf{v}\|_2^q} \leq \frac{nmn\eta^{-1}\sigma_0^{2-q}\sigma_\xi^{-q}d^{-q/2}}{2^{q+4}q[4\log(8mn/\delta)]^{(q-1)/2}} \leq T_1^+.$$

Hence, by the definition of $T_{1,1}$, $T_{1,1} \leq T_1 \leq T_1^+/2$ holds. A similar proof applies for $j = -1$, where we can prove that $\max_r\Gamma_{-1,r}^{(T_{1,-1})} \geq 2$ with $T_{1,-1} \leq T_1 \leq T_1^+/2$, thereby completing the proof. □

### G.2. Second Stage

It is clear that we have the following results at the end of the first stage:

$$\mathbf{w}_{j,r}^{(T_1)} = \mathbf{w}_{j,r}^{(0)} + j \cdot \Gamma_{j,r}^{(T_1)} \cdot \frac{\mathbf{v}}{\|\mathbf{v}\|_2^2} + \sum_{i=1}^n \bar{\Phi}_{j,r,i}^{(T_1)} \cdot \frac{\boldsymbol{\xi}_i}{\|\boldsymbol{\xi}_i\|_2^2} + \sum_{i=1}^n \underline{\Phi}_{j,r,i}^{(T_1)} \cdot \frac{\boldsymbol{\xi}_i}{\|\boldsymbol{\xi}_i\|_2^2} - \eta\sum_{s=1}^{T_1}\mathbf{z}_s$$

Meanwhile, at the beginning of the second stage, we have the following results:

- $\max_r\Gamma_{j,r}^{(T_1)} \geq 2, \forall j \in \{\pm 1\}.$

- $\max_{j,r,i} |\Phi_{j,r,i}^{(T_1)}| \le \widehat{\beta}$ where $\widehat{\beta} = \sigma_0 \sigma_\xi \sqrt{d}/2$.

Based on Lemma 4.2 and Lemma 4.4, we conclude that signal learning does not deteriorate over time. Specifically, for any $T_1 \le t \le T_p^*$, it holds that $\Gamma_{j,r}^{(t+1)} \ge \Gamma_{j,r}^{(t)}$, which implies $\max_r \Gamma_{j,r}^{(t)} \ge 2$. If we consider $\mathbf{w}_{j,r}^* = \mathbf{w}_{j,r}^{(0)} + 2qm\log(2q/\kappa) \cdot j \cdot \frac{\mathbf{v}}{\|\mathbf{v}\|_2^2}$, then we can derived:

**Lemma G.2.** *Under the same conditions as signal learning, we have that* $\|\mathbf{W}^{(T_1)} - \mathbf{W}^*\|_F \le \widetilde{O}(m^{3/2}\|\mathbf{v}\|_2^{-1})$.

**Proof of Lemma G.2.** According to triangle inequality, we have

$$
\begin{aligned}
\|\mathbf{W}^{(T_1)} - \mathbf{W}^*\|_F &\le \|\mathbf{W}^{(T_1)} - \mathbf{W}^{(0)}\|_F + \|\mathbf{W}^{(0)} - \mathbf{W}^*\|_F \\
&\overset{(i)}{\le} \sum_{j,r} \frac{\Gamma_{j,r}^{(T_1)}}{\|\mathbf{v}\|_2} + \sum_{j,r,i} \frac{|\bar{\Phi}_{j,r,r}^{(T_1)}|}{\|\boldsymbol{\xi}_i\|_2} + \sum_{j,r,i} \frac{|\underline{\Phi}_{j,r,i}^{(T_1)}|}{\|\boldsymbol{\xi}_i\|_2} + \sum_{j,r} |\eta \sum_{s=1}^{T_1} \mathbf{z}_s| + O(m^{3/2}\log(1/\kappa))\|\mathbf{v}\|_2^{-1} \\
&\overset{(ii)}{\le} \widetilde{O}(m\|\mathbf{v}\|^{-1}) + O(nm\sigma_0) + O(m\sigma_0) + O(m^{3/2}\log(1/\kappa))\|\mathbf{v}\|_2^{-1} \\
&\overset{(iii)}{\le} \widetilde{O}(m^{3/2}\|\mathbf{v}\|_2^{-1}) + O(nm\sigma_0) \\
&\overset{(iv)}{\le} \widetilde{O}(m^{3/2}\|\mathbf{v}\|_2^{-1}).
\end{aligned}
$$

Here, $(i)$ holds due to the decomposition of $\mathbf{w}$ in Definition 4.1 and the definition of $\mathbf{W}^*$; $(ii)$ follows from Proposition F.3, Lemma G.1 and Lemma F.8; $(iii)$ comes from the conditions of $\sigma_0$ in signal learning; $(iv)$ holds due to the choice of $\sigma_0$. $\qquad\square$

**Lemma G.3.** *Under the same conditions as signal learning, we have that* $y_i \langle \nabla f(\mathbf{W}^{(t)}, \mathbf{x}_i), \mathbf{W}^* \rangle \ge q\log(2q/\kappa)$ *for all* $i \in [n]$ *and* $T_1 \le t \le T^*$.

**Proof of Lemma G.3.** We know that

$$
f(\mathbf{W}^{(t)}, \mathbf{x}_i) = (1/m) \sum_{j,r} j \cdot [\sigma(\langle \mathbf{w}_{j,r}, y_i \cdot \mathbf{v} \rangle) + \sigma(\langle \mathbf{w}_{j,r}, \boldsymbol{\xi}_i \rangle)].
$$

Therefore, it holds that

$$
\begin{aligned}
y_i \langle \nabla f(\mathbf{W}^{(t)}, \mathbf{x}_i), \mathbf{W}^* \rangle =& \frac{1}{m} \sum_{j,r} \sigma'(\langle \mathbf{w}_{j,r}^{(t)}, y_i\mathbf{v} \rangle)\langle \mathbf{v}, j\mathbf{w}_{j,r}^* \rangle + \frac{1}{m}\sum_{j,r} \sigma'(\langle \mathbf{w}_{j,r}^{(t)}, \boldsymbol{\xi}_i \rangle)\langle y_i\boldsymbol{\xi}_i, j\mathbf{w}_{j,r}^* \rangle \\
\overset{(i)}{=}& \frac{1}{m}\sum_{j,r}\sigma'(\langle \mathbf{w}_{j,r}^{(t)}, y_i\mathbf{v}\rangle)2qm\log(2q/\kappa) + \frac{1}{m}\sum_{j,r}\sigma'(\langle \mathbf{w}_{j,r}^{(t)}, y_i\mathbf{v}\rangle)\langle \mathbf{v}, j\mathbf{w}_{j,r}^{(0)}\rangle \\
&+ \frac{1}{m}\sum_{j,r}\sigma'(\langle \mathbf{w}_{j,r}^{(t)}, \boldsymbol{\xi}_i\rangle)\langle y_i\boldsymbol{\xi}_i, j\mathbf{w}_{j,r}^{(0)}\rangle \\
\overset{(ii)}{\ge}& \frac{1}{m}\sum_{j,r}\sigma'(\langle \mathbf{w}_{j,r}^{(t)}, y_i\mathbf{v}\rangle)2qm\log(2q/\kappa) - \frac{1}{m}\sum_{j,r}\sigma'(\langle \mathbf{w}_{j,r}^{(t)}, y_i\mathbf{v}\rangle)\widetilde{O}(\sigma_0\|\mathbf{v}\|_2) \\
&- \frac{1}{m}\sum_{j,r}\sigma'(\langle \mathbf{w}_{j,r}^{(t)}, \boldsymbol{\xi}_i\rangle)\widetilde{O}(\sigma_0\sigma_\xi\sqrt{d}),
\end{aligned}
$$

where $(i)$ holds due to the definition of $\mathbf{w}^*$ and $(ii)$ follows from Lemma E.3. Moreover, according to Lemma F.4, we have that for $j = y_i$:

$$
\max_r\{\langle \mathbf{w}_{j,r}^{(t)}, y_i\mathbf{v}\rangle\} = \max_r\{\Gamma_{j,r}^{(t)} + \langle \mathbf{w}_{j,r}^{(0)}, y_i\mathbf{v}\rangle - \eta\sum_{s=1}^{t}\langle \mathbf{z}_s, y_i\mathbf{v}\rangle\} \overset{(i)}{\ge} 2 - \widetilde{O}(\sigma_0\|\mathbf{v}\|_2) \overset{(ii)}{\ge} 1.
$$

Here, $(i)$ holds due to the analysis in Lemma G.1 and $(ii)$ comes from $\sigma_0 \le \widetilde{O}(n^{-1/2}\|\mathbf{v}\|_2)$. Additionally, we can also have

$$|\langle \mathbf{w}_{j,r}^{(t)}, \mathbf{v}\rangle| \overset{(i)}{\le} |\langle \mathbf{w}_{j,r}^{(0)}, \mathbf{v}\rangle| + |\Gamma_{j,r}^{(t)}| + |\eta \sum_{s=1}^{t} \langle \mathbf{z}_s, \mathbf{v}\rangle | \overset{(ii)}{\le} \widetilde{O}(1)$$

$$|\langle \mathbf{w}_{j,r}^{(t)}, \boldsymbol{\xi}_i\rangle| \overset{(iii)}{\le} |\langle \mathbf{w}_{j,r}^{(0)}, \boldsymbol{\xi}_i\rangle| + |\underline{\Phi}_{j,r,i}^{(t)}| + |\bar{\Phi}_{j,r,i}^{(t)}| + 8n\sqrt{\frac{\log(4n^2/\delta)}{d}}\alpha + |\eta \sum_{s=1}^{t} \langle \mathbf{z}_s, \boldsymbol{\xi}_i\rangle | \overset{(iv)}{\le} \widetilde{O}(1).$$

Here, $(i)$ holds due to Lemma F.4 and Lemma F.8; $(ii)$ is given by Lemma F.5; $(ii)$ and $(iv)$ follows from Proposition F.3 and Lemma F.8. Combining these results, we return to $y_i\langle \nabla f(\mathbf{W}^{(t)}, \mathbf{x}_i), \mathbf{W}^*\rangle$, which gives:

$$y_i\langle \nabla f(\mathbf{W}^{(t)}, \mathbf{x}_i), \mathbf{W}^*\rangle \ge 2q\log(2q/\kappa) - \widetilde{O}(\sigma_0\|\mathbf{v}\|_2) - \widetilde{O}(\sigma_0\sigma_\xi\sqrt{d}) \ge q\log(2q/\kappa),$$

where the last inequality is driven by the conditions of $\sigma_0$ as signal learning. $\qquad\square$

**Lemma G.4.** *Under the same conditions as signal learning, it holds that*

$$\|\mathbf{W}^{(t)} - \mathbf{W}^*\|_F^2 - \|\mathbf{W}^{(t+1)} - \mathbf{W}^*\|_F^2 \ge (2q-1)\eta L_D(\mathbf{W}^{(t)}) - \eta\kappa - \eta^2\widetilde{O}(d\sigma_z^2) - \eta\widetilde{O}(\sigma_z m^{3/2}\|\mathbf{v}\|_2^{-1})$$

*for all $T_1 \le t \le T^*$.*

**Proof of Lemma G.4.** According to the optimization properties, we know the first equality holds:

$$\begin{aligned}
&\|\mathbf{W}^{(t)} - \mathbf{W}^*\|_F^2 - \|\mathbf{W}^{(t+1)} - \mathbf{W}^*\|_F^2 \\
&= 2\eta\langle \nabla L_S(\mathbf{W}^{(t)}), \mathbf{W}^{(t)} - \mathbf{W}^*\rangle - \eta^2\|\nabla L_S(\mathbf{W}^{(t)})\|_F^2 \\
&\overset{(i)}{=} \frac{2\eta}{n} \sum_{i=1}^{n} \ell_i'^{(t)}[qy_if(\mathbf{W}^{(t)}, \mathbf{x}_i) - \langle \nabla f(\mathbf{W}^{(t)}, \mathbf{x}_i), \mathbf{W}^*\rangle] + \eta\langle \mathbf{z}_t, \mathbf{W}^{(t)} - \mathbf{W}^*\rangle \\
&\quad - \eta^2(O(\max\{\|\mathbf{v}\|_2^2, \sigma_\xi^2 d\})L_D(\mathbf{W}^{(t)}) + O(\sigma_z^2 d\log(1/\delta))) \\
&\overset{(ii)}{\ge} \frac{2\eta}{n} \sum_{i=1}^{n} \ell_i'^{(t)}[qy_if(\mathbf{W}^{(t)}, \mathbf{x}_i) - q\log(2q/\kappa)] \\
&\quad + \eta\langle \mathbf{z}_t, \mathbf{W}^{(t)} - \mathbf{W}^*\rangle - \eta^2(O(\max\{\|\mathbf{v}\|_2^2, \sigma_\xi^2 d\})L_D(\mathbf{W}^{(t)}) + O(\sigma_z^2 d\log(1/\delta))) \\
&\overset{(iii)}{\ge} \frac{2q\eta}{n} \sum_{i=1}^{n} [\ell(y_if(\mathbf{W}^{(t)}, \mathbf{x}_i)) - \kappa/(2q)] \\
&\quad - \eta\widetilde{O}(\sigma_z m^{3/2}\|\mathbf{v}\|_2^{-1}) - \eta^2(O(\max\{\|\mathbf{v}\|_2^2, \sigma_\xi^2 d\})L_D(\mathbf{W}^{(t)}) + O(\sigma_z^2 d\log(1/\delta))) \\
&\overset{(iv)}{\ge} (2q-1)\eta L_D(\mathbf{W}^{(t)}) - \eta\kappa - \eta^2\widetilde{O}(d\sigma_z^2) - \eta\widetilde{O}(\sigma_z m^{3/2}\|\mathbf{v}\|_2^{-1}).
\end{aligned} \qquad (19)$$

Here, $(i)$ holds due to the definition of noisy gradient, the neural network is $q$ homogeneous, and Lemma F.9; $(ii)$ is driven from Lemma G.3; $(iii)$ is due to the convexity of the cross entropy function; $(iv)$ comes from definition of $L_D$. $\qquad\square$

**Lemma G.5** (Restatement of Corollary 4.7). *Let $T, T_1$ be defined in respectively. Then under the same conditions as signal learning, for any $t \in [T_1, T]$, it holds that $|\Gamma_{j,r}^{(t)}| \le \sigma_0\|\mathbf{v}\|_2$ for all $j \in \{\pm 1\}$ and $r \in [m]$. Moreover, let $\mathbf{W}^*$ be the collection of CNN parameters with convolution filters $\mathbf{w}_{j,r}^* = \mathbf{w}_{j,r}^{(0)} + 2qm\log(2q/\kappa) \cdot j \cdot \|\mathbf{v}\|_2^{-2} \cdot \mathbf{v}$. Then the following bound holds*

$$\frac{1}{t - T_1 + 1}\sum_{s=T_1}^{t} L_D(\mathbf{W}^{(s)}) \le \frac{\|\mathbf{W}^{(T_1)} - \mathbf{W}^*\|_F^2}{(2q-1)\eta(t - T_1 + 1)} + \frac{\kappa}{(2q-1)} + \underbrace{\frac{\eta d\sigma_z^2 + \widetilde{O}(\sigma_z m^{3/2}\|\mathbf{v}\|_2^{-1})}{(2q-1)}}_{Private\ terms}$$

*for all $t \in [T_1, T]$, where we denote $\|\mathbf{W}\|_F = \sqrt{\|\mathbf{W}_{+1}\|_F^2 + \|\mathbf{W}_{-1}\|_F^2}$.*

**Proof of Lemma G.5.** According to Lemma G.4, we have, for any $t \leq T$:

$$\|\mathbf{W}^{(t)} - \mathbf{W}^*\|_F^2 - \|\mathbf{W}^{(t+1)} - \mathbf{W}^*\|_F^2 \geq (2q-1)\eta L_D(\mathbf{W}^{(t)}) - \eta\kappa - \eta^2\widetilde{O}(d\sigma_z^2) - \eta\widetilde{O}(\sigma_z m^{3/2}\|\mathbf{v}\|_2^{-1}).$$

By summing over all terms and dividing $t - T_1 - 1$ on both sides, we obtain:

$$\frac{1}{t - T_1 + 1}\sum_{s=T_1}^{t} L_D(\mathbf{W}^{(s)}) \leq \frac{\|\mathbf{W}^{(T_1)} - \mathbf{W}^*\|_F^2}{(2q-1)\eta(t - T_1 + 1)} + \frac{\kappa}{(2q-1)} + \frac{\eta d\sigma_z^2 + \widetilde{O}(\sigma_z m^{3/2}\|\mathbf{v}\|_2^{-1})}{(2q-1)}.$$

If we have $T = T_1 + \lfloor\frac{\|\mathbf{W}^{(T_1)} - \mathbf{W}^*\|_E^2}{2\eta\kappa}\rfloor = \frac{Cmn\varepsilon}{\eta\mu(\|\mathbf{v}\|_2 + \|\boldsymbol{\xi}\|_2)} \geq \kappa^{-1}$, then it holds that

$$\frac{\|\mathbf{W}^{(T_1)} - \mathbf{W}^*\|_F^2}{(2q-1)\eta(T - T_1 + 1)} + \frac{\kappa}{2q-1} \leq \frac{3\kappa}{2q-1},$$

and with $\sigma_z = \frac{1}{\eta\mu\sqrt{T}}$:

$$\frac{\eta d\sigma_z^2 + \widetilde{O}(\sigma_z m^{3/2}\|\mathbf{v}\|_2^{-1})}{(2q-1)} \leq \frac{d}{\eta\mu^2 T(2q-1)} + \frac{m^{3/2}\|\mathbf{v}\|_2^{-1}}{\eta\mu\sqrt{T}(2q-1)} \overset{(i)}{\leq} \frac{\kappa}{(2q-1)},$$

where $(i)$ comes from the assumption of $\eta$. Therefore, combining above results, we conclude that

$$\frac{1}{t - T_1 + 1}\sum_{s=T_1}^{t} L_D(\mathbf{W}^{(s)}) \leq \kappa.$$

Next, we will use induction to prove that $\Psi_t = \max_{i,j,t}|\Phi_{i,j,r}^t| \leq 2\sigma_0\|\boldsymbol{\xi}\|_2$ holds for all $t \in [T_1, T]$. According to Lemma G.1, we know it holds for $T_1$. Now, assume it holds for some $t \in [T_1, T)$, and we will show that it also holds for $t+1$.

$$\Psi^{(t+1)} \overset{(i)}{\leq} \Psi^{(t)} + \max_{j,r,i}\left\{\frac{\eta}{nm}\cdot|\ell_i^{(t)}|\cdot\sigma'(\langle\mathbf{w}_{j,r}^{(0)}, \boldsymbol{\xi}_i\rangle) + 2\sum_{i'=1}^{n}\Psi^{(t)}\cdot\frac{|\langle\boldsymbol{\xi}_{i'}, \boldsymbol{\xi}_i\rangle|}{\|\boldsymbol{\xi}_{i'}\|_2^2} - \eta\sum_{s=1}^{t}\langle\mathbf{z}_s, \boldsymbol{\xi}_i\rangle)\cdot\|\boldsymbol{\xi}_{i'}\|_2^2\right\}$$

$$\overset{(i)}{=} \Psi^{(t)} + \max_{j,r,i}\left\{\frac{\eta}{nm}\cdot|\ell_i'^{(t)}|\cdot\sigma'(\langle\mathbf{w}_{j,r}^{(0)}, \boldsymbol{\xi}_i\rangle) + 2\Psi^{(t)} + 2\sum_{i'\neq i}^{n}\Psi^{(t)}\cdot\frac{|\langle\boldsymbol{\xi}_{i'}, \boldsymbol{\xi}_i\rangle|}{\|\boldsymbol{\xi}_{i'}\|_2^2} - \eta\sum_{s=1}^{t}\langle\mathbf{z}_s, \boldsymbol{\xi}_i\rangle)\cdot\|\boldsymbol{\xi}_{i'}\|_2^2,\right.$$

$$\overset{(ii)}{\leq} \Psi^{(t)} + \frac{\eta q}{nm}\cdot\max_i|\ell_i^{(t)}|\cdot[4\cdot\sqrt{\log(8mn/\delta)}\cdot\sigma_0\sigma_\xi\sqrt{d} + (2 + \frac{4n\sigma_\xi^2\cdot\sqrt{d\log(4n^2/\delta)}}{\sigma_\xi^2 d/2})\cdot\Psi^{(t)}]^{q-1}\cdot 2\sigma_\xi^2 d$$

$$\overset{(iii)}{\leq} \Psi^{(t)} + \frac{\eta q}{nm}\cdot\max_i|\ell_i^{(t)}|\cdot(4\cdot\sqrt{\log(8mn/\delta)}\cdot\sigma_0\sigma_\xi\sqrt{d} + 4\cdot\Psi^{(t)})^{q-1}\cdot 2\sigma_\xi^2 d.$$

Here, $(i)$ holds due to Definition F.1; $(ii)$ comes from Lemma E.3, Lemma E.2 and the choice of $T$. $(iii)$ is due to the condition of $d$. By summing the above over $t$, we have:

$$\Psi^{(t)} \overset{(i)}{\leq} \Psi^{(T_1)} + \frac{\eta q}{nm}\sum_{s=T_1}^{t-1}\max_i|\ell_i^{(s)}|\widetilde{O}(\sigma_\xi^2 d)(\sigma_0\|\boldsymbol{\xi}\|_2)^{q-1}$$

$$\overset{(ii)}{\leq} \Psi^{(T_1)} + \frac{\eta q}{nm}\widetilde{O}(\sigma_\xi^2 d)(\sigma_0\|\boldsymbol{\xi}\|_2)^{q-1}\sum_{s=T_1}^{t-1}\max_i\ell_i^{(s)}$$

$$\overset{(iii)}{\leq} \Psi^{(T_1)} + \widetilde{O}(\eta m^{-1}\sigma_\xi^2 d)(\sigma_0\|\boldsymbol{\xi}\|_2)^{q-1}\sum_{s=T_1}^{t-1}L_S(\mathbf{W}^{(s)})$$

$$\overset{(iv)}{\leq} \Psi^{(T_1)} + \widetilde{O}(m^2\,\mathrm{SNR}^{-2})(\sigma_0\|\boldsymbol{\xi}\|_2)^{q-1}$$

$$\overset{(v)}{\leq} (\sigma_0\|\boldsymbol{\xi}\|_2) + \widetilde{O}(m^2(n\varepsilon)^2(n\varepsilon)^{-q-3-2/q})(\sigma_0\|\boldsymbol{\xi}\|_2)$$

$$\overset{(vi)}{\leq} 2(\sigma_0\|\boldsymbol{\xi}\|_2).$$

Here, $(i)$ holds due to induction hypothesis; $(ii)$ is by $|\ell'| \leq \ell$, $(iii)$ comes from $\max_i \ell_i^{(s)} \leq \sum_i \ell_i^{(s)} = nL_D(\mathbf{W}^{(s)})$; $(iv)$ is due to $\sum_{s=T_1}^{t-1} L_D(\mathbf{W}^{(s)}) \leq \sum_{s=T_1}^{T} L_D(\mathbf{W}^{(s)}) = \widetilde{O}(\eta^{-1}m^3\|\mathbf{v}\|_2^2)$ from the choice of $T$; $(v)$ is by the conditions of $\sigma_0$ and SNR; $(vi)$ is due to $(n\varepsilon)^{q+1} \geq m$. $\qquad\square$

Now we consider the generalization performance of the privately trained model. Given a new data point $(\mathbf{x}, y)$ drawn from the distribution defined in Definition 3.1, we assume $\mathbf{x} = [y\mathbf{v}, \boldsymbol{\xi}]$ without loss of generality.

**Lemma G.6.** *Under the same conditions as signal learning, we have that* $\max_{j,r} |\langle \mathbf{w}_{j,r}^{(t)}, \boldsymbol{\xi}_i \rangle| \leq 1/2$ *for all* $0 \leq t \leq T$.

**Proof of Lemma G.6.** According to the signal-noise decomposition, the private model satisfies:

$$\mathbf{w}_{j,r}^{(t)} = \mathbf{w}_{j,r}^{(0)} + j \cdot \Gamma_{j,r}^{(t)} \cdot \|\mathbf{v}\|_2^{-2} \cdot \mathbf{v} + \sum_{i=1}^n \bar{\Phi}_{j,r,i}^{(t)} \cdot \|\boldsymbol{\xi}_i\|_2^{-2} \cdot \boldsymbol{\xi}_i + \sum_{i=1}^n \underline{\Phi}_{j,r,i}^{(t)} \cdot \|\boldsymbol{\xi}_i\|_2^{-2} \cdot \boldsymbol{\xi}_i - \eta \sum_{s=1}^t \mathbf{z}_s.$$

Then, we have

$$|\langle \mathbf{w}_{j,r}^{(t)}, \boldsymbol{\xi}_i \rangle| \overset{(i)}{\leq} |\langle \mathbf{w}_{j,r}^{(0)}, \boldsymbol{\xi}_i \rangle| + |\underline{\Phi}_{j,r,i}^{(t)}| + |\bar{\Phi}_{j,r,i}^{(t)}| + 8n\sqrt{\frac{\log(4n^2/\delta)}{d}}\alpha + 0.1$$

$$\overset{(ii)}{\leq} 2\sqrt{\log(8mn/\delta)} \cdot \sigma_0\sigma_\xi\sqrt{d} + \sigma_0\sigma_\xi\sqrt{d} + 8n\sqrt{\frac{\log(4n^2/\delta)}{d}}\alpha + 0.1$$

$$\overset{(iii)}{\leq} 1/2.$$

Here, $(i)$ holds due to Lemma F.8 and the assumption of training iterations; $(ii)$ comes from Lemma H.6; $(iii)$ is driven from the condition of $\sigma_0$ and the assumptions of $n\varepsilon, d$. $\qquad\square$

**Lemma G.7.** *Under the same assumptions as Theorem 4.3, with probability at least* $1 - 4mT\exp(-C_1^{-1}\sigma_0^{-2}\sigma_\xi^{-2}d^{-1})$, *we have* $\max_{j,r} |\langle \mathbf{w}_{j,r}^{(t)}, \boldsymbol{\xi} \rangle| \leq 1/2$ *for all* $0 \leq t \leq T$, *where* $C_1 = \widetilde{O}(1)$.

**Proof of Lemma G.7.** Define $\widetilde{\mathbf{w}}_{j,r}^{(t)} = \mathbf{w}_{j,r}^{(t)} - j \cdot \Gamma_{j,r}^{(t)} \cdot \frac{\mathbf{v}}{\|\mathbf{v}\|_2^2}$. It follows that $\langle \widetilde{\mathbf{w}}_{j,r}^{(t)}, \boldsymbol{\xi} \rangle = \langle \mathbf{w}_{j,r}^{(t)}, \boldsymbol{\xi} \rangle$. Additionally, we have:

$$\|\widetilde{\mathbf{w}}_{j,r}^{(t)}\|_2 \leq \widetilde{O}(\sigma_0\sqrt{d} + n\sigma_0 + \sigma_0\sigma_\xi\sqrt{d}) = \widetilde{O}(\sigma_0\sqrt{d}),$$

where the equality holds due to the condition $d \geq \widetilde{\Omega}(m^2n^4)$ and the analysis in Theorem 4.6. Thus, we know that $\max_{j,r} \|\widetilde{\mathbf{w}}_{j,r}^{(t)}\|_2 \leq C_1\sigma_0\sqrt{d}$, where $C_1 = \widetilde{O}(1)$. Since $\langle \widetilde{\mathbf{w}}_{j,r}^{(t)}, \boldsymbol{\xi} \rangle$ follows a Gaussian distribution with mean zero and standard deviation bounded by $C_1\sigma_0\sigma_\xi\sqrt{d}$, the probability of deviation can be bounded as:

$$\mathbb{P}(|\langle \widetilde{\mathbf{w}}_{j,r}^{(t)}, \boldsymbol{\xi} \rangle| \geq 1/2) \leq 2\exp(-\frac{1}{8C_1^2\sigma_0^2\sigma_\xi^2 d})$$

By applying a union bound over all indices $j, r$, and $t$, the proof is complete. $\qquad\square$

**Lemma G.8** (Restatement of Corollary 4.8). *Under the same conditions as above, for any* $t \leq T$ *with* $L_D(\mathbf{W}^{(t)}) \leq \kappa$, *with at least probability* $1 - 1/d$, *it holds that* $L_{\mathcal{D}}(\mathbf{W}^{(t)}) \leq 6\kappa + \exp(\varepsilon^{-2/q})$.

**Proof of Corollary 4.8.** Let $\mathcal{K}$ represent the event where Lemma G.7 holds. We can partition $L_{\mathcal{D}}(\mathbf{W}^{(t)})$ into two components:

$$\mathbb{E}[\ell(yf(\mathbf{W}^{(t)}, \mathbf{x}))] = \underbrace{\mathbb{E}[\mathbf{1}(\mathcal{K})\ell(yf(\mathbf{W}^{(t)}, \mathbf{x}))]}_{I_1} + \underbrace{\mathbb{E}[\mathbf{1}(\mathcal{K}^c)\ell(yf(\mathbf{W}^{(t)}, \mathbf{x}))]}_{I_2}$$

We will now bound $I_1$ and $I_2$ separately. The term $I_1$ can be bounded by $6L_D(\mathbf{W}^t) \leq \kappa$ according to Lemma D.8 in (Cao et al., 2022). Next, we bound the second term $I_2$. We select an arbitrary training data point $(\mathbf{x}_{i'}, y_{i'})$ such that $y_{i'} = y$.

Then, we have:

$$\ell(yf(\mathbf{W}^{(t)}, \mathbf{x})) \leq \log(1 + \exp(F_{-y}(\mathbf{W}^{(t)}, \mathbf{x})))$$

$$\overset{(i)}{\leq} 1 + F_{-y}(\mathbf{W}^{(t)}, \mathbf{x})$$

$$\overset{(ii)}{=} 1 + \frac{1}{m} \sum_{j=-y, r \in [m]} \sigma(\langle \mathbf{w}_{j,r}^{(t)}, y\mathbf{v} \rangle) + \frac{1}{m} \sum_{j=-y, r \in [m]} \sigma(\langle \mathbf{w}_{j,r}^{(t)}, \boldsymbol{\xi} \rangle)$$

$$\overset{(iii)}{\leq} 1 + F_{-y_{i'}}(\mathbf{W}_{-y_{i'}}, \mathbf{x}_{i'}) + \frac{1}{m} \sum_{j=-y, r \in [m]} \sigma(\langle \mathbf{w}_{j,r}^{(t)}, \boldsymbol{\xi} \rangle)$$

$$\overset{(v)}{\leq} 2 + \frac{1}{m} \sum_{j=-y, r \in [m]} \sigma(\langle \mathbf{w}_{j,r}^{(t)}, \boldsymbol{\xi} \rangle)$$

$$\overset{(iv)}{\leq} 2 + \widetilde{O}((\sigma_0 \sqrt{d})^q) \|\boldsymbol{\xi}\|^q.$$

Here, $(i)$ is due to $F_y(\mathbf{W}^{(t)}, \mathbf{x}) \geq 0$; $(ii)$ follows from the property of the logarithmic function; $(iii)$ holds due to

$$\frac{1}{m} \sum_{j=-y, r \in [m]} \sigma(\langle \mathbf{w}_{j,r}^{(t)}, y\mathbf{v} \rangle) \leq F_{-y}(\mathbf{W}_{-y}, \mathbf{x}_{i'}) = F_{-y_{i'}}(\mathbf{W}_{-y_{i'}}, \mathbf{x}_{i'});$$

$(v)$ is by Lemma F.6; $(iv)$ comes from Lemma G.7 that $\|\widetilde{\mathbf{w}}_{j,r}^{(t)}\|_2 \leq \widetilde{O}(\sigma_0 \sqrt{d})$. Therefore, we can bound term 2 as follows:

$$I_2 \overset{(i)}{\leq} \sqrt{\mathbb{E}[\mathbb{1}(\mathcal{K}^c)]} \cdot \sqrt{\mathbb{E}[\ell(yf(\mathbf{W}^{(t)}, \mathbf{x}))^2]}$$

$$\overset{(ii)}{\leq} \sqrt{\mathbb{P}(\mathcal{K}^c)} \cdot \sqrt{4 + \widetilde{O}((\sigma_0 \sqrt{d})^{2q}) \mathbb{E}[\|\boldsymbol{\xi}\|_2^{2q}]}$$

$$\overset{(iii)}{\leq} \exp[-\widetilde{\Omega}(\sigma_0^{-2} \sigma_\xi^{-2} d^{-1}) + \text{poly} \log(d)]$$

$$\overset{(v)}{\leq} \exp((n\varepsilon)^{-1-1/q}).$$

Here, $(i)$ holds due to Cauchy-Schwartz inequality; $(ii)$ and $(iii)$ come from the fact $\sqrt{4 + \widetilde{O}((\sigma_0 \sqrt{d})^{2q}) \mathbb{E}[\|\boldsymbol{\xi}\|_2^{2q}]} = O(\text{poly}(d))$ and Lemma G.7; $(v)$ is by the condition of $\sigma_0 \leq (n\varepsilon)^{-1-1/q} \|\xi\|_2^{-1}$. This completes the proof. $\qquad\square$

## H. Noise Memorization

**Lemma H.1.** *Under the same conditions as noise memorization, then it holds that $\bar{\beta} \geq \sigma_0 \sigma_\xi \sqrt{d}/4 \geq 20n\sqrt{\frac{\log(4n^2/\delta)}{d}}\alpha$, if we have $\sigma_0 \geq 80n\sqrt{\frac{\log(4n^2/\delta)}{d}}\alpha \cdot \min\{(\sigma_\xi \sqrt{d})^{-1}, \|\mathbf{v}\|_2^{-1}\}$.*

**Proof of Lemma H.1.** Given the SNR condition in noise memorization $\sigma_\xi^q(\sqrt{d})^q \geq \widetilde{\Omega}(n\|\mathbf{v}\|_2^q)$, it follows that: $\sigma_\xi \sqrt{d} \geq \|\mathbf{v}\|_2$. Thus, we have:

$$\bar{\beta} \geq \frac{\sigma_0 \sigma_\xi \sqrt{d}}{4} = \frac{\sigma_0}{4} \cdot \max\{\sigma_\xi \sqrt{d}, \|\mathbf{v}\|_2\} \geq 20n\sqrt{\frac{\log(4n^2/\delta)}{d}}\alpha.$$

where the first inequality follows from and the last inequality is a result of the lower bound condition on $\sigma_0$ stated in noise memorization.

$\square$

### H.1. First Stage

**Lemma H.2** (Restatement of Theorem 4.10). *Under the same conditions as noise memorization, in particular, if we choose*

$$n^{-1} \text{SNR}^{-q} \geq \frac{C2^{q+2} \log(20/(\sigma_0 \sigma_\xi \sqrt{d}))(\sqrt{2\log(8m/\delta)})^{q-2}}{0.15^{q-2}}, \quad \frac{\varepsilon}{(1+\text{SNR})} \geq \frac{C \log(10/\sigma_0 \sigma_\xi \sqrt{d})}{0.15^{q-2}q} \qquad (20)$$

where $C = O(1)$ is a positive constant, then there exist

$$T_1 = \frac{C \log(10/(\sigma_0 \sigma_\xi \sqrt{d}))4mn}{0.15^{q-2} \eta q \sigma_0^{q-2} (\sigma_\xi^2 \sqrt{d})^q}$$

such that

- $\max_{j,r} \bar{\Phi}_{j,r,i}^{(T_1)} \geq 2$ for all $i \in [n]$.
- $\max_{j,r} \Gamma_{j,r}^{(t)} = \widetilde{O}(\sigma_0 \|\mathbf{v}\|_2)$ for all $0 \leq t \leq T_1$.
- $\max_{j,r,i} |\Phi_{j,r,i}^{(t)}| = \widetilde{O}(\sigma_0 \sigma_\xi \sqrt{d})$ for all $0 \leq t \leq T_1$.

**Proof of Lemma H.2.** First, let $T_1^+ = \min\{\frac{m}{\eta q 2^{q-1}(\sqrt{2 \log(8m/\delta)})^{q-2}\sigma_0^{q-2}\|\mathbf{v}\|_2^q}, \frac{\sigma_0 mn\varepsilon}{\eta(\|\mathbf{v}\|_2 + \|\boldsymbol{\xi}\|_2)}\}$. According to the proof of Proposition F.3, it follows that $\underline{\Phi}_{j,r,i}^{(t)} \geq -\beta - 16n\sqrt{\frac{\log(4n^2/\delta)}{d}}\alpha - 0.2$. Notably, the constant 0.2 here is chosen for simplicity in the proof and can be replaced with any value. For example, if we take $\underline{\Phi}_{j,r,i}^{(t)} \geq -\beta - 16n\sqrt{\frac{\log(4n^2/\delta)}{d}}\alpha - \widetilde{O}(\sigma_0 \sigma_\xi \sqrt{d})$, the proof of Proposition F.3 still holds, provided that $T \leq \frac{\sigma_0 mn\varepsilon}{\eta(\|\mathbf{v}\|_2 + \|\boldsymbol{\xi}\|_2)}$. Moreover, we have $\Phi_{j,r,i}^{(t)} \leq 0$ and $\bar{\beta} \leq \beta = \widetilde{O}(\sigma_0 \sigma_\xi \sqrt{d})$. Therefore, $\max_{j,r,i} |\Phi_{j,r,i}^{(t)}| = \widetilde{O}(\sigma_0 \sigma_\xi \sqrt{d})$.

Now, we proceed to prove the dynamics of $\Gamma_{j,r}^{(t+1)}$. Similar to the signal learning, we define $A^{(t)} = \max_{j,r}\{\Gamma_{j,r}^{(t)} + |\langle \mathbf{w}_{j,r}^{(0)}, \mathbf{v}\rangle| - y_i\eta\sum_{s=1}^{t}\langle \mathbf{z}_s, \mathbf{v}\rangle\}$, then it holds that

$$
\begin{aligned}
\Gamma_{j,r}^{(t+1)} &= \Gamma_{j,r}^{(t)} - \frac{\eta}{nm}\cdot\sum_{i=1}^{n}\ell_i'^{(t)}\cdot\sigma'(\langle \mathbf{w}_{j,r}^{(t)}, y_i\cdot\mathbf{v}\rangle)\|\mathbf{v}\|_2^2 \\
&\leq \Gamma_{j,r}^{(t)} + \frac{\eta}{nm}\cdot\sum_{i=1}^{n}\sigma'(|\langle \mathbf{w}_{j,r}^{(0)}, \mathbf{v}\rangle| + \Gamma_{j,r}^{(t)} - \eta y_i\sum_{s=1}^{t}\langle \mathbf{z}_s, \mathbf{v}\rangle)\|\mathbf{v}\|_2^2 \qquad (21)\\
A^{(t+1)} &\leq A^{(t)} + \frac{\eta q\|\mathbf{v}\|_2^2}{m}[A^{(t)}]^{q-1} + \eta y_i\langle \mathbf{z}_{t+1}, \mathbf{v}\rangle.
\end{aligned}
$$

Next, we will prove $A^{(0)} \leq 3A^{(0)}$ for $t \leq T_1^+$ by induction. First, $A^{(0)} \leq 3A^{(0)}$ holds at $t = 0$ due to the definition and we assume it holds for $t$. Now suppose that there exists some $t \leq T_1^+$ such that $A^{(s)} \leq 2A^{(0)}$ holds $0 \leq s \leq t-1$. Applying a telescoping sum to Equation (21) yields:

$$
\begin{aligned}
A^t &\leq A^{(0)} + \sum_{s=0}^{t}\frac{\eta q\|\mathbf{v}\|_2^2}{m}[A^{(s)}]^{q-1} + \sum_{s=0}^{t}\eta y_i\langle \mathbf{z}_{s+1}, \mathbf{v}\rangle \\
&\overset{(i)}{\leq} A^{(0)} + \frac{\eta q\|\mathbf{v}\|_2^2 T_1^+ 3^{q-1}}{m}[A^{(0)}]^{q-1} + \widetilde{O}(\sigma_0\|\mathbf{v}\|_2) \\
&\overset{(ii)}{\leq} A^{(0)} + \frac{\eta q\|\mathbf{v}\|_2^2 T_1^+ 3^{q-1}}{m}[\sqrt{2\log(8m/\delta)}\cdot\sigma_0\|\mathbf{v}\|_2]^{q-2}A^{(0)} + A^{(0)} \\
&\overset{(ii)}{\leq} 3A^{(0)}.
\end{aligned}
$$

Here, $(i)$ holds due to the induction hypothesis and $T_1^+ \leq \frac{\sigma_0 mn\varepsilon}{\eta(\|\mathbf{v}\|_2 + \|\boldsymbol{\xi}\|_2)}$; $(ii)$ follows Lemma E.3 and $(iii)$ is derived from $T_1^+$. Moreover, we have $\max_{j,r}\Gamma_{j,r}^{(t)} \leq A^{(t)} + \max_{j,r}\{|\langle \mathbf{w}_{j,r}^{(0)}, \mathbf{v}\rangle| + y_i\eta\sum_{s=1}^{t}\langle \mathbf{z}_s, \mathbf{v}\rangle\} \leq 5A^{(0)} = \widetilde{O}(\sigma_0\|\mathbf{v}\|_2)$.

Now, we consider proving that the maximum of noise memorization is larger than 2. For $y_i = j$, according to Lemma F.5, we have:

$$
\begin{aligned}
\langle \mathbf{w}_{j,r}^{(t)}, \boldsymbol{\xi}_i\rangle &\geq \langle \mathbf{w}_{j,r}^{(0)}, \boldsymbol{\xi}_i\rangle + \bar{\Phi}_{j,r,i}^{(t)} - 8n\sqrt{\frac{\log(4n^2/\delta)}{d}}\alpha - \sum_{s=1}^{t}\langle \mathbf{z}_s, \boldsymbol{\xi}_i\rangle \\
&\geq \bar{\Phi}_{j,r,i}^{(t)} + \langle \mathbf{w}_{j,r}^{(0)}, \boldsymbol{\xi}_i\rangle - 0.4\bar{\beta} - \sum_{s=1}^{t}\langle \mathbf{z}_s, \boldsymbol{\xi}_i\rangle.
\end{aligned}
$$

Similar to the proof of signal learning, let $B_i^{(t)} = \max_{j=y_i,r}\{\bar{\Phi}_{j,r,i}^{(t)} + \langle \mathbf{w}_{j,r}^{(0)}, \boldsymbol{\xi}_i \rangle - 0.4\bar{\beta} - \sum_{s=1}^{t}\langle \mathbf{z}_s, \boldsymbol{\xi}_i \rangle\}$. For each $i$, let $T_1^{(i)}$ denote the first time in the period $[0, T_1^+]$ such that $\bar{\Phi}_{j,r,i}^{(t)} \geq 2$. For $t \leq T_1^{(i)}$, it holds that $\max_{j,r}\{|\bar{\Phi}_{j,r,i}^{(t)}|, |\Phi_{j,r,i}^{(t)}|\} = O(1)$ and $\max_{j,r}\Gamma_{j,r}^{(t)} \leq 4A^{(0)} = O(1)$. Therefore, by Lemma F.7 and Lemma F.6, we have $F_{-1}(\mathbf{W}^{(t)}, \mathbf{x}_i), F_{+1}(\mathbf{W}^{(t)}, \mathbf{x}_i) = O(1)$. As a result, there exists a positive constant $C_1$ such that $-\ell_i'^{(t)} \geq C_1$ for all $0 \leq t \leq T_1^{(i)}$. Additionally, it is clear that $B_i^{(0)} \geq 0.6\bar{\beta} \geq 0.15\sigma_0\sigma_\xi\sqrt{d}$. We can then analyze the dynamics of $B_i^{(t)}$.

$$B_i^{(t+1)} \geq B_i^{(t)} + \frac{C_1\eta q\|\boldsymbol{\xi}\|_2^2}{2mn}[B_i^{(t)}]^{q-1} - \eta\langle \mathbf{z}_{t+1}, \boldsymbol{\xi}\rangle$$

$$\overset{(i)}{\geq} (1 + \frac{C_1\eta q\|\boldsymbol{\xi}\|_2^2}{2mn}[B_i^{(0)}]^{q-2})B_i^{(t)} - \eta\langle \mathbf{z}_{t+1}, \boldsymbol{\xi}\rangle$$

$$\overset{(ii)}{\geq} (1 + \frac{C_1 0.15^{q-2}\eta q\sigma_0^{q-2}\|\boldsymbol{\xi}\|_2^q}{mn})B_i^{(t)} - \eta\langle \mathbf{z}_{t+1}, \boldsymbol{\xi}\rangle$$

Let $\quad q_\Phi = (1 + \frac{C_1 0.15^{q-2}\eta q\sigma_0^{q-2}\|\boldsymbol{\xi}\|_2^q}{mn})$, then $\quad \overset{(iii)}{=} q_\Phi^t B_i^{(0)} - (\frac{q_\Phi^t - 1}{q_\Phi - 1})\eta\langle \mathbf{z}_{t+1}, \boldsymbol{\xi}\rangle.$

Here, $(i)$ and $(ii)$ follow from that the facts $A^{(t)}$ is increasing and $\max_r\langle \mathbf{w}_{1,r}^{(0)}, \boldsymbol{\xi}\rangle - \eta\sum_{s=1}^{t}\langle \mathbf{z}_s, \boldsymbol{\xi}\rangle \geq 0$; $(iii)$ is the summation of geometric series. Additionally, we know that $1 + z \geq \exp(z/2)$ for $z \leq 2$ and $1 + z \leq \exp(z)$ for $z \geq 0$, then the following inequality holds

$$B_i^{(t)} \geq (1 + \frac{C_1\eta q 0.15^{q-2}\sigma_0^{q-2}\|\boldsymbol{\xi}\|_2^q}{4mn})^t B_i^{(0)} - \frac{4mn}{C_1 0.15^{q-2}q\sigma_0^{q-2}\|\boldsymbol{\xi}\|_2^q}(q_\Phi^t - 1)\cdot\frac{\|\boldsymbol{\xi}\|_2(\|\mathbf{v}\|_2 + \|\boldsymbol{\xi}\|_2)}{mn\varepsilon}$$

$$\geq \exp(\frac{C_1\eta q\sigma_0^{q-2}0.15^{q-2}\|\boldsymbol{\xi}\|_2^q}{4mn}t)\cdot 0.15\sigma_0\|\boldsymbol{\xi}\|_2$$

$$- \frac{4mn}{C_1 q 0.15^{q-2}\sigma_0^{q-2}\|\boldsymbol{\xi}\|_2^q}\cdot\exp(\frac{C_1\eta q\sigma_0^{q-2}0.15^{q-2}\|\boldsymbol{\xi}\|_2^q}{4mn}t)\cdot\frac{\|\boldsymbol{\xi}\|_2(\|\mathbf{v}\|_2 + \|\boldsymbol{\xi}\|_2)}{mn\varepsilon}$$

$$= (0.15\sigma_0\|\boldsymbol{\xi}\|_2 - \frac{4(\|\mathbf{v}\|_2 + \|\boldsymbol{\xi}\|_2)}{C_1 q\sigma_0^{q-2}\|\boldsymbol{\xi}\|_2^{q-1}\varepsilon})\cdot\exp(\frac{C_1\eta q 0.15^{q-2}\sigma_0^{q-2}\|\boldsymbol{\xi}\|_2^q}{4mn}t)$$

$$\geq 0.1\sigma_0\|\boldsymbol{\xi}\|_2\cdot\exp(\frac{C_1\eta q 0.15^{q-2}\sigma_0^{q-2}\|\boldsymbol{\xi}\|_2^q}{4mn}t),$$

where the last inequality holds due to the choice of $\sigma_0$. Therefore, it is clear that $B_i^{(t)}$ will reach 3 within $T_1 = \frac{C\log(10/(\sigma_0\sigma_\xi\sqrt{d}))4mn}{0.15^{q-2}\eta q\sigma_0^{q-2}(\sigma_\xi^2\sqrt{d})^q}$ iterations, which indicates that $\max_{j=y_i,r}\bar{\Phi}_{j,r,i}^{(t)}$ will reach 2 within $T_1^{(i)}$ iterations. Moreover, we can verify that

$$T_1 = \frac{C\log(10/(\sigma_0\sigma_\xi\sqrt{d}))4mn}{0.15^{q-2}\eta q\sigma_0^{q-2}(\sigma_\xi^2\sqrt{d})^q} \leq \eta^{-1}\sigma_0 mn\varepsilon(\|\mathbf{v}\|_2 + \|\boldsymbol{\xi}\|_2)^{-1} = T_1^+,$$

The inequality follows from the SNR condition in Equation (20). Hence, by the definition of $T_1^{(i)}$, $T_1^{(i)} \leq T_1 \leq T_1^+/2$ holds.

$\square$

## H.2. Second Stage

It is clear that we have the following results at the end of the first stage:

$$\mathbf{w}_{j,r}^{(T_1)} = \mathbf{w}_{j,r}^{(0)} + j\cdot\Gamma_{j,r}^{(T_1)}\cdot\frac{\mathbf{v}}{\|\mathbf{v}\|_2^2} + \sum_{i=1}^{n}\bar{\Phi}_{j,r,i}^{(T_1)}\cdot\frac{\boldsymbol{\xi}_i}{\|\boldsymbol{\xi}_i\|_2^2} + \sum_{i=1}^{n}\Phi_{j,r,i}^{(T_1)}\cdot\frac{\boldsymbol{\xi}_i}{\|\boldsymbol{\xi}_i\|_2^2} - \eta\sum_{s=1}^{T_1}\mathbf{z}_s$$

Meanwhile, at the beginning of the second stage, we have the following results:

- $\max_{j,r}\bar{\Phi}_{j,r,i}^{(T_1)} \geq 2$ for all $i \in [n]$.

- $\max_{j,r} \Gamma_{j,r}^{(t)} = \widetilde{O}(\sigma_0 \|\mathbf{v}\|_2)$ for all $0 \le t \le T_1$.

- $\max_{j,r,i} |\Phi_{j,r,i}^{(t)}| = \widetilde{O}(\sigma_0 \sigma_\xi \sqrt{d})$ for all $0 \le t \le T_1$.

Based on Lemma 4.2 and Lemma 4.4, we conclude that noise memorization $\bar{\Phi}_{j,r,i}^{(T_1)}$ does not deteriorate over time. Specifically, for any $T_1 \le t \le T_p^*$, it holds that $\bar{\Phi}_{j,r,i}^{(t+1)} \ge \bar{\Phi}_{j,r,i}^{(t)}$, which implies $\max_{j,r} \bar{\Phi}_{j,r,i}^{(t)} \ge 2$. If we consider $\mathbf{w}_{j,r}^* = \mathbf{w}_{j,r}^{(0)} + 2qm \log(2q/\kappa))[\sum_{i=1}^n \mathbb{1}(j = y_i) \cdot \frac{\xi_i}{\|\xi_i\|_2}]$, then we can derived:

**Lemma H.3.** *Under the same conditions as noise memorization, we have that* $\|\mathbf{W}^{(T_1)} - \mathbf{W}^*\|_F \le \widetilde{O}(m^2 n^{1/2} \sigma_\xi^{-1} d^{-1/2}) + O(nm\sigma_0)$.

**Proof of Lemma H.3.** According to triangle inequality, we have

$$\|\mathbf{W}^{(T_1)} - \mathbf{W}^*\|_F \le \|\mathbf{W}^{(T_1)} - \mathbf{W}^{(0)}\|_F + \|\mathbf{W}^{(0)} - \mathbf{W}^*\|_F$$
$$\overset{(i)}{\le} \sum_{j,r} \frac{\Gamma_{j,r}^{(T_1)}}{\|\mathbf{v}\|_2} + \sum_{j,r,i} \frac{|\bar{\Phi}_{j,r,r}^{(T_1)}|}{\|\xi_i\|_2} + \sum_{j,r,i} \frac{|\Phi_{j,r,i}^{(T_1)}|}{\|\xi_i\|_2} + \sum_{j,r} |\eta \sum_{s=1}^{T_1} \mathbf{z}_s| + O(m^{3/2} \log(1/\kappa)) \|\mathbf{v}\|_2^{-1}$$
$$\overset{(ii)}{\le} \widetilde{O}(m\|\mathbf{v}\|^{-1}) + \widetilde{O}(n\sqrt{m}\sigma_\xi^{-1} d^{-1/2}) + O(m\sigma_0) + O(m^{3/2} n^{1/2} \log(1/\kappa) \sigma_\xi^{-1} d^{-1/2})$$
$$\overset{(iii)}{\le} \widetilde{O}(m^2 n^{1/2} \sigma_\xi^{-1} d^{-1/2}).$$

Here, $(i)$ holds due to the decomposition of $\mathbf{w}$ in Definition 4.1 and the definition of $\mathbf{W}^*$; $(ii)$ follows from Proposition F.3, Lemma H.2 and Lemma F.8; $(iii)$ comes from the conditions of $\sigma_0$ in noise memorization. $\square$

**Lemma H.4.** *Under the same conditions as noise memorization, we have that* $y_i \langle \nabla f(\mathbf{W}^{(t)}, \mathbf{x}_i), \mathbf{W}^* \rangle \ge q \log(2q/\kappa)$ *for all* $i \in [n]$ *and* $T_1 \le t \le T^*$.

**Proof of Lemma H.4.** We know that

$$f(\mathbf{W}^{(t)}, \mathbf{x}_i) = (1/m) \sum_{j,r} j \cdot [\sigma(\langle \mathbf{w}_{j,r}, y_i \cdot \mathbf{v} \rangle) + \sigma(\langle \mathbf{w}_{j,r}, \xi_i \rangle)].$$

Therefore, it holds that

$$y_i \langle \nabla f(\mathbf{W}^{(t)}, \mathbf{x}_i), \mathbf{W}^* \rangle = \frac{1}{m} \sum_{j,r} \sigma'(\langle \mathbf{w}_{j,r}^{(t)}, y_i \mathbf{v} \rangle) \langle \mathbf{v}, j\mathbf{w}_{j,r}^* \rangle + \frac{1}{m} \sum_{j,r} \sigma'(\langle \mathbf{w}_{j,r}^{(t)}, \xi_i \rangle) \langle y_i \xi_i, j\mathbf{w}_{j,r}^* \rangle$$
$$\overset{(i)}{=} \frac{1}{m} \sum_{j,r} \sum_{i'=1}^n \sigma'(\langle \mathbf{w}_{j,r}^{(t)}, \xi_i \rangle) 2qm \log(2q/\kappa) \mathbb{1}(j = y_{i'}) \cdot \frac{\langle \xi_{i'}, \xi_i \rangle}{\|\xi_{i'}\|_2}$$
$$+ \frac{1}{m} \sum_{j,r} \sigma'(\langle \mathbf{w}_{j,r}^{(t)}, y_i \mathbf{v} \rangle) \langle \mathbf{v}, j\mathbf{w}_{j,r}^{(0)} \rangle + \frac{1}{m} \sum_{j,r} \sigma'(\langle \mathbf{w}_{j,r}^{(t)}, \xi_i \rangle) \langle y_i \xi_i, j\mathbf{w}_{j,r}^{(0)} \rangle$$
$$\overset{(ii)}{\ge} \frac{1}{m} \sum_{j,r} \sigma'(\langle \mathbf{w}_{j,r}^{(t)}, y_i \mathbf{v} \rangle) 2qm \log(2q/\kappa) - \frac{1}{m} \sum_{j,r} \sigma'(\langle \mathbf{w}_{j,r}^{(t)}, y_i \mathbf{v} \rangle) \widetilde{O}(\sigma_0 \|\mathbf{v}\|_2)$$
$$- \frac{1}{m} \sum_{j,r} \sigma'(\langle \mathbf{w}_{j,r}^{(t)}, \xi_i \rangle) \widetilde{O}(\sigma_0 \sigma_\xi \sqrt{d}) - \frac{1}{m} \sum_{j,r} \sigma'(\langle \mathbf{w}_{j,r}^{(t)}, \xi_i \rangle) \widetilde{O}(mnd^{-1/2}),$$

where $(i)$ holds due to the definition of $\mathbf{w}^*$ and $(ii)$ follows from Lemma E.2. Moreover, according to Lemma F.4, we have that for $j = y_i$:

$$\max_r \{\langle \mathbf{w}_{j,r}^{(t)}, \xi_i \rangle\} = \max_r \{\bar{\Phi}_{j,r}^{(t)} + \langle \mathbf{w}_{j,r}^{(0)}, \xi_i \rangle - 8n\sqrt{\frac{\log(4n^2/\delta)}{d}}\alpha - \eta \sum_{s=1}^t \langle \mathbf{z}_s, y_i \mathbf{v} \rangle\} \overset{(i)}{\ge} 1.$$

Here, $(i)$ holds due to the analysis in Lemma H.2. Additionally, we can also have

$$|\langle \mathbf{w}_{j,r}^{(t)}, \mathbf{v}\rangle| \overset{(i)}{\leq} |\langle \mathbf{w}_{j,r}^{(0)}, \mathbf{v}\rangle| + |\Gamma_{j,r}^{(t)}| + |\eta \sum_{s=1}^{t} \langle \mathbf{z}_s, \mathbf{v}\rangle| \overset{(ii)}{\leq} \widetilde{O}(1)$$

$$|\langle \mathbf{w}_{j,r}^{(t)}, \boldsymbol{\xi}_i\rangle| \overset{(iii)}{\leq} |\langle \mathbf{w}_{j,r}^{(0)}, \boldsymbol{\xi}_i\rangle| + |\underline{\Phi}_{j,r,i}^{(t)}| + |\bar{\Phi}_{j,r,i}^{(t)}| + 8n\sqrt{\frac{\log(4n^2/\delta)}{d}}\alpha + |\eta \sum_{s=1}^{t} \langle \mathbf{z}_s, \boldsymbol{\xi}_i\rangle| \overset{(iv)}{\leq} \widetilde{O}(1).$$

Here, $(i)$ holds due to Lemma F.4 and Lemma F.8; $(ii)$ is given by Lemma F.5; $(ii)$ and $(iv)$ follows from Proposition F.3 and Lemma F.8. Combining these results, we return to $y_i\langle \nabla f(\mathbf{W}^{(t)}, \mathbf{x}_i), \mathbf{W}^*\rangle$, which gives:

$$y_i\langle \nabla f(\mathbf{W}^{(t)}, \mathbf{x}_i), \mathbf{W}^*\rangle \geq 2q\log(2q/\kappa) - \widetilde{O}(\sigma_0\|\mathbf{v}\|_2) - \widetilde{O}(\sigma_0\sigma_\xi\sqrt{d}) - \widetilde{O}(mnd^{-1/2}) \geq q\log(2q/\kappa),$$

where the last inequality is driven by the conditions as noise memorization that $\varepsilon \geq 1/q\log(2q/\kappa)$. $\qquad\square$

**Lemma H.5.** *Under the same conditions as noise memorization, it holds that*

$$\|\mathbf{W}^{(t)} - \mathbf{W}^*\|_F^2 - \|\mathbf{W}^{(t+1)} - \mathbf{W}^*\|_F^2 \geq (2q-1)\eta L_D(\mathbf{W}^{(t)}) - \eta\kappa - \eta^2\widetilde{O}(d\sigma_z^2)$$
$$- \eta\widetilde{O}(\sigma_z m^2 n^{1/2}\sigma_\xi^{-1}d^{-1/2}) - O(\sigma_z nm\sigma_0)$$

*for all $T_1 \leq t \leq T^*$.*

**Proof of Lemma H.5.** According to the optimization properties, we know the first equality holds:

$$\|\mathbf{W}^{(t)} - \mathbf{W}^*\|_F^2 - \|\mathbf{W}^{(t+1)} - \mathbf{W}^*\|_F^2$$
$$= 2\eta\langle \nabla L_S(\mathbf{W}^{(t)}), \mathbf{W}^{(t)} - \mathbf{W}^*\rangle - \eta^2\|\nabla L_S(\mathbf{W}^{(t)})\|_F^2$$
$$\overset{(i)}{=} \frac{2\eta}{n}\sum_{i=1}^{n}\ell_i'^{(t)}[qy_if(\mathbf{W}^{(t)}, \mathbf{x}_i) - \langle \nabla f(\mathbf{W}^{(t)}, \mathbf{x}_i), \mathbf{W}^*\rangle] + \eta\langle \mathbf{z}_t, \mathbf{W}^{(t)} - \mathbf{W}^*\rangle$$
$$- \eta^2(O(\max\{\|\mathbf{v}\|_2^2, \sigma_\xi^2 d\})L_D(\mathbf{W}^{(t)}) + O(\sigma_z^2 d\log(1/\delta)))$$
$$\overset{(ii)}{\geq} \frac{2\eta}{n}\sum_{i=1}^{n}\ell_i'^{(t)}[qy_if(\mathbf{W}^{(t)}, \mathbf{x}_i) - q\log(2q/\kappa)] \qquad\qquad (22)$$
$$+ \eta\langle \mathbf{z}_t, \mathbf{W}^{(t)} - \mathbf{W}^*\rangle - \eta^2(O(\max\{\|\mathbf{v}\|_2^2, \sigma_\xi^2 d\})L_D(\mathbf{W}^{(t)}) + O(\sigma_z^2 d\log(1/\delta)))$$
$$\overset{(iii)}{\geq} \frac{2q\eta}{n}\sum_{i=1}^{n}[\ell(y_if(\mathbf{W}^{(t)}, \mathbf{x}_i)) - \kappa/(2q)]$$
$$\eta\widetilde{O}(\sigma_z m^2 n^{1/2}\sigma_\xi^{-1}d^{-1/2}) - \eta^2(O(\max\{\|\mathbf{v}\|_2^2, \sigma_\xi^2 d\})L_D(\mathbf{W}^{(t)}) + O(\sigma_z^2 d\log(1/\delta)))$$
$$\overset{(iv)}{\geq} (2q-1)\eta L_D(\mathbf{W}^{(t)}) - \eta\kappa - \eta^2\widetilde{O}(d\sigma_z^2) - \eta\widetilde{O}(\sigma_z m^2 n^{1/2}\sigma_\xi^{-1}d^{-1/2}).$$

Here, $(i)$ holds due to the definition of noisy gradient, the neural network is $q$ homogeneous, and Lemma F.9; $(ii)$ is driven from Lemma H.4; $(iii)$ is due to the convexity of the cross entropy function; $(iv)$ comes from definition of $L_D$. $\qquad\square$

**Lemma H.6** (Restatement of Corollary 4.11). *Let $T, T_1$ be defined in respectively. Then under the same conditions as signal learning, for any $t \in [T_1, T]$, it holds that $|\Gamma_{j,r}^{(t)}| \leq \sigma_0\|\mathbf{v}\|_2$ for all $j \in \{\pm 1\}$ and $r \in [m]$. Moreover, let $\mathbf{W}^*$ be the collection of CNN parameters with convolution filters $\mathbf{w}_{j,r}^* = \mathbf{w}_{j,r}^{(0)} + 2qm\log(2q/\kappa))[\sum_{i=1}^{n}\mathbb{1}(j = y_i) \cdot \frac{\boldsymbol{\xi}_i}{\|\boldsymbol{\xi}_i\|_2}]$. Then the following bound holds*

$$\frac{1}{t - T_1 + 1}\sum_{s=T_1}^{t} L_D(\mathbf{W}^{(s)}) \leq \frac{\|\mathbf{W}^{(T_1)} - \mathbf{W}^*\|_F^2}{(2q-1)\eta(t - T_1 + 1)} + \frac{\kappa}{(2q-1)} + \underbrace{\frac{\eta d\sigma_z^2 + \widetilde{O}(\sigma_z m^2 n^{1/2}\sigma_\xi^{-1}d^{-1/2})}{(2q-1)}}_{Private\ terms}$$

*for all $t \in [T_1, T]$, where we denote $\|\mathbf{W}\|_F = \sqrt{\|\mathbf{W}_{+1}\|_F^2 + \|\mathbf{W}_{-1}\|_F^2}$.*

**Proof of Lemma H.6.** According to Lemma G.4, we have, for any $t \leq T$:

$$\|\mathbf{W}^{(t)} - \mathbf{W}^*\|_F^2 - \|\mathbf{W}^{(t+1)} - \mathbf{W}^*\|_F^2 \geq (2q-1)\eta L_D(\mathbf{W}^{(t)}) - \eta\kappa - \eta^2 \widetilde{O}(d\sigma_z^2) - \eta\widetilde{O}(\sigma_z m^{3/2} \|\mathbf{v}\|_2^{-1}).$$

By summing over all terms and dividing $t - T_1 - 1$ on both sides, we obtain:

$$\frac{1}{t - T_1 + 1} \sum_{s=T_1}^{t} L_D(\mathbf{W}^{(s)}) \leq \frac{\|\mathbf{W}^{(T_1)} - \mathbf{W}^*\|_F^2}{(2q-1)\eta(t - T_1 + 1)} + \frac{\kappa}{(2q-1)} + \frac{\eta d\sigma_z^2 + \widetilde{O}(\sigma_z m^2 n^{1/2} \sigma_\xi^{-1} d^{-1/2})}{(2q-1)}.$$

If we have $T = T_1 + \lfloor \frac{\|\mathbf{W}^{(T_1)} - \mathbf{W}^*\|_E^2}{2\eta\kappa} \rfloor = \frac{Cmn\varepsilon}{\eta\mu(\|\mathbf{v}\|_2 + \|\xi\|_2)} \geq \kappa^{-1}$, then it holds that

$$\frac{\|\mathbf{W}^{(T_1)} - \mathbf{W}^*\|_F^2}{(2q-1)\eta(T - T_1 + 1)} + \frac{\kappa}{2q - 1} \leq \frac{3\kappa}{2q - 1},$$

and with $\sigma_z = \frac{1}{\eta\mu\sqrt{T}}$:

$$\frac{\eta d\sigma_z^2 + \widetilde{O}(\sigma_z m^2 n^{1/2} \sigma_\xi^{-1} d^{-1/2})}{(2q-1)} \leq \frac{d}{\eta\mu^2 T(2q-1)} + \frac{m^2 n^{1/2} \|\xi\|_2^{-1} \|\mathbf{v}\|_2^{-1}}{\eta\mu\sqrt{T}(2q-1)} \overset{(i)}{\leq} \frac{\kappa}{(2q-1)},$$

where $(i)$ comes from the assumption of $\eta$. Therefore, combining above results, we conclude that

$$\frac{1}{t - T_1 + 1} \sum_{s=T_1}^{t} L_D(\mathbf{W}^{(s)}) \leq \kappa.$$

Moreover, we will use induction to prove that $\max_{j,t} |\Gamma_{j,r}^t| \leq 2\sigma_0 \|\mathbf{v}\|_2$ holds for all $t \in [T_1, T]$. According to Lemma G.1, we know it holds for $T_1$. Now, assume it holds for some $t \in [T_1, T)$, and we will show that it also holds for $t + 1$.

$$\Gamma_{j,r}^{(t)} = \Gamma_{j,r}^{(T_1)} - \frac{\eta}{nm} \sum_{s=T_1}^{t-1} \sum_{i=1}^{n} \ell_i^{(t)} \cdot \sigma'(\langle \mathbf{w}_{j,r}^{(0)}, y_i \cdot \mathbf{v} \rangle + \Gamma_{j,r}^s - \langle \mathbf{z}_s, \mathbf{v} \rangle) \|\mathbf{v}\|_2^2,$$

$$\overset{(i)}{\leq} \Gamma_{j,r}^{(T_1)} + \frac{q5^{q-1}\eta}{nm} \|\mathbf{v}\|_2^2 (\sigma_0 \|\mathbf{v}\|_2)^{q-1} \sum_{s=T_1}^{t-1} \sum_{i=1}^{n} |\ell_i^{(t)}|$$

$$\overset{(ii)}{\leq} \Gamma_{j,r}^{(T_1)} + q5^{q-1}\eta m^{-1} \|\mathbf{v}\|_2^2 (\sigma_0 \|\mathbf{v}\|_2)^{q-1} \sum_{s=T_1}^{t-1} L_S(\mathbf{W}^{(s)})$$

$$\overset{(iii)}{\leq} \Gamma_{j,r}^{(T_1)} + (\sigma_0 \|\mathbf{v}\|_2)^{q-1} \widetilde{O}(m^2 n \, \text{SNR}^2)$$

$$\overset{(iv)}{\leq} \Gamma_{j,r}^{(T_1)} + (\sigma_0 \|\mathbf{v}\|_2)(n\varepsilon)^{-(q-2)/q} \widetilde{O}(m^2 n^{1-2/q})$$

$$\overset{(v)}{\leq} 2\widehat{\beta}'.$$

Here, $(i)$ is due to induction hypothesis and the choice of $T$; $(ii)$ holds by $|\ell'| \leq \ell$; $(iii)$ comes from Lemma H.3 and the choice of $T$; $(iv)$ is driven from $\sigma_0 \leq (n\varepsilon)^{-1/q} \|\mathbf{v}\|_2^{-1}$ and SNR; $(v)$ holds due to $n^{1/q}\varepsilon \geq m$. $\qquad\square$

**Lemma H.7** (Restatement of Corollary 4.12)**.** *Under the same conditions as data noise memorization, within $T$ iterations, regardless of how the sample size $n$ and privacy budget $\varepsilon$ chosen, with at least probability $1 - 1/d$, we can find $\mathbf{W}^{(\widetilde{T})}$ such that $L_D(\mathbf{W}^{(\widetilde{T})}) \leq \kappa$. Additionally, for any $0 \leq t \leq \widetilde{T}$ we have that $L_{\mathcal{D}}(\mathbf{W}^{(t)}) \geq 0.1$.*

**Proof of Lemma H.7.** Consider a new sample $(\mathbf{x}, y)$ drawn from Definition 3.1, we have:

$$\|\mathbf{w}_{j,r}^{(t)}\|_2 = \|\mathbf{w}_{j,r}^{(0)} + j \cdot \Gamma_{j,r}^{(t)} \cdot \frac{\mathbf{v}}{\|\mathbf{v}\|_2^2} + \sum_{i=1}^{n} \bar{\Phi}_{j,r,i}^{(t)} \cdot \frac{\xi_i}{\|\xi_i\|_2^2} + \sum_{i=1}^{n} \underline{\Phi}_{j,r,i}^{(t)} \cdot \frac{\xi_i}{\|\xi_i\|_2^2} - \eta \sum_{s=1}^{t} \mathbf{z}_s\|_2$$

$$\overset{(i)}{\leq} \|\mathbf{w}_{j,r}^{(0)}\|_2 + \frac{\Gamma_{j,r}^{(t)}}{\|\mathbf{v}\|_2} + \sum_{i=1}^{n} \frac{\bar{\Phi}_{j,r,i}^{(t)}}{\|\xi_i\|_2} + \sum_{i=1}^{n} \frac{|\underline{\Phi}_{j,r,i}^{(t)}|}{\|\xi_i\|_2} + \|\eta \sum_{s=1}^{t} \mathbf{z}_s\|_2$$

$$\overset{(ii)}{\leq} O(\sigma_0 \sqrt{d}) + \widetilde{O}(n\sigma_\xi^{-1} d^{-1/2}) + O(\eta\sqrt{t}d\sigma_z)$$

Here, $(i)$ is due to triangle inequality; $(ii)$ holds by Lemma H.2 and Proposition F.3. Additionally, we know that $\langle \mathbf{w}_{j,r}^{(t)}, \boldsymbol{\xi} \rangle \sim \mathcal{N}(0, \sigma_\xi^2 \|\mathbf{w}_{j,r}^{(t)}\|_2^2)$, it holds that with probability at least $1 - 1/4$, $|\langle \mathbf{w}_{j,r}^{(t)}, \boldsymbol{\xi} \rangle| \leq \widetilde{O}(\sigma_0 \sigma_\xi \sqrt{d} + nd^{-1/2} + \frac{\sqrt{d}\sigma_\xi \sigma_0}{\mu})$ due to $\sigma_z = \sigma_0/(\eta\sqrt{T}\mu)$. Moreover, according to Lemma H.6, we have $\max_{j,r} \Gamma_{j,r}^{(t)} \leq \widetilde{O}(\sigma_0\|\mathbf{v}\|_2)$, which also indicates that $|\langle \mathbf{w}_{j,r}^{(t)}, \mathbf{v} \rangle| \leq \widetilde{O}(\sigma_0\|\mathbf{v}\|_2)$.

Then, by the union bound, with probability at least $1 - 1/2$, we have

$$
\begin{aligned}
F_j(\mathbf{W}_j^{(t)}, \mathbf{x}) &= \frac{1}{m}\sum_{r=1}^m \sigma(\langle \mathbf{w}_{j,r}^{(t)}, y\mathbf{v} \rangle) + \frac{1}{m}\sum_{r=1}^m \sigma(\langle \mathbf{w}_{j,r}^{(t)}, \boldsymbol{\xi} \rangle) \\
&\leq \max_r |\langle \mathbf{w}_{j,r}^{(t)}, \mathbf{v} \rangle|^q + \max_r |\langle \mathbf{w}_{j,r}^{(t)}, \boldsymbol{\xi} \rangle|^q \\
&\leq \widetilde{O}(\sigma_0^q \sigma_\xi^q d^{q/2} + n^q d^{-q/2} + \sigma_0^q \|\mathbf{v}\|_2^q + \frac{\sigma_0^q \|\boldsymbol{\xi}\|_2^q}{\mu^q}) \\
&\overset{(i)}{\leq} \widetilde{O}(\frac{1}{\varepsilon^q} + n^q d^{-q/2} + \frac{1}{n\varepsilon} + \frac{1}{\varepsilon^q \mu^q}) \\
&\overset{(ii)}{\leq} 1.
\end{aligned}
$$

Here, $(i)$ and $(ii)$ holds due to we restrict $\sigma_0 = \widetilde{O}(\varepsilon^{-1}\|\boldsymbol{\xi}\|_2)$ and $\varepsilon^q \geq \widetilde{O}(1)$. Notice that here 1 can be any number, and we use 1 without loss of generality. Therefore, with probability at least $1 - 1/2$, we have $\ell(y \cdot f(\mathbf{W}^{(t)}, \mathbf{x})) \geq \log(1 + e^{-1})$, which indicates that $L_\mathcal{D}(\mathbf{W}^{(t)}) \geq \log(1 + e^{-1}) \cdot 0.5 \geq 0.1$. $\qquad\square$

