# OpenReview forum: "Understanding Private Learning From Feature Perspective"
_ICML.cc/2026/Conference — ICML 2026 regular_

### Official Review · Reviewer_W4LF · 2026-03-07

**Soundness:** 3
**Presentation:** 3
**Significance:** 3
**Originality:** 3
**Overall Recommendation:** 5
**Confidence:** 4

**Summary:**

This paper introduces a data distribution model that decomposes input data into label-dependent feature signals $\mathbf{v}$ and label-independent noise $\xi$, , with signal-to-noise ratio defined as $SNR = \lVert\mathbf{v}\rVert/\lVert\xi\rVert.$ A two-layer CNN with polynomial ReLU activation is trained using NoisyGD, which minimizes logistic loss while updating filters via the Gaussian mechanism. Under this decomposition, the training dynamics of DP-GD can be described as a linear combination of the random initialization, feature signals $\mathbf{v}$, data noise $\xi$, and accumulated private noise. Based on this analysis, the authors conclude: (1) Private learning requires a higher SNR than non-private training for effective feature extraction; (2) If noise is memorized during non-private training, it will persist under DP-GD once it exceeds an SNR-dependent threshold, leading to poor generalization despite low training loss.

**Compliance With Llm Reviewing Policy:**

Affirmed.

**Ethics Expertise Needed:**

["Privacy and Security (e.g., personally identifiable information)"]

**Final Justification:**

I have no further questions and will raise my rating to 5.

**Key Questions For Authors:**

Please refer to the weaknesses part.

**Limitations:**

Yes

**Strengths And Weaknesses:**

Strengths: By introducing a signal-noise decomposition, this work offers a novel perspective on feature learning in DP training. This insight facilitates the development of more effective privacy-preserving models by improving feature extraction and noise suppression. Moreover, the problem studied is important, and the theoretical analysis is solid.

Weakness and questions:

1. The theoretical framework is limited to shallow neural networks. It would be great if the authors could discuss the potential extensions to architecture includes deeper layers or complex transformer structures.

2. Some assumptions need further clarification. In Condition 4.5, the first part states that $d$ is a polynomial of $m,n$, while the second part states that $m,n$ are polynomials of $\log d$. The order appears inconsistent between these two conditions. Although the authors cite several works supporting these assumptions in the non-private setting, additional details on the order relationships would help readers better understand the theory.


3. The relationship between convergence, data dimension, and model size seems overly complicated in the current theory. Including a short remark to explain how model size and data dimension affect convergence would improve the paper’s clarity and accessibility.


5. Two related works on feature learning under DP are not cited: [A] empirically investigates the representations (features) of foundation models. [B] provides a theoretical analysis of private learning for a two-layer neural network. Both works highlight that large noise may distort learned features. It would be helpful if the authors could discuss whether and how large noise might distort features in their setting, or suggest this as a direction for future work. Moreover, a more recent work [C] highlighted the importance of optimal hyperparameters especially the learning rate in private fine-tuning.

[A] Why Does Private Fine-Tuning Resist Differential Privacy Noise? A Representation Learning Perspective. Xia et al., 2025

[B] Characterizing the Training Dynamics of Private Fine-tuning with Langevin diffusion. Ke et al., 2024.

[C] On Optimal Hyperparameters for Differentially Private Deep Transfer Learning. Rehn et al., 2025.

---

> ### Author Rebuttal · Authors · 2026-03-30
>
> We sincerely thank Reviewer W4LF for the valuable time, constructive suggestions, and positive feedback on our work. We hope to address your concerns accordingly.
>
> **Responses to Question 1:** Thanks for the question. We believe our analysis can potentially be extended to other deep learning models such as transformer structures, as long as their parameter updates can be decomposed in a form similar to Definition 4.1 and the corresponding coefficients remain traceable throughout training. That said, whether such an extension is possible depends on the specific model architecture and training problem, and establishing it is beyond the scope of the current paper. We view this as an important direction for future work.
>
> **Responses to Question 2:** Thanks for the comment. We agree that the relationship among $d, m$, and $n$ in Condition 4.5 should be clarified more explicitly. The two conditions are intended to describe a compatible asymptotic regime rather than contradictory requirements: the first requires $d$ to be sufficiently large relative to polynomial functions of $m$ and $n$, while the second only requires $m$ and $n$ to be at least polylogarithmic in $d$. For example, one may take $m=(\log d)^2$ and $n=(\log d)^3$. Then the condition $m, n=\Omega($ poly $\log d)$ is satisfied, while the requirement on $d$ becomes $d \geq \widetilde{\Omega}\left(m^2 n^4\right)=\widetilde{\Omega}\left((\log d)^{16}\right)$, which holds for sufficiently large $d$. We will revise the paper to clarify this scaling relationship and explain more clearly how these assumptions are compatible.
>
> **Responses to Question 3:** Thanks for the comment. We would like to clarify that the main goal of our theory is not to characterize a clean convergence rate in terms of data dimension and model size, but rather to understand how private noise affects feature signal learning and data noise memorization. For this reason, the convergence-related discussion only appears in the corollaries. Additionally, the effect of data dimension and model size on convergence is not only direct, and it also enters indirectly through $T$. We will clarify the role of these parameters and improve readability in the revised version.
>
> **Responses to Question 4:** Thanks for the helpful comment. We will add and discuss these related works in the revised version. We agree that studying how large private noise distorts learned features is an important direction. Our current analysis is closer to a training-from-scratch setting, rather than private fine-tuning with pretrained representations.
> To model fine-tuning more faithfully, the framework would likely need to be reformulated: the data may need to distinguish pretrained features from new task-specific features, and the model updates should start from a pretrained initialization rather than a random initialization. At this stage, it is unclear whether our current conclusions would still hold in that setting, but we believe this is a very interesting direction for future work.

---

> > ### Author Rebuttal · Reviewer_W4LF · 2026-04-02
> >
> > My concerns have been adequately addressed.

---

> > > ### Author Response · Authors · 2026-04-02
> > >
> > > Thank you again for your positive feedback. We are very glad to hear that your concerns have been addressed. If you find it appropriate, we would sincerely appreciate it if you could consider updating your evaluation accordingly. We would also be happy to clarify any remaining questions during the discussion period!

---

### Official Review · Reviewer_xpfG · 2026-03-12

**Soundness:** 3
**Presentation:** 3
**Significance:** 3
**Originality:** 3
**Overall Recommendation:** 5
**Confidence:** 4

**Summary:**

The author uses the Feature Perspective theorem to analyze NoisyDG (a simple version of DP-SGD) on a 2-layer CNN with polynomial ReLU of order q and shows that, in DP Learning, stricter privacy budgets require higher SNR than in ordinary training to ensure the model learns the signal. If the SNR of the training data is too low, data noise memorization occurs in standard non-private training and in DP Learning with a sufficiently large privacy budget.

The author first proves that the coefficients of signal and data noise are bounded by a logarithmic function of the training step, with the step upper-bounded by a polynomial function of hyperparameters, and that the private noise's influence on the task-relevant space is also bounded. Also, the author proves that their "NoisyGD" satisfies (epsilon, delta)-DP with high probability in this section.

Then the author demonstrates that with sufficient SNR and DP epsilon and by conditionally guaranteeing the model's ability to learn feature signals, the data noise coefficient is smaller than the feature signal coefficient so that the model can learn features instead of data noise memorization, and private learning requires stronger features (larger SNR) than non-private learning. Moreover, the author demonstrates that, under a stronger assumption, private learning can achieve training loss comparable to that of non-private learning and testing loss that scales with DP epsilon.

Then the author studies the scenario in which the model primarily learns data noise rather than the feature signal and shows that, with sufficient privacy budget or small SNR, the data noise coefficient is smaller than the feature signal coefficient, leading to data noise memorization. Moreover, the author demonstrates that, with an additional assumption, private learning can achieve training loss comparable to that of non-private learning but exhibits poor generalization.

At the end, the author verifies their result by experimenting on the same dataset and CNN setting, and the experimental result aligns with the theoretical analysis. The author also experiments on CIFAR-10 and demonstrates that higher SNR improves model accuracy across various privacy budgets.

**Compliance With Llm Reviewing Policy:**

Affirmed.

**Final Justification:**

I have no further questions. The authors have addressed my concerns during the rebuttal. I will keep my positive assessment.

**Key Questions For Authors:**

Q1. Is it possible to generalize the proposed theory to more complex and commonly used deep learning models?

**Limitations:**

The authors do not discuss the limitations of their theoretical analysis of privacy guarantees and the gap between NoisyGD and DP-SGD.

**Strengths And Weaknesses:**

**Strengths**
- This work is technically sound. The author provides a sound theoretical analysis using methods and techniques widely used in this field, and the assumption on which the proof is based is also widely used in other works in this field. Also, this work uses an empirical experiment to verify the theoretical analysis result.

- This work is well-presented, well-structured, and clearly written. The author constructs the proof with a clear structure and clearly articulates the formal rigor, which makes the submission easy to follow.

- This paper, for the first time, adopts a feature-learning perspective to analyze DP-SGD for CNNs, which is widely used in the modern privacy-preserving deep learning realm, and compares it with linear models, providing a clear condition for when the model will learn features and when it will learn data noise.

- This paper firstly uses the feature perspective theorem to analyze DP-SGD on a nonlinear complex model.


**Weaknesses**
- The real-world data experiment does not clearly show the numerical relationship given by the theoretical analysis, which only shows the trend predicted by the theoretical analysis.

- This paper only verifies the theoretical analysis about model utility (e.g., coefficient of feature signal or coefficient of data noise) through experiments, while it does not verify the theoretical analysis about privacy guarantees (e.g., the NoisyGD satisfies (epsilon, delta)-DP w.h.p).

- The gap between DP-SGD and NoisyGD may make the theoretical result not hold in DP-SGD, where the bias introduced by clipping may change the learning dynamics, which limits the theoretical result's applicability to real-world privacy-preserving deep learning.

---

> ### Author Rebuttal · Authors · 2026-03-30
>
> We sincerely thank Reviewer xpfG for the valuable time, constructive suggestions, and positive feedback on our work. We hope to address your concerns accordingly.
>
> **Responses to Weaknesses 1:** Thanks for the comment. We agree that the real-world experiment does not aim to verify the exact numerical relationship in the theory. Instead, its purpose is to provide qualitative evidence for the trend predicted by the analysis, namely that stronger effective signal relative to noise leads to better private learning performance. In realistic datasets and models, many additional factors make an exact numerical match difficult, so we view these experiments as supporting the theoretical intuition rather than as a precise validation of the derived theoretical results.
>
> **Responses to Weaknesses 2:** Thanks for the comment. We agree that our experiments only validate the utility-side predictions of the theory and do not empirically verify the privacy guarantee itself. This is because the DP guarantee is established theoretically rather than through empirical evaluation. In that sense, privacy here is a formal property proved under our assumptions, while the experiments are mainly used to examine whether the predicted learning dynamics appear in practice.
>
> **Responses to Weaknesses 3:** Thanks for the comment. We agree that there is a gap between the analyzed NoisyGD and practical DP-SGD. Our current results should be viewed as a first step toward analyzing private learning from a feature perspective, and thus we adopt a simplified setting to make the dynamics tractable.
> We also agree that extending the analysis from NoisyGD to clipped DP-SGD is an important direction for future work. In particular, it would be valuable to incorporate a clipping threshold and the corresponding clipping factors into the dynamics and final bounds, which may help bridge the gap between theory and real-world privacy-preserving deep learning. We will add this limitation and discussion to the revised version.
>
> **Responses to Question 1:** Thanks for the question. We believe our analysis can potentially be extended to other deep learning models, as long as their parameter updates can be decomposed in a form similar to Definition 4.1 and the corresponding coefficients remain traceable throughout training. That said, whether such an extension is possible depends on the specific model architecture and training problem, and establishing it is beyond the scope of the current paper. We view this as an important direction for future work.

---

> > ### Author Rebuttal · Reviewer_xpfG · 2026-04-03
> >
> > I have no further questions.

---

> > > ### Author Response · Authors · 2026-04-03
> > >
> > > Thank you again for your positive feedback. We are very glad to hear that your concerns have been addressed. We would also be happy to clarify any remaining questions during the discussion period!

---

### Official Review · Reviewer_sRMk · 2026-03-12

**Soundness:** 3
**Presentation:** 3
**Significance:** 2
**Originality:** 3
**Overall Recommendation:** 4
**Confidence:** 3

**Summary:**

This paper studies private learning from feature learning perspective. Most of the contribution is from theoretical side. The setting is based on multi-patch data model. They authors decompose the input into label-dependent feature signal and label-independent data noise and analyze how these two components behave during training under a two-layer CNN trained with noisyGD(without clipping). The main takeaway is that the private training requires higher signal-to-noise ratio than non-private training to successfully learn label-dependent features. Also, the data noise memorization is not automatically eliminated by privacy noise and can lead to poor generalization despite small training loss. The authors also provide qualitative story with synthetic experiments on small scaled dataset.

**Compliance With Llm Reviewing Policy:**

Affirmed.

**Key Questions For Authors:**

My main question is about how do we use the conclusion from this paper. I mean the theoretical SNR here is very clean under the patch model, but for practice it would be helpful to explain how one should think about, estimate, or improve SNR when using pretrained representations or realistic image models.

Other questions:
1. Could the authors clarify the derivation of the statement following theorem 4.10 that the exceptional regime only occurs when epsilon\leq n^{-q}?
2. In theorem 4.6, is the extra private-learning threshold SNR n \epsilon information-theoretically necessary? Or is it mainly an artifact of the proof technique and good-event sensitivity control in Lemma 4.4/proposition 4.3?

**Limitations:**

There is no negative societal impact. Like I have explained in the previous section. It would be better if authors could be up front about the technical limitations.

**Strengths And Weaknesses:**

Strengths:
1. I think majority of the DP learning community focuses on optimization bounds or through utility. This paper studies the private learning from feature signal perspective. The concept is interesting and potentially provide some more insights for future works. Also the theoretical structure is nontrivial.

Weaknesses:
1. The paper defines standard approximate-DP but the result (line 237) only shows that it holds under good event, which is distribution dependent and weakens the DP guarantee.
2. This is also related to the previous one. The analyzed NoisyGD mechanism does not have clipping. In practice, the clipping could completely change the training dynamics.
3. The data-patch model is useful but might be too stylized. Like for example, in real image, the background or texture might not be pure label independent noise. It can correlate with the class and semantic feature. Currently it assumes there is a clean separation between signal and noise.

I understand that these assumptions might be from prior works, it would be better if the authors could briefly acknowledge the limitation of above assumptions somewhere in the paper to provide a complete story.

---

> ### Author Rebuttal · Authors · 2026-03-30
>
> We sincerely thank Reviewer sRMk for the valuable time, constructive suggestions, and positive feedback on our work. We hope to address your concerns accordingly.
>
> **Responses to Weaknesses 1 and 2:** Thanks for pointing this out. We agree that our current analysis considers only NoisyGD without gradient clipping, which yields a weaker privacy guarantee. We will clarify this limitation in the revised version. We also agree that clipping may change the training dynamics. As future work, we hope to extend the analysis to clipped NoisyGD / DP-SGD by introducing a clipping threshold C, and then tracking the corresponding clipping factors, which may depend on C and the data-attribution terms, in the final bounds. We will add this discussion to the revised version.
>
> **Responses to Weaknesses 3:**  Thanks for this insightful comment. We agree that our patch-based data model is stylized and assumes a clean separation between label-dependent signal and label-independent noise. Since this is, to our knowledge, a first step toward analyzing private learning from a feature perspective, we adopt a simplified setting to make the dynamics tractable. In future work, we plan to study more realistic data models as you suggested. For example, the noise/background component may be partially label-dependent, such as $A_y \xi$, where its distribution varies with the class label. More generally, one could consider contextual features that are correlated with labels but are not the core semantic signal, such as skies for airplanes or trees/background texture for birds. We will add this limitation and discussion to the revised version.
>
> **Responses to Question 1:**  Thanks for this helpful question. We agree that the SNR in our theory is a clean quantity defined under a stylized patch model, and is not meant to be directly computed in realistic image models. Rather, we view it as a conceptual measure of how strong a label-relevant signal is relative to private learning. From this perspective, pretrained representations, feature preprocessing, and stronger denoising all help private learning by improving the effective SNR. In practice, one may estimate an effective notion of SNR through simple representation-level statistics, e.g., whether samples from the same class cluster tightly while samples from different classes are well separated. We will clarify this practical interpretation and add discussion in the revised version.
>
> **Responses to Question 2:**  Thanks for the question. Before presenting the derivation of Theorem 4.10, we first provide a high-level overview of our proof. Our framework begins by decomposing the model updates into interpretable components. Specifically, Definition 4.1 decomposes the model weights into initialization, feature signal learning ( $\Gamma$ ), data noise memorization ( $\Phi$ ), and accumulated private noise. Lemma 4.2 then characterizes the dynamics of these components, showing how private noise affects both processes. Proposition 4.3 further provides bounds for each component under the corresponding conditions.
> The result of Theorem 4.10 is then established by induction. In particular, by writing out the dynamics of $\Phi$ according to Lemma 4.2, and combining the bounds from Proposition 4.3 and Lemma 4.4, we can show that the induction goes through when $\epsilon \leq n^{-q}$, due to the iteration constraints in Proposition 4.3.
>
> **Responses to Question 3:** Thanks for the insightful question. At this stage, we do not claim that the extra threshold SNR $\cdot n \epsilon$ is information-theoretically necessary. In our current analysis, it arises from the proof technique, in particular from controlling the cumulative effect of private noise and the bounded-sensitivity argument in Lemma 4.4 / Proposition 4.3. Establishing whether this threshold is intrinsic or can be improved by a sharper analysis, is an interesting direction for future work.

---

> > ### Author Rebuttal · Reviewer_sRMk · 2026-04-03
> >
> > My questions have been adequately addressed. Thanks authors for the explanations.

---

> > > ### Author Response · Authors · 2026-04-03
> > >
> > > Thank you again for your positive feedback. We are very glad to hear that your concerns have been addressed. If you find it appropriate, we would sincerely appreciate it if you could consider updating your evaluation accordingly. We would also be happy to clarify any remaining questions during the discussion period

---

### Decision · Program_Chairs · 2026-04-30

**Decision:**

Accept (regular)

**Comment:**

This paper studies private learning from feature learning perspective. The author uses the Feature Perspective theorem to analyze NoisyDG (a simple version of DP-SGD) on a 2-layer CNN with ReLU. All reviewers are in agreement as to the novelty in this work, all are impressed by the sound theoretical work and all seem to think this work can and should stem interesting further discussion. We thus recommend acceptance.